# Structured Preconditioners in Adaptive Optimization: A Unified Analysis

**Shuo Xie**[1] **Tianhao Wang**[1] **Sashank Reddi**[2] **Sanjiv Kumar**[2] **Zhiyuan Li**[1 2]

## Abstract

We present a novel unified analysis for a broad class of adaptive optimization algorithms with structured (e.g., layerwise, diagonal, and kronecker-factored) preconditioners for both online regret minimization and offline convex optimization. Our analysis not only provides matching rate to several important structured preconditioned algorithms including diagonal AdaGrad, full-matrix AdaGrad, and AdaGrad-Norm, but also gives an improved convergence rate for a one-sided variant of Shampoo over that of original Shampoo. Interestingly, more structured preconditioners (e.g., diagonal Adagrad, AdaGrad-Norm which use less space and compute) are often presented as computationally efficient approximations to full-matrix Adagrad, aiming for improved optimization performance through better approximations. Our unified analysis challenges this prevailing view and reveals, perhaps surprisingly, that more structured preconditioners, despite using less space and computation per step, can outperform their less structured counterparts. To demonstrate this, we show that one-sided Shampoo, which is relatively much cheaper than full-matrix AdaGrad could outperform it both theoretically and experimentally.

## 1. Introduction

Adaptive optimization algorithms (Streeter & McMahan, 2010; Duchi et al., 2011; Kingma & Ba, 2014) play a pivotal role in modern machine learning, especially in the expensive training of large foundation models. Within the machine learning community, full-matrix AdaGrad is considered as an ideal adaptive preconditioner for fast convergence in terms of number of steps. The computation of full-matrix AdaGrad precondtioner typically involves inverse square root of a $d \times d$ matrix where $d$ is the number of parameters. Thus, for large-scale settings, the huge computation and memory cost makes it prohibitively expensive. This has inspired works on designing more efficient adaptive optimizers by using a structured preconditioner, such as coordinate-wise adaptivity employed by AdaGrad (Streeter & McMahan, 2010; Duchi et al., 2011), Kronecker product based preconditioner employed by Shampoo (Gupta et al., 2018; Anil et al., 2020) and layerwise adaptivity employed by LARS (You et al., 2017) and LAMB (You et al., 2019)), which all aim to provide a computational and memory efficient approximation of full-matrix AdaGrad. These algorithms have been shown to be very effective for general deep learning settings. It is often assumed that a better approximation of full-matrix preconditioner usually results in better optimization convergence, as seen with methods like Shampoo (Gupta et al., 2018). In contrast, preconditioners with more structure such as diagonal preconditioner have inferior performance. In this paper, we challenge these prevailing notions by providing theoretical and empirical evidence against them.

Conceptually, one could equate the degree of structure in a preconditioner to the ease of its computation and storage. In this view, full-matrix AdaGrad can be considered as the least structured and most expensive preconditioner while AdaGrad-Norm (Ward et al., 2020), which only maintains a scalar and uses the same preconditioning for every direction, is the most structured and least expensive. The conventional wisdom here is that using a less structured preconditioner, which requires more space and compute per step, reduces the number of steps needed for training. Thus, choosing the right structure balances this trade-off between convergence speed and training step cost.

Among these preconditioning methods, Shampoo (Gupta et al., 2018) has gained notable attention for its Kronecker-factored preconditioning approach, which promises improved convergence in large-scale optimization tasks (Dahl et al., 2023). Shampoo was originally proposed as a computationally efficient surrogate for full-matrix AdaGrad (Duchi et al., 2011). Despite its popularity, existing analyses of Shampoo (Gupta et al., 2018) are limited and do not provide a full justification for its effectiveness. In particular, we argue that the best-known regret bounds for both Shampoo as well as full-matrix AdaGrad are consistently worse

---

[1]Toyota Technological Institute at Chicago [2]Google Research. Correspondence to: Shuo Xie <shuox@ttic.edu>, Zhiyuan Li <zhiyuanli@ttic.edu>.

*Proceedings of the $42^{nd}$ International Conference on Machine Learning*, Vancouver, Canada. PMLR 267, 2025. Copyright 2025 by the author(s).

| Algorithm | Subalgebra $\mathcal{K}$ | $\|\boldsymbol{x}\|_{\mathcal{H}}$ | $\|\|\boldsymbol{g}_{1:T}\|\|_{\mathcal{H}}$ | Regret Bound $= \|\mathcal{X}\|_{\mathcal{H}} \cdot \|\|\boldsymbol{g}_{1:T}\|\|_{\mathcal{H}}$ |
|---|---|---|---|---|
| AdaGrad-Norm | $c \cdot \boldsymbol{I}_d$ for $c \in \mathbb{R}$ | $\frac{\|\boldsymbol{x}\|_2}{\sqrt{d}}$ | $\sqrt{d}\sqrt{\sum\limits_{t=1}^{T}\|\boldsymbol{g}_t\|_2^2}$ | $\|\mathcal{X}\|_2 \sqrt{\sum\limits_{t=1}^{T}\|\boldsymbol{g}_t\|_2^2}$ (Streeter & McMahan, 2010) |
| AdaGrad | Diagonal Matrices | $\|\boldsymbol{x}\|_\infty$ | $\sum\limits_{i=1}^{d}\sqrt{\sum\limits_{t=1}^{T}g_{t,i}^2}$ | $\|\mathcal{X}\|_\infty \sum\limits_{i=1}^{d}\sqrt{\sum\limits_{t=1}^{T}g_{t,i}^2}$ (Streeter & McMahan, 2010) |
| Full-matrix AdaGrad | All Matrices | $\|\boldsymbol{x}\|_2$ | $\mathrm{Tr}[(\sum\limits_{t=1}^{T}\boldsymbol{g}_t\boldsymbol{g}_t^\top)^{\frac{1}{2}}]$ | $\|\mathcal{X}\|_2 \, \mathrm{Tr}\left[\left(\sum\limits_{t=1}^{T}\boldsymbol{g}_t\boldsymbol{g}_t^\top\right)^{\frac{1}{2}}\right]$ (Duchi et al., 2011) |
| One-sided Shampoo | $\mathbb{R}^{d_L \times d_L} \otimes \boldsymbol{I}_{d_R}$ | $\frac{\|\boldsymbol{X}\|_{\mathrm{op}}}{\sqrt{d_R}}$ | $\mathrm{Tr}[(d_R \sum\limits_{t=1}^{T}\boldsymbol{G}_t\boldsymbol{G}_t^\top)^{\frac{1}{2}}]$ | $\|\mathcal{X}\|_{\mathrm{op}} \, \mathrm{Tr}\left[\left(\sum\limits_{t=1}^{T}\boldsymbol{G}_t\boldsymbol{G}_t^\top\right)^{\frac{1}{2}}\right]$ (Theorem 4.2) |

*Table 1.* Regret bound obtained by Theorem 3.4 for different algorithms. Theorem 4.2 proves the rate specifically for one-sided shampoo and other rows match existing results in literature. $\mathcal{H}$ is chosen as $\mathcal{K} \cap \mathcal{S}_+^d$. In the third column, $\boldsymbol{x}$ denotes the parameter in vector form and $\boldsymbol{X}$ denotes the parameter in matrix form with $\boldsymbol{x} = \overline{\mathrm{vec}}(\boldsymbol{X})$. In the fourth column, $\boldsymbol{g}_t$ denotes the gradient in vector form and $\boldsymbol{G}_t$ denotes the gradient in matrix form with $\boldsymbol{g}_t = \overline{\mathrm{vec}}(\boldsymbol{G}_t)$. We omit $O(\cdot)$ in complexity measures and regret bound for convenience.

than a more memory- and computationally efficient variant of these methods like diagonal AdaGrad and AdaGrad-Norm (Streeter & McMahan, 2010; Duchi et al., 2011; Ward et al., 2020). This demonstrates that more structure on the preconditioner may not necessarily restrict its optimization performance. In fact, our unified analysis uncovers an interesting finding: we show a simpler, more structured variant of Shampoo, called one-sided Shampoo, actually has substantially better regret bound compared to original Shampoo and full-matrix Adagrad (Section 3.3).

### 1.1. Main contributions

In light of the above discussion, we highlight the main contributions of the paper.

- We present a comprehensive and unified theoretical framework (Theorem 2.1) for adaptive optimization with structured preconditioners, encompassing several popular methods including diagonal AdaGrad, full-matrix AdaGrad, AdaGrad-Norm, and layerwise adaptive methods for both online convex optimization and stochastic smooth convex optimization. In particular, our analysis integrates and extends key insights from existing work (Gupta et al., 2017), which presents a unified and elegant way (Algorithm 1) to derive the aforementioned structured preconditioners, but only allows analysis on a case-by-case basis. To our knowledge, this is the *first* truly unified analysis of a large family of adaptive optimization algorithms.

- To enable a unified analysis, we identify a novel sufficient condition, *well-structured preconditioners* (Definition 3.1), which overcomes a key technical barrier in Gupta et al. (2017); thereby, allowing a unified analysis for several important adaptive algorithms. A more detailed discussion is provided in Section 2 and a summary of our results is presented in Table 1.

- Our unified analysis provides a new regret bound (Theorem 4.2) for a one-sided variant of Shampoo, which is always better than the existing bound for two-sided Shampoo (Gupta et al., 2018) and could be smaller by a multiplicative factor of $d$, where we assume the matrix-shaped parameter is of size $\sqrt{d}$-by-$\sqrt{d}$. This also leads to a novel convergence rate of one-sided shampoo for stochastic convex optimization (Theorem 4.4) via a standard offline-to-online reduction (Levy et al., 2018).

- Conceptually, our findings challenge the conventional wisdom that using a larger set of preconditioners which require more memory and compute leads to better optimization performance in terms of number of steps. In particular, while using a larger set of preconditioners reduces the gradient term $\|\|\boldsymbol{g}_{1:t}\|\|_{\mathcal{H}}$ in our regret bound, it increases the other term involving the magnitude of the optimal solution by increasing the norm metric; thus, leading to a worse regret bound (Theorem 2.1 and Table 1). For instance, we demonstrate that one-sided Shampoo can outperform full-matrix AdaGrad both theoretically (Section 4.3) and experimentally (Section 5). This suggest that one-sided Shampoo is not just a computational-efficient surrogate of full-matrix AdaGrad, but could also be fundamentally better, depending on the optimization problem.

### 1.2. Notations

Let $\mathcal{M}^d$ be the set of all $d$-by-$d$ matrices, and $\mathcal{S}^d \subset \mathcal{M}^d$ be the subset of all symmetric matrices. We use $\mathcal{S}_+^d$ to denote the set of positive semi-definite matrices, and $\mathcal{S}_{++}^d$ to denote the set of positive definite matrices. $\mathcal{D}^d$ is the set of all $d$-dimensional diagonal matrices, and $\mathcal{D}_+^d = \mathcal{D}^d \cap \mathcal{S}_+^d$. We denote by $\boldsymbol{I}_d$ the $d$-by-$d$ identity matrix. For matrices $\boldsymbol{A}, \boldsymbol{B}$, we denote their inner product by $\langle \boldsymbol{A}, \boldsymbol{B} \rangle = \mathrm{Tr}(\boldsymbol{A}^\top \boldsymbol{B})$.

For any $\boldsymbol{H} \in \mathcal{S}_+^d$ such that $\boldsymbol{H} \neq 0$, we denote $\overline{\boldsymbol{H}} = \boldsymbol{H}/\mathrm{Tr}(\boldsymbol{H})$. For $\boldsymbol{H} \in \mathcal{S}_+^d$, $\|\boldsymbol{x}\|_{\boldsymbol{H}} := \sqrt{\boldsymbol{x}^\top \boldsymbol{H} \boldsymbol{x}}$ is the

(semi-)norm of $\boldsymbol{x} \in \mathbb{R}^d$ with respect to $\boldsymbol{H}$. For a convex set $\mathcal{H} \subseteq \mathcal{S}_+^d$, we define

$$\|\boldsymbol{x}\|_{\mathcal{H}} := \sup_{\boldsymbol{H} \in \mathcal{H}, \mathrm{Tr}(\boldsymbol{H}) \leq 1} \|\boldsymbol{x}\|_{\boldsymbol{H}}. \tag{1}$$

For a convex set $\mathcal{X} \subseteq \mathbb{R}^d$ and any norm $\|\cdot\|$, we define $\|\mathcal{X}\| := \sup_{\boldsymbol{x} \in \mathcal{X}} \|\boldsymbol{x}\|$ and $\|\mathcal{X}\|_{\mathcal{H}} := \sup_{\boldsymbol{x} \in \mathcal{X}} \|\boldsymbol{x}\|_{\mathcal{H}}$. For any $\boldsymbol{H} \succ 0$, the projection of $\boldsymbol{x}$ onto $\mathcal{X}$ with respect to $\|\cdot\|_{\boldsymbol{H}}$ is defined as $\Pi_{\mathcal{X}}^{\boldsymbol{H}}(\boldsymbol{x}) := \arg\min_{\boldsymbol{x}' \in \mathcal{X}} \|\boldsymbol{x} - \boldsymbol{x}'\|_{\boldsymbol{H}}$.

Throughout the paper, we consider the factorization $d = d_L d_R$, and denote the corresponding matrix form of $\boldsymbol{x} \in \mathbb{R}^d$ by $\boldsymbol{X} \in \mathbb{R}^{d_L \times d_R}$. We denote $\boldsymbol{x} = \overline{\mathrm{vec}}(\boldsymbol{X})$ and $\boldsymbol{X} = \overline{\mathrm{vec}}^{-1}(\boldsymbol{x})$ for conversion between the vector form and the matrix form. Then for a function $L(\boldsymbol{x})$ defined on $\boldsymbol{x} \in \mathbb{R}^d$, we extend its definition to matrices by letting $L(\boldsymbol{X})$ denote $L(\overline{\mathrm{vec}}(\boldsymbol{X}))$, and we will use $L(\boldsymbol{x})$ and $L(\boldsymbol{X})$ interchangeably when the context is clear. We define the gradient as $\boldsymbol{g}_t = \nabla L(\boldsymbol{x}_t)$ in vector form and $\boldsymbol{G}_t = \nabla L(\boldsymbol{X}_t)$ in matrix form, so $\boldsymbol{g}_t = \overline{\mathrm{vec}}(\boldsymbol{G}_t)$ and $\boldsymbol{G}_t = \overline{\mathrm{vec}}^{-1}(\boldsymbol{g}_t)$.

For a matrix $\boldsymbol{X} \in \mathbb{R}^{d_L \times d_R}$, we denote its operator norm as $\|\boldsymbol{X}\|_{\mathrm{op}}$ and its Frobenious norm as $\|\boldsymbol{X}\|_{\mathrm{F}}$. For a vector $\boldsymbol{x} \in \mathbb{R}^d$, we denotes its $\ell_\infty$ norm by $\|\boldsymbol{x}\|_\infty = \max_{i \in [d]} |x_i|$ and $\ell_2$ norm by $\|\boldsymbol{x}\|_2$.

## 2. Background: Unified Adaptive Regularization with Non-Unified Analysis

Seminal work Gupta et al. (2017) presented an adaptive regularization meta-algorithm, AdaReg (Algorithm 1), which can be used to derive various adaptive optimization algorithms known at that time in a unified approach. For example, AdaReg becomes full-matrix AdaGrad (Duchi et al., 2011), diagonal AdaGrad (Duchi et al., 2011), and AdaGrad-Norm (Streeter & McMahan, 2010; Ward et al., 2020) by choosing the set of preconditioners as the set of all PSD matrices, diagonal PSD matrices, and mutipliers of identity matrix respectively (see Table 1). The original AdaReg also allows other choices of potential function $\Phi$, e.g., $\Phi(\cdot) = \log\det(\cdot)$ for Online Newton Step (Hazan et al., 2007), while we are only interested in the case of $\Phi(\cdot) = \eta^2 \mathrm{Tr}(\cdot)$ in this work.

In addition to the unified approach to deriving various adaptive optimization algorithms, Gupta et al. (2017) also attempts to give a unified analysis for the convergence rate or regret of these adaptive algorithms, which can be summarized by the following theorem.

**Theorem 2.1** (Gupta et al. (2017)). *Let $\{\boldsymbol{x}_t\}_{t=1}^T$ be the iterates of Algorithm 1. Then for any $\boldsymbol{x}^* \in \mathcal{X}$,*

$$\sum_{t=1}^T L_t(\boldsymbol{x}_t) - \sum_{t=1}^T L_t(\boldsymbol{x}^*) \tag{2}$$
$$\leq \frac{1}{2}\left(\langle \boldsymbol{M}_T, \boldsymbol{H}_T^{-1}\rangle + \eta^2 \mathrm{Tr}(\boldsymbol{H}_T) - \eta^2 \mathrm{Tr}(\boldsymbol{H}_0)\right)$$
$$+ \frac{1}{2}\sum_{t=1}^T \left(\|\boldsymbol{x}_t - \boldsymbol{x}^*\|_{\boldsymbol{H}_t}^2 - \|\boldsymbol{x}_{t+1} - \boldsymbol{x}^*\|_{\boldsymbol{H}_t}^2\right).$$

---

**Algorithm 1** Adaptive Regularization Meta-Algorithm AdaReg (Gupta et al., 2017)

---

**Hyperparam:** $\epsilon > 0$, convex set $\mathcal{X} \subseteq \mathbb{R}^d$, learning rate $\eta$, preconditioners $\mathcal{H} \subset \mathcal{S}_+^d$
**Input:** initialization $\boldsymbol{x}_1$, loss functions $\{L_t\}_{t=1}^T : \mathbb{R}^d \to \mathbb{R}$
  $\boldsymbol{M}_0 \leftarrow \epsilon \boldsymbol{I}_d$
  **for** $t = 1, 2, \ldots, T$ **do**
    $\boldsymbol{g}_t \leftarrow \nabla L_t(\boldsymbol{x}_t)$
    $\boldsymbol{M}_t \leftarrow \boldsymbol{M}_{t-1} + \boldsymbol{g}_t \boldsymbol{g}_t^\top$
    $\boldsymbol{H}_t \leftarrow \arg\min_{\boldsymbol{H} \in \mathcal{H}} \langle \boldsymbol{M}_t, \boldsymbol{H}^{-1}\rangle + \eta^2 \mathrm{Tr}(\boldsymbol{H})$
    $\boldsymbol{x}_{t+1} \leftarrow \Pi_{\mathcal{X}}^{\boldsymbol{H}_t}\left(\boldsymbol{x}_t - \boldsymbol{H}_t^{-1}\boldsymbol{g}_t\right)$
  **Return** $\boldsymbol{x}_1, \ldots, \boldsymbol{x}_T$

---

The above bound is obtained by first applying a standard bound for online mirror descent to get a bound in the form of $\sum_{t=1}^T \|\boldsymbol{g}_t\|_{\boldsymbol{H}_t}$ plus the second term on the RHS of (2). Then the choice of $\boldsymbol{H}_t$ in Algorithm 1 enables the application of FTL-BTL lemma (Kalai & Vempala, 2005) to further bound $\sum_{t=1}^T \|\boldsymbol{g}_t\|_{\boldsymbol{H}_t}$ by the first term on the RHS of (2).

To proceed from Theorem 2.1, Gupta et al. (2017) relies on a crucial assumption that $\boldsymbol{H}_{t-1} \preceq \boldsymbol{H}_t$ for each $t$. Or more generally, for any $\boldsymbol{M} \succ 0$, define

$$P_{\mathcal{H}}(\boldsymbol{M}) := \arg\min_{\boldsymbol{H} \in \mathcal{H}} \langle \boldsymbol{M}, \boldsymbol{H}^{-1}\rangle + \eta^2 \mathrm{Tr}(\boldsymbol{H}). \tag{3}$$

Then we hope it holds that

$$P_{\mathcal{H}}(\boldsymbol{M}) \preceq P_{\mathcal{H}}(\boldsymbol{M}') \quad \text{for any} \quad 0 \prec \boldsymbol{M} \preceq \boldsymbol{M}' \tag{4}$$

In other words, we need $P_{\mathcal{H}} : \mathcal{S}_{++}^d \to \mathcal{S}_{++}^d$ to be *operator monotone* under the semi-definite ordering. With this assumption, a critical step in the derivation of the regret bound in Gupta et al. (2017) is to further rewrite and upper bound the second term on the RHS of Equation (2) by

$$\|\boldsymbol{x}_1 - \boldsymbol{x}^*\|_{\boldsymbol{H}_1}^2 + \sum_{t=2}^T \|\boldsymbol{x}_t - \boldsymbol{x}^*\|_{\boldsymbol{H}_t - \boldsymbol{H}_{t-1}}^2$$
$$\leq 4\|\mathcal{X}\|_{\boldsymbol{H}_1}^2 + 4\sum_{t=2}^T \|\mathcal{X}\|_{\boldsymbol{H}_t - \boldsymbol{H}_{t-1}}^2. \tag{5}$$

Note that we need $\boldsymbol{H}_{t-1} \preceq \boldsymbol{H}_t$ to ensure that $\|\cdot\|_{\boldsymbol{H}_t - \boldsymbol{H}_{t-1}}^2$ is indeed a (pseudo) norm and that the last inequality holds. Such analysis has been done for a few notable variants of AdaGrad in Gupta et al. (2017), where the condition $\boldsymbol{H}_{t-1} \preceq \boldsymbol{H}_t$ is verified in a case-by-case way for specific choice of $\mathcal{H}$. However, the following question remains unclear for optimizers described by general $\mathcal{H}$:

**Question 1.** *For a cone $\mathcal{H} \subseteq \mathcal{S}_+^d$, does $P_{\mathcal{H}}(\boldsymbol{M}) \preceq P_{\mathcal{H}}(\boldsymbol{M}')$ hold whenever $0 \prec \boldsymbol{M} \preceq \boldsymbol{M}'$?*

Indeed, the answer is *no* for "ill-structured" $\mathcal{H}$, and we mention two negative examples here for illustration.

**Example 2.2.** Let $\mathcal{H} = \{\boldsymbol{A} \otimes \boldsymbol{B} \succeq 0 \mid \boldsymbol{A} \in \mathcal{M}^{d_L}, \boldsymbol{B} \in \mathcal{M}^{d_R}\}$ with $d_L d_R = d$, i.e., the set of preconditioners for

two-sided Shampoo (Algorithm 3). We show that in the special case where $d_L = d_R = 2$, for $\boldsymbol{M} = \mathrm{diag}(1, \epsilon, \epsilon, \epsilon)$ and $\boldsymbol{M}' = \mathrm{diag}(1, \epsilon, \epsilon, 1)$, $P_{\mathcal{H}}(\boldsymbol{M}) \preceq P_{\mathcal{H}}(\boldsymbol{M}')$ does not hold for sufficiently small $\epsilon > 0$, although we have $\boldsymbol{M} \preceq \boldsymbol{M}'$. See Appendix A.4.1 for a detailed proof.

**Example 2.3.** The second example of ill-structured $\mathcal{H}$ involves tridiagonal PSD matrices, i.e., matrices that only have nonzero elements on the main diagonal, the first diagonal above the main diagonal, and the first diagonal below the main diagonal. Specifically, for $\mathcal{H}$ containing all 3-dimensional PSD matrices, we provide numerical evidence demonstrating that the desired condition Equation (4) breaks for very simple instances. See details in Appendix A.4.2.

These failure modes naturally lead to the second question:

**Question 2.** *Is there a sufficient yet general condition (which covers all existing examples) on $\mathcal{H}$ for the inequality to hold?*

As one of the main contributions of our work, we give an affirmative answer to Question 2 in Section 3 by proposing a notion of well-structured preconditioners (Definition 3.1) and deriving a unified analysis correspondingly.

# 3. Unified Analysis for Well-Structured Preconditioners

We establish a unified framework for adaptive optimization with structured preconditioners. In Section 3.1, we propose the notion of well-structured preconditioners and show that they satisfy the desired condition Equation (4). Then in Section 3.2, we present a unified analysis for adaptive optimization with well-structured preconditioners. We discuss several prominent examples in Section 3.3.

## 3.1. Well-structured preconditioners

For a set of $d$-by-$d$ matrices $\mathcal{K} \subseteq \mathcal{M}^d$, we say that $\mathcal{K}$ is a *subalgebra* if it is closed under scalar multiplication, matrix addition, and matrix multiplication. More concretely, we require that for any $\alpha \in \mathbb{R}$ and $\boldsymbol{A}, \boldsymbol{B} \in \mathcal{K}$, it holds that $\alpha \boldsymbol{A}, \boldsymbol{A}\boldsymbol{B}, \boldsymbol{A} + \boldsymbol{B} \in \mathcal{K}$. Based on this, we propose the following core concept of our paper.

**Definition 3.1** (Well-structured preconditioner sets). $\mathcal{H} \subseteq \mathcal{S}_+^d$ is said to be a *well-structured preconditioner set* if $\mathcal{H} = \mathcal{S}_+^d \cap \mathcal{K}$ for some matrix subalgebra $\mathcal{K} \subseteq \mathcal{M}^d$ with $\boldsymbol{I}_d \in \mathcal{K}$.

As a positive response to Question 2, the following proposition shows that our notion of well-structured preconditioner sets provides a sufficient condition for $P_{\mathcal{H}}(\cdot)$ to be operator monotone. See Appendix A for a proof.

**Proposition 3.2.** *Let $\mathcal{H}$ be a well-structured preconditioner set under Definition 3.1. For any $\boldsymbol{M} \succ 0$, there exists a unique solution $P_{\mathcal{H}}(\boldsymbol{M}) \succ 0$ to the optimization problem*

*in (3). Furthermore, for any $\boldsymbol{M} \succ 0$, $P_{\mathcal{H}}(\boldsymbol{M})$ satisfies the following properties:*

*(a)* $\langle \boldsymbol{M}, P_{\mathcal{H}}(\boldsymbol{M})^{-1} \rangle = \eta^2 \, \mathrm{Tr}(P_{\mathcal{H}}(\boldsymbol{M}))$.

*(b)* $\overline{P_{\mathcal{H}}(\boldsymbol{M})} = \arg\min_{\boldsymbol{H} \in \mathcal{H}, \mathrm{Tr}(\boldsymbol{H}) \leq 1} \langle \boldsymbol{M}, \boldsymbol{H}^{-1} \rangle$, *where we recall that* $\overline{P_{\mathcal{H}}(\boldsymbol{M})} = P_{\mathcal{H}}(\boldsymbol{M}) / \mathrm{Tr}(P_{\mathcal{H}}(\boldsymbol{M}))$.

*Moreover, for any $0 \prec \boldsymbol{M} \preceq \boldsymbol{M}'$, $P_{\mathcal{H}}(\boldsymbol{M}') - P_{\mathcal{H}}(\boldsymbol{M}) \in \mathcal{H}$ holds. In particular, it implies that $P_{\mathcal{H}}(\boldsymbol{M}) \preceq P_{\mathcal{H}}(\boldsymbol{M}')$.*

The closure under both matrix addition and matrix multiplication is crucial in the proof of Proposition 3.2. Violation of any of the two properties could lead to problems: for the preconditioner set of two-sided Shampoo in Example 2.2, it is not closed under matrix addition; while for Example 2.3 regarding tridiagonal matrices, we note that the set of tridiagonal matrices is not closed under matrix multiplication.

## 3.2. Unified analysis for adaptive optimization

To proceed with the regret bound, recall from Proposition 3.2 that $\overline{\boldsymbol{H}}_t = \boldsymbol{H}_t / \mathrm{Tr}(\boldsymbol{H}_t)$ is a solution to the following constrained optimization problem

$$\overline{\boldsymbol{H}}_t = \underset{\boldsymbol{H} \in \mathcal{H}, \mathrm{Tr}(\boldsymbol{H}) \leq 1}{\arg\min} \langle \boldsymbol{M}_t, \boldsymbol{H}^{-1} \rangle. \tag{6}$$

Also recall that $\boldsymbol{M}_t = \sum_{s=1}^{t} \boldsymbol{g}_s \boldsymbol{g}_s^\top$ (assuming $\epsilon = 0$ for illustration). We interpret the optimal value of this constrained optimization problem as the magnitude of the sequence of gradients $\boldsymbol{g}_{1:t} = (\boldsymbol{g}_1, \ldots, \boldsymbol{g}_t)$ with respect to the best preconditioner in hindsight. Motivated by this, for any sequence of gradients $\boldsymbol{g}_{1:t}$, we define their *adaptive gradient norm* with respect to $\mathcal{H}$ to be

$$\|\|\boldsymbol{g}_{1:t}\|\|_{\mathcal{H}} := \inf_{\boldsymbol{H} \in \mathcal{H}, \mathrm{Tr}(\boldsymbol{H}) \leq 1} \sqrt{\left\langle \sum_{s=1}^{t} \boldsymbol{g}_s \boldsymbol{g}_s^\top, \boldsymbol{H}^{-1} \right\rangle}. \tag{7}$$

Indeed, this definition of adaptive gradient norm corresponds to the dual norm of the norm $\| \cdot \|_{\mathcal{H} \otimes \boldsymbol{I}_t}$, denoted as $\| \cdot \|_{\mathcal{H} \otimes \boldsymbol{I}_t}^*$. Specifically, we define

$$\| \overline{\mathrm{vec}}(\boldsymbol{g}_{1:t}) \|_{\mathcal{H} \otimes \boldsymbol{I}_t}^* = \sup_{\boldsymbol{w} \in \mathbb{R}^{td} : \|\boldsymbol{w}\|_{\mathcal{H} \otimes \boldsymbol{I}_t} \leq 1} \overline{\mathrm{vec}}(\boldsymbol{g}_{1:t})^\top \boldsymbol{w}.$$

Then we have $\|\|\boldsymbol{g}_{1:t}\|\|_{\mathcal{H}} = \| \overline{\mathrm{vec}}(\boldsymbol{g}_{1:t}) \|_{\mathcal{H} \otimes \boldsymbol{I}_t}^* / \sqrt{t}$ by the following lemma, which is proved in Appendix A.3.

**Lemma 3.3.** *Let $\mathcal{H}$ be a well-structured preconditioner set under Definition 3.1. Then for any $t \geq 1$ and $\boldsymbol{g}_1, \ldots, \boldsymbol{g}_t \in \mathbb{R}^d$, it holds that*

$$\inf_{\boldsymbol{H} \in \mathcal{H}, \mathrm{Tr}(\boldsymbol{H}) \leq 1} \sqrt{\left\langle \sum_{s=1}^{t} \boldsymbol{g}_s \boldsymbol{g}_s^\top, \boldsymbol{H}^{-1} \right\rangle} = \frac{1}{\sqrt{t}} \| \overline{\mathrm{vec}}(\boldsymbol{g}_{1:t}) \|_{\mathcal{H} \otimes \boldsymbol{I}_t}^*.$$

Given this definition of adaptive gradient norm, combining (6) and the fact that $\mathrm{Tr}(\boldsymbol{H}_t) = \eta^{-2} \langle \boldsymbol{M}_t, \boldsymbol{H}_t^{-1} \rangle$ by Proposition 3.2, we obtain $\mathrm{Tr}(\boldsymbol{H}_t) = \eta^{-1} \|\|\boldsymbol{g}_{1:t}\|\|_{\mathcal{H}}$.

Now recall the upper bound in (5) for the second term in the regret bound (2) from Theorem 2.1. Note that for each $t$, Proposition 3.2 guarantees that $\boldsymbol{H}_t - \boldsymbol{H}_{t-1} \in \mathcal{H}$, so we can further bound $\|\mathcal{X}\|^2_{\boldsymbol{H}_t - \boldsymbol{H}_{t-1}} \leq \|\mathcal{X}\|^2_{\mathcal{H}} \cdot \text{Tr}(\boldsymbol{H}_t - \boldsymbol{H}_{t-1})$. This allows us to telescope the sum in Equation (5) to get

$$\|\mathcal{X}\|^2_{\boldsymbol{H}_1} + \sum_{t=2}^{T} \|\mathcal{X}\|^2_{\boldsymbol{H}_t - \boldsymbol{H}_{t-1}} = \frac{\|\mathcal{X}\|^2_{\mathcal{H}}}{\eta} \|\!|\boldsymbol{g}_{1:T}|\!\|_{\mathcal{H}} \quad (8)$$

Observe that the upper bound is factored into two parts: 1) $\|\mathcal{X}\|_{\mathcal{H}}$, the diameter of the domain under norm $\|\cdot\|_{\mathcal{H}}$; and 2) $\|\!|\boldsymbol{g}_{1:T}|\!\|_{\mathcal{H}}$, the adaptive gradient norm with respect to $\mathcal{H}$. We pause here for an important remark. Note that $\|\mathcal{X}\|^2_{\boldsymbol{H}_t - \boldsymbol{H}_{t-1}}$ is defined by taking the supremum over $\boldsymbol{x} \in \mathcal{X}$, which precludes telescoping the sum over $t \in [T]$ at first glance. We address this issue by proposing the norm $\|\mathcal{X}\|_{\mathcal{H}}$, which allows us to extract the factor $\text{Tr}(\boldsymbol{H}_t - \boldsymbol{H}_{t-1})$. Again, this unified analysis is possible thanks to Proposition 3.2 for well-structured preconditioner sets, in contrast to the case-by-case analysis done by Gupta et al. (2017). Furthermore, we remark that the above factored bound is crucial for us to identify the correct norm metric for Shampoo, leading to an improved analysis. See Section 4.1 for details.

Finally, combining (8) and the original regret bound in (2) yields the final regret bound for Algorithm 1. This is summarized in the following Theorem 3.4. The complete proof can be found in Appendix D.

**Theorem 3.4.** *Let $\mathcal{H}$ be a well-structured preconditioner set under Definition 3.1. Then for any convex loss functions $L_1, \ldots, L_T$, the regret of Algorithm 1 compared to any $\boldsymbol{x}^* \in \mathcal{X}$ can be bounded as*

$$\sum_{t=1}^{T} L_t(\boldsymbol{x}_t) - \sum_{t=1}^{T} L_t(\boldsymbol{x}^*) \leq \left( \frac{D^2}{2\eta} + \eta \right) (G + d\sqrt{\epsilon})$$

*where $G = \|\!|\boldsymbol{g}_{1:T}|\!\|_{\mathcal{H}}$, $D = \max_{t \in [T]} \|\boldsymbol{x}_t - \boldsymbol{x}^*\|_{\mathcal{H}}$.*

**Corollary 3.5.** *Under the setting of Theorem 2.1, further suppose that $\mathcal{X}$ is a bounded set in $\mathbb{R}^d$. Then choosing $\eta = \sqrt{2} \|\mathcal{X}\|_{\mathcal{H}}$, the regret bound for Algorithm 1 becomes*

$$\sum_{t=1}^{T} L_t(\boldsymbol{x}_t) - \sum_{t=1}^{T} L_t(\boldsymbol{x}^*) \leq 2\sqrt{2} \|\mathcal{X}\|_{\mathcal{H}} (G + d\sqrt{\epsilon}).$$

Ignoring $\epsilon$, the bound reveals an intrinsic trade-off between $\|\mathcal{X}\|_{\mathcal{H}}$ and $\|\!|\boldsymbol{g}_{1:t}|\!\|_{\mathcal{H}}$: as $\mathcal{H}$ gets larger, $\|\mathcal{X}\|_{\mathcal{H}}$ increases while $\|\!|\boldsymbol{g}_{1:t}|\!\|_{\mathcal{H}}$ decreases. The previous common belief that more adaptivity (larger $\mathcal{H}$) helps optimization could be largely due to the loose upper bound on $\|\mathcal{X}\|_{\mathcal{H}}$, i.e., always measuring the size of domain $\mathcal{X}$ by Frobenius norm instead of potentially much smaller $\|\cdot\|_{\mathcal{H}}$. The fact that Kroneckered-factored subalgebra induces smaller $\|\cdot\|_{\mathcal{H}}$ than the entire matrix subalgebra is the key reason behind our surprising finding that one-sided Shampoo can outperform

full-matrix AdaGrad in terms of the number of steps. See a more detailed analysis in Section 4.3.

Next, we present the convergence rate of Algorithm 1 for stochastic convex smooth loss functions. We use Definition 3.6 to characterize the smoothness of a loss function. It is an extension of $\Phi$-smoothness in (Xie et al., 2025), which only applies to block-diagonal preconditioners (corresponding to blockwise Adam).

**Definition 3.6** ($\mathcal{H}$-smoothness). For a set $\mathcal{H} \subseteq \mathcal{S}^d_+$ and any loss function $L : \mathbb{R}^d \to \mathbb{R}$, we define the $\mathcal{H}$-smoothness of $L$, denoted by $H(L, \mathcal{H})$, as the smallest number $H \geq 0$ such that there exists a matrix $\boldsymbol{H}^* \in \mathcal{H}$ satisfying $H = \text{Tr}(\boldsymbol{H}^*)$ and for any $\boldsymbol{x} \in \mathbb{R}^d$, it holds that $-\boldsymbol{H}^* \preceq \nabla^2 L(\boldsymbol{x}) \preceq \boldsymbol{H}^*$. In the case of convex $L$, this requirement becomes $\nabla^2 L(\boldsymbol{x}) \preceq \boldsymbol{H}^*$. Furthermore, we extend the notation to matrices $\boldsymbol{A} \in \mathcal{M}^d$ by defining $H(\boldsymbol{A}, \mathcal{H})$ as $H(\boldsymbol{x} \mapsto \frac{1}{2}\boldsymbol{x}^\top \boldsymbol{A}\boldsymbol{x}, \mathcal{H})$.

We need the following assumption on the stochastic noise.

**Assumption 3.7.** For any $t \in [T]$ and any $\boldsymbol{x} \in \mathcal{X}$, $\mathbb{E}[L_t(\boldsymbol{x})] = L(\boldsymbol{x})$ and there exists some $\boldsymbol{\Sigma} \in \mathcal{S}^d_+$ such that $\mathbb{E}[(\nabla L_t(\boldsymbol{x}) - \nabla L(\boldsymbol{x})) (\nabla L_t(\boldsymbol{x}) - \nabla L(\boldsymbol{x}))^\top] \preceq \boldsymbol{\Sigma}$.

Now we are ready to state our main results on the convergence rate of Algorithm 1 for stochastic convex functions.

**Theorem 3.8.** *Let $\mathcal{H}$ be a well-structured preconditioner set under Definition 3.1. Consider any independent stochastic convex loss functions $L_1, \ldots, L_T$ satisfying Assumption 3.7, and let $H(L, \mathcal{H})$ be the $\mathcal{H}$-smoothness of their expectation $L$. Suppose the global minimizer of $L$, denoted by $\boldsymbol{x}^*$, is in $\mathcal{X}$. Then for the iterates $\boldsymbol{x}_1, \ldots, \boldsymbol{x}_T$ of Algorithm 1, denoting $\bar{\boldsymbol{x}}_{1:T} = \frac{1}{T} \sum_{t=1}^{T} \boldsymbol{x}_t$, it holds that*

$$\mathbb{E}[L(\bar{\boldsymbol{x}}_{1:T}) - L(\boldsymbol{x}^*)]$$
$$\leq \frac{16}{T} \|\mathcal{X}\|^2_{\mathcal{H}} H(L, \mathcal{H}) + \frac{4\sqrt{2}}{\sqrt{T}} \|\mathcal{X}\|_{\mathcal{H}} \sigma + \frac{4\sqrt{2}d\sqrt{\epsilon}}{T} \|\mathcal{X}\|_{\mathcal{H}}$$

*where $\sigma = \inf_{\boldsymbol{H} \in \mathcal{H}, \text{Tr}(\boldsymbol{H}) \leq 1} \sqrt{\langle \boldsymbol{\Sigma}, \boldsymbol{H}^{-1} \rangle}$.*

Our analysis naturally extends to Algorithm 4, which replaces the direct sum of past gradient outer products in Algorithm 1 with an exponential moving average (EMA). This modification is widely used in adaptive optimizers, including Adam (Kingma & Ba, 2014) and AdaSGD (Wang & Wiens, 2020), which is an ema version of AdaGrad. The detailed discussion is in Appendix C.

### 3.3. Examples of well-structured preconditioner sets

Next, we demonstrate that our Definition 3.1 is general enough to cover existing examples, by discussing several important matrix subalgebra $\mathcal{K}$. For each associated well-structured $\mathcal{H} = \mathcal{K} \cap \mathcal{S}^d_+$, recall the optimization problem

over $\mathcal{H}$ defined in (3). We will show that every minimizer $P_{\mathcal{H}}(\boldsymbol{M})$ corresponds to the preconditioner used in a specific adaptive optimization algorithm. We list the correspondence relationship below and the results are summarized in Table 1. The detailed derivation and calculation of the norms $\|\boldsymbol{x}\|_{\mathcal{H}}$ and $\|\|\boldsymbol{g}_{1:t}\|\|_{\mathcal{H}}$ can be found in Appendix B.

**Example 3.9** (AdaGrad-Norm: scalar matrices). For the scalar matrix subalgebra $\mathcal{K} = \{c \cdot \boldsymbol{I}_d \mid c \in \mathbb{R}\}$, we have $\mathcal{H} = \{c \cdot \boldsymbol{I}_d \mid c \geq 0\}$. Then solving (3) for $\boldsymbol{M}_t$ yields

$$\boldsymbol{H}_t = \frac{1}{\eta}\sqrt{\mathrm{Tr}(\boldsymbol{M}_t)/d} \cdot \boldsymbol{I}_d = \frac{1}{\eta}\sqrt{\epsilon + \sum_{s=1}^{t}\|\boldsymbol{g}_s\|_2^2/d} \cdot \boldsymbol{I}_d.$$

This is the preconditioner used in AdaGrad-Norm.

**Example 3.10** (Diagonal AdaGrad: diagonal matrices). For the diagonal matrix subalgebra $\mathcal{K} = \mathcal{D}^d$, we have $\mathcal{H} = \mathcal{D}_+^d$. Correspondingly,

$$\boldsymbol{H}_t = \frac{1}{\eta}\mathrm{diag}\left(\sqrt{\epsilon + \sum_{s=1}^{t}\|g_{s,i}\|_2^2} : i \in [d]\right).$$

This is the preconditioner used in diagonal AdaGrad.

**Example 3.11** (Full-matrix AdaGrad: all matrices). For $\mathcal{K} = \mathcal{M}^d$, we have $\mathcal{H} = \mathcal{S}_+^d$. In this case, solving (3) for $\boldsymbol{M}_t$ yields that $\boldsymbol{H}_t = \frac{1}{\eta}\boldsymbol{M}_t^{\frac{1}{2}}$, which corresponds to the update rule of full-matrix AdaGrad.

**Example 3.12** (One-sided Shampoo: factored matrices). Let $d$ be factored as $d = d_L d_R$. Then for the factored matrix algebra $\mathcal{K} = \mathbb{R}^{d_L \times d_L} \otimes \boldsymbol{I}_{d_R}$, we have $\mathcal{H} = \mathcal{S}_+^{d_L} \otimes \boldsymbol{I}_{d_R}$. Now writing $\boldsymbol{G}_t \in \mathbb{R}^{d_L \times d_R}$ as the matricized version of $\boldsymbol{g}_t \in \mathbb{R}^d$, solving the corresponding problem in (3) leads to

$$\boldsymbol{H}_t = \frac{1}{\eta}\left(\epsilon \cdot \boldsymbol{I}_{d_L} + \frac{1}{d_R}\sum_{s=1}^{t}\boldsymbol{G}_s\boldsymbol{G}_s^{\top}\right)^{\frac{1}{2}} \otimes \boldsymbol{I}_{d_R},$$

which updates $\boldsymbol{x}_t$ the same as one-sided Shampoo[1] displayed in Algorithm 2, where we write the algorithm in the matrix form for convenience. More specifically, note that $\boldsymbol{g}_t = \overline{\mathrm{vec}}(\boldsymbol{G}_t)$ and $\boldsymbol{H}_t = \boldsymbol{L}_t^{\frac{1}{2}} \otimes \boldsymbol{I}_{d_R}$. Moreover, $\boldsymbol{M}_t$ corresponds to $\boldsymbol{L}_t$ by the fact that $\langle \boldsymbol{M}_t, (\boldsymbol{H}_L \otimes \boldsymbol{I}_{d_R})^{-1}\rangle = \langle \boldsymbol{L}_t, \boldsymbol{H}_L^{-1}\rangle$ for any $\boldsymbol{H}_L \in \mathcal{S}_{++}^{d_L}$.

The detailed calculations of the norms $\|\cdot\|_{\mathcal{H}}$ and $\|\|\cdot\|\|_{\mathcal{H}}$ for the above examples can be found in Appendix B.

**Generate new well-structured preconditioner sets.** Beyond the previous examples, it is also possible to generate new well-structured preconditioner sets based on existing ones. We discuss in particular an example of layerwise combination for parameters in a neural network. Specifically,

for $d$ parameters of an $N$-layer neural network, decompose $\mathbb{R}^d = \mathbb{R}^{d_1} \times \cdots \times \mathbb{R}^{d_N}$ where $d = \sum_{n=1}^{N} d_n$, and each $\mathbb{R}^{d_n}$ corresponds to the $d_n$ parameters in the $n$-th layer. For each $n \in [N]$, let $\mathcal{K}_n \subseteq \mathcal{M}^{d_n}$ be a matrix subalgebra. Then we define $\mathcal{K} = \oplus_{n=1}^{N}\mathcal{K}_n = \{\oplus_{n=1}^{N}\boldsymbol{A}_n \mid \boldsymbol{A}_n \in \mathcal{K}_n, n \in [N]\}$, and it is easy to verify that $\mathcal{K}$ is also a subalgebra. Then the corresponding well-structured preconditioner set $\mathcal{H} = \oplus_{n=1}^{N}(\mathcal{K}_n \cap \mathcal{S}_+^{d_n}) = \oplus_{n=1}^{N}\mathcal{H}_n$ contains preconditioners that apply individual types of transforms to gradient of parameters in different layers[2]. Such $\mathcal{H}$ made up by direct sum of smaller cones also has very compositional property in its induced complexity metrics, namely, $\|\cdot\|_{\mathcal{H}} = \max_{1 \leq n \leq N}\|\cdot\|_{\mathcal{H}_n}$ and $\|\|\cdot\|\|_{\mathcal{H}} = \sum_{n=1}^{N}\|\|\cdot\|\|_{\mathcal{H}_n}$. This operation provides a useful tool for designing layerwise preconditioning methods (Bernstein & Newhouse, 2024a).

Other possible operations that can generate new matrix subalgebra $\mathcal{K}'$ from the original subalgebra $\mathcal{K}$ include taking Kronecker product with the identity matrix, i.e. $\mathcal{K}' = \{\boldsymbol{A}' = \boldsymbol{A} \otimes \boldsymbol{I}_{d'} \mid \boldsymbol{A} \in \mathcal{K}\}$, and rotation by an orthogonal matrix, i.e. $\mathcal{K}' = \{\boldsymbol{A}' = \boldsymbol{U}^{\top}\boldsymbol{A}\boldsymbol{U} \mid \boldsymbol{A} \in \mathcal{K}\}$ where $\boldsymbol{U}$ is an orthogonal matrix.

# 4. Improved Convergence Analysis for One-sided Shampoo

We now turn to one-sided Shampoo (Algorithm 2), a special example of Algorithm 1. In Section 4.1, we present our main results on its regret bound and convergence rate, and then compare to the previous results for two-sided Shampoo in Section 4.2. In Section 4.3, we present a comprehensive comparison between the regret bound of one-sided Shampoo and those of the AdaGrad variants, which suggests why Shampoo can outperform other adaptive algorithms on some real tasks.

## 4.1. Our results for one-sided Shampoo

We first characterize the norm $\|\cdot\|_{\mathcal{H}}$ for one-sided Shampoo.

**Lemma 4.1** ($\|\cdot\|_{\mathcal{H}}$ for one-sided Shampoo). *Recall from Example 3.12 that for one-sided Shampoo (Algorithm 2), $\mathcal{H} = \mathcal{S}_+^{d_L} \otimes \boldsymbol{I}_{d_R}$ where $d_L d_R = d$. Then for any $\boldsymbol{x} \in \mathbb{R}^d$, it holds that $\|\boldsymbol{x}\|_{\mathcal{H}} = \frac{1}{\sqrt{d_R}}\|\boldsymbol{X}\|_{\mathrm{op}}$ with $\boldsymbol{X} = \overline{\mathrm{vec}}^{-1}(\boldsymbol{x})$, and thus $\|\mathcal{X}\|_{\mathcal{H}} = \frac{1}{\sqrt{d_R}}\|\mathcal{X}\|_{\mathrm{op}}$.*

See Appendix B.4 for the proof.

With this, we can apply Theorem 3.4 to get the regret bound for one-sided Shampoo, as summarized below in Theorem 4.2. See Appendix E.2 for its proof.

**Theorem 4.2** (Regret bound for one-sided Shampoo). *For convex functions $L_1, \ldots, L_T$, the regret of one-sided Sham-*

---

[1] We add a normalized factor $\frac{1}{d_R}$ compared to the $\boldsymbol{L}_t$ in Algorithm 3 so that it can be exactly derived from Algorithm 1.

[2] This can be applied to any partition of the parameters, not only for partition based on layers.

---

**Algorithm 2** One-sided Shampoo

> **Hyperparam:** learning rate $\eta > 0$, convex set $\mathcal{X} \subseteq \mathbb{R}^{d_L \times d_R}$, $\boldsymbol{L}_0 = \epsilon \boldsymbol{I}_{d_L}$ for $\epsilon \geq 0$
> **Input:** initialization $\boldsymbol{x}_0$, stochastic loss functions $\{L_t\}_{t=1}^T : \mathbb{R}^{d_L \times d_R} \to \mathbb{R}$
> **for** $t = 1, 2, \cdots, T$ **do**
> $\quad \boldsymbol{G}_t \leftarrow \nabla L_t(\boldsymbol{X}_{t-1})$
> $\quad \boldsymbol{L}_t \leftarrow \boldsymbol{L}_{t-1} + \frac{1}{d_R} \boldsymbol{G}_t \boldsymbol{G}_t^\top$
> $\quad \boldsymbol{X}_t \leftarrow \Pi_{\mathcal{X}}^{\boldsymbol{L}_t^{\frac{1}{2}} \otimes \boldsymbol{I}_{d_R}} \left( \boldsymbol{X}_{t-1} - \eta_t \boldsymbol{L}_t^{-\frac{1}{2}} \boldsymbol{G}_t \right)$
> **Return** $\boldsymbol{x}_T$

---

**Algorithm 3** Two-sided Shampoo (Gupta et al., 2018)

> **Hyperparam:** learning rate $\eta$, convex set $\mathcal{X} \subseteq \mathbb{R}^{d_L \times d_R}$, $\boldsymbol{L}_0 = \epsilon \boldsymbol{I}_{d_L}$, $\boldsymbol{R}_0 = \epsilon \boldsymbol{I}_{d_R}$ for $\epsilon \geq 0$
> **Input:** initialization $\boldsymbol{x}_0$, stochastic loss functions $\{L_t\}_{t=1}^T : \mathbb{R}^{d_L \times d_R} \to \mathbb{R}$
> **for** $t = 1, 2, \cdots, T$ **do**
> $\quad \boldsymbol{G}_t \leftarrow \nabla L_t(\boldsymbol{X}_{t-1})$
> $\quad \boldsymbol{L}_t \leftarrow \boldsymbol{L}_{t-1} + \boldsymbol{G}_t \boldsymbol{G}_t^\top$
> $\quad \boldsymbol{R}_t \leftarrow \boldsymbol{R}_{t-1} + \boldsymbol{G}_t^\top \boldsymbol{G}_t$
> $\quad \boldsymbol{X}_t \leftarrow \Pi_{\mathcal{X}}^{\boldsymbol{L}_t^{\frac{1}{4}} \otimes \boldsymbol{R}_t^{\frac{1}{4}}} \left( \boldsymbol{X}_{t-1} - \eta_t \boldsymbol{L}_t^{-\frac{1}{4}} \boldsymbol{G}_t \boldsymbol{R}_t^{-\frac{1}{4}} \right)$
> **Return** $\boldsymbol{x}_T$

---

poo (Algorithm 2) compared to any $\boldsymbol{X}^* \in \mathbb{R}^{d_L \times d_R}$ satisfies

$$\sum_{t=1}^T L_t(\boldsymbol{X}_t) - \sum_{t=1}^T L_t(\boldsymbol{X}^*) \leq \left( \frac{D_{\text{op}}^2}{2 d_R \eta} + \eta \right) \left( G + d\sqrt{\epsilon} \right),$$

where $D_{\text{op}} = \max_{t \in [T]} \|\boldsymbol{X}_t - \boldsymbol{X}^*\|_{\text{op}}$ and $G = \sqrt{d_R} \operatorname{Tr} \left[ \left( \sum_{t=1}^T \boldsymbol{G}_t \boldsymbol{G}_t^\top \right)^{\frac{1}{2}} \right]$. When the domain $\mathcal{X}$ is bounded in operator norm, i.e., $\|\mathcal{X}\|_{\text{op}} < \infty$, further choosing $\eta = \sqrt{2/d_R} \|\mathcal{X}\|_{\text{op}}$, it holds

$$\sum_{t=1}^T L_t(\boldsymbol{X}_t) - \sum_{t=1}^T L_t(\boldsymbol{X}^*)$$
$$\leq 2\sqrt{2} \|\mathcal{X}\|_{\text{op}} \left( \operatorname{Tr} \left[ \left( \sum_{t=1}^T \boldsymbol{G}_t \boldsymbol{G}_t^\top \right)^{\frac{1}{2}} \right] + \frac{d}{\sqrt{d_R}} \sqrt{\epsilon} \right).$$

Next, before presenting the convergence rate of one-sided Shampoo, we first provide a more interpretable formulation of the $\mathcal{H}$-smoothness for one-sided Shampoo, which we call *left smoothness*. See Appendix E.1 for a proof of Lemma 4.3.

**Lemma 4.3** (Left smoothness for one-sided Shampoo). *Let $\mathcal{H} = \mathcal{S}_+^{d_L} \otimes \boldsymbol{I}_{d_R}$ be the well-structured preconditioner set for one-sided Shampoo. Then the $\mathcal{H}$-smoothness $H(L, \mathcal{H})$ defined in Definition 3.6 is equal to the smallest number $H \geq 0$ such that there exists $\boldsymbol{H}_{d_L}^* \in \mathbb{R}^{d_L \times d_L}$ satisfying that $H = d_R \operatorname{Tr}(\boldsymbol{H}_{d_L}^*)$ and that for any $\boldsymbol{X}, \boldsymbol{\Delta} \in \mathbb{R}^{d_L \times d_R}$,*

$$\left| \nabla^2 L(\boldsymbol{X})[\boldsymbol{\Delta}, \boldsymbol{\Delta}] \right| \leq \left\langle \boldsymbol{H}_{d_L}^*, \boldsymbol{\Delta} \boldsymbol{\Delta}^\top \right\rangle.$$

*In this case, the $\mathcal{H}$-smoothness is denoted by $H_{\text{left}}(L)$.*

Then the convergence rate of one-sided Shampoo can be obtained by specializing Theorem 3.8 to $\|\mathcal{X}\|_{\mathcal{H}} = \frac{1}{\sqrt{d_R}} \|\mathcal{X}\|_{\text{op}}$ from Lemma 4.1 and the left smoothness from Lemma 4.3.

**Theorem 4.4** (Convergence rate of one-sided Shampoo). *Let $L_1, \ldots, L_T$ be stochastic convex loss functions satisfying Assumption 3.7, and let $\boldsymbol{x}_1, \ldots, \boldsymbol{x}_T$ be the corresponding iterates of one-sided Shampoo (Algorithm 2) with learn-*

ing rate $\eta = \sqrt{2} \|\mathcal{X}\|_{\text{op}}$. Then for $\bar{\boldsymbol{x}}_{1:T} = \frac{1}{T} \sum_{t=1}^T \boldsymbol{x}_t$,

$$\mathbb{E}[L(\bar{\boldsymbol{x}}_{1:T}) - L(\boldsymbol{x}^*)]$$
$$\leq \frac{16}{T d_R} \|\mathcal{X}\|_{\text{op}}^2 H_{\text{left}}(L) + \frac{4\sqrt{2}\sigma}{\sqrt{T d_R}} \|\mathcal{X}\|_{\text{op}} + \frac{4\sqrt{2} d \sqrt{\epsilon}}{T} \|\mathcal{X}\|_{\text{op}}$$

where $\sigma = \inf_{\boldsymbol{H} \in \mathcal{H}, \operatorname{Tr}(\boldsymbol{H}) \leq 1} \sqrt{\langle \boldsymbol{\Sigma}, \boldsymbol{H}^{-1} \rangle}$, and $H_{\text{left}}(L)$ is the left smoothness of the expected loss $L$ identified in Lemma 4.3.

### 4.2. Comparison with previous results on Shampoo

We compare our main results for one-sided Shampoo with the original results in Gupta et al. (2018). Here, we restate their original regret bound for easier comparison.

**Theorem 4.5** (Regret bound of two-sided Shampoo (Gupta et al., 2018)). *For convex functions $\{L_t\}_{t=1}^T$, suppose their gradients $(\boldsymbol{G}_t = \nabla L_t(\boldsymbol{X}_t))_{t=1}^T$ are matrices of rank at most $r$. Then the regret of two-sided Shampoo (Algorithm 3[3]) compared to any $\boldsymbol{X}^* \in \mathbb{R}^{d_L \times d_R}$ is bounded as*

$$\sum_{t=1}^T L_t(\boldsymbol{X}_t) - \sum_{t=1}^T L_t(\boldsymbol{X}^*) \leq \left( \frac{D_{\text{F}}^2}{2\eta} + r\eta \right) \operatorname{Tr}(\boldsymbol{L}_T^{\frac{1}{4}}) \operatorname{Tr}(\boldsymbol{R}_T^{\frac{1}{4}}),$$

where $D_{\text{F}} = \max_{t \in [T]} \|\boldsymbol{X}_t - \boldsymbol{X}^*\|_{\text{F}}$, $\boldsymbol{L}_T = \epsilon \boldsymbol{I}_{d_L} + \sum_{t=1}^T \boldsymbol{G}_t \boldsymbol{G}_t^\top$, and $\boldsymbol{R}_T = \epsilon \boldsymbol{I}_{d_R} + \sum_{t=1}^T \boldsymbol{G}_t^\top \boldsymbol{G}_t$. When $\|\mathcal{X}\|_{\text{F}} < \infty$, we further choose $\eta = \sqrt{2/r} \|\mathcal{X}\|_{\text{F}}$, then

$$\sum_{t=1}^T L_t(\boldsymbol{X}_t) - \sum_{t=1}^T L_t(\boldsymbol{X}^*) \leq \sqrt{2r} \|\mathcal{X}\|_{\text{F}} \operatorname{Tr}(\boldsymbol{L}_T^{\frac{1}{4}}) \operatorname{Tr}(\boldsymbol{R}_T^{\frac{1}{4}}).$$

We now compare our regret bound in Theorem 4.2 and the original regret bound in Theorem 4.5 by Gupta et al. (2018) when $\epsilon = 0$, i.e., $\boldsymbol{L}_T = \sum_{t=1}^T \boldsymbol{G}_t \boldsymbol{G}_t^\top$ and $\boldsymbol{R}_T = \sum_{t=1}^T \boldsymbol{G}_t^\top \boldsymbol{G}_t$. For a matrix $\boldsymbol{M} \in \mathcal{S}_+^d$, it always holds that $\sqrt{\operatorname{Tr}(\boldsymbol{M})} \leq \operatorname{Tr}(\boldsymbol{M}^{\frac{1}{2}})$. Therefore,

$$\operatorname{Tr}(\boldsymbol{L}_T^{\frac{1}{4}}) \operatorname{Tr}(\boldsymbol{R}_T^{\frac{1}{4}}) \geq \operatorname{Tr}(\boldsymbol{L}_T^{\frac{1}{4}}) \operatorname{Tr}(\boldsymbol{R}_T)^{\frac{1}{4}} = \operatorname{Tr}(\boldsymbol{L}_T^{\frac{1}{4}}) \operatorname{Tr}(\boldsymbol{L}_T)^{\frac{1}{4}}$$
$$\geq \operatorname{Tr}(\boldsymbol{L}_T^{\frac{1}{4}}) \|\boldsymbol{L}_T\|_{\text{op}}^{\frac{1}{4}} \geq \operatorname{Tr}(\boldsymbol{L}_T^{\frac{1}{2}}).$$

---

[3]The original two-sided Shampoo analysis in (Gupta et al., 2017) is without per step projection to the bounded domain. We adapt their theorem into the projected version in a standard way.

This implies that regret bound in Theorem 4.5 is always no smaller than the regret bound in Theorem 4.2 because $r \geq 1$ and $\|\mathcal{X}\|_{\mathrm{F}} \geq \|\mathcal{X}\|_{\mathrm{op}}$ as $\|\boldsymbol{X}\|_{\mathrm{F}} \geq \|\boldsymbol{X}\|_{\mathrm{op}}$ for any $\boldsymbol{X} \in \mathcal{X}$.

Moreover, in the worst case, the Frobenius norm can be $\sqrt{\min(d_L, d_R)}$ times larger than the operator norm. As a concrete example, suppose $\boldsymbol{G}_t$ satisfies that $\boldsymbol{G}_t[i, j] = 1$ only for $(j - i) \equiv t \pmod{\min(d_L, d_R)}$ and all other elements are zero. Then each $\boldsymbol{G}_t$ has rank $r = \min(d_L, d_R)$. At step $T = d_L d_R$, $\boldsymbol{L}_T = d_R \cdot \min(d_L, d_R) \cdot \boldsymbol{I}_{d_L}$ and $\boldsymbol{R}_T = d_L \cdot \min(d_L, d_R) \cdot \boldsymbol{I}_{d_R}$, and thus $\mathrm{Tr}(\boldsymbol{L}_T^{\frac{1}{4}}) \mathrm{Tr}(\boldsymbol{R}_T^{\frac{1}{4}}) = d_L^{\frac{5}{4}} d_R^{\frac{5}{4}}$ while $\mathrm{Tr}(\boldsymbol{L}_T^{\frac{1}{2}}) = d_L d_R^{\frac{1}{2}}$. In this case, the regret bound of two-sided Shampoo is $\min(d_L, d_R) d_L^{\frac{1}{4}} d_R^{\frac{3}{4}}$ times larger than the our regret bound for one-sided Shampoo.

Duvvuri et al. (2024) introduced CASPR as an alternative to Shampoo for approximating full-matrix AdaGrad and achieved the same regret bound as Theorem 4.5. Our previous comparison thus applies to both Shampoo and CASPR.

### 4.3. Comparison with AdaGrad variants

Next we show one-sided Shampoo can achieve the best theoretical upper bound of the suboptimality gap for a specific class of functions, where each loss function has the form

$$L(\boldsymbol{X}) = \langle \boldsymbol{H}, (\boldsymbol{X} - \boldsymbol{X}^*)(\boldsymbol{X} - \boldsymbol{X}^*)^\top \rangle \qquad (9)$$

where $\boldsymbol{X} \in \mathbb{R}^{d_L \times d_R}$, $\boldsymbol{H} \in \mathcal{S}_+^{d_L}$ with $\mathrm{Tr}(\boldsymbol{H}) \leq 1$, and $\|\boldsymbol{X}^*\|_{\mathrm{op}} \leq 1$. For this function class, we compare the largest possible value of convergence rate of different algorithms given by Theorem 3.8, which is summarized in Table 2.

For each algorithm defined by Algorithm 1 with specific $\mathcal{H}$, we will pick $\mathcal{X} = \{\boldsymbol{x} \mid \|\boldsymbol{x}\|_{\mathcal{H}} \leq \|\boldsymbol{x}^*\|_{\mathcal{H}}\}$ so that the global minimizer $\boldsymbol{x}^*$ is reachable. To get the convergence rate, it boils down to calculate $\|\mathcal{X}\|_{\mathcal{H}} = \|\boldsymbol{x}^*\|_{\mathcal{H}}$ and $H(L, \mathcal{H})$ associated with each algorithm. We have already derived the explicit form of $\|\mathcal{X}\|_{\mathcal{H}}$ for each algorithm in Section 3.3, as shown in the third column of Table 2. For the $\mathcal{H}$-smoothness, note that $\nabla^2 L(\boldsymbol{X}) = \boldsymbol{H} \otimes \boldsymbol{I}_{d_R}$ for the loss $L$ in (9), and then it is straightforward to calculate $H(L, \mathcal{H})$ according to Definition 3.6, as shown in the fourth column of Table 2.

Below we present the worst case of the convergence rate on this problem class. We can see that one-sided Shampoo is strictly better the other three adaptive algorithms.

**One-sided Shampoo.** The worst case of convergence rate for one-sided Shampoo is $\frac{1}{T}$.

**AdaGrad-Norm.** Since $\max_{\mathrm{Tr}(\boldsymbol{H}) \leq 1} \lambda_{\max}(\boldsymbol{H}) = 1$ and $\max_{\|\boldsymbol{X}^*\|_{\mathrm{op}} \leq 1} \|\overline{\mathrm{vec}}(\boldsymbol{X}^*)\|_2 = \max_{\|\boldsymbol{X}^*\|_{\mathrm{op}} \leq 1} \|(\boldsymbol{X}^*)\|_{\mathrm{F}} = \sqrt{\min\{d_L, d_R\}}$, the worst case of convergence rate is $\min(d_L, d_R)/T$.

**AdaGrad.** For psd matrix $\boldsymbol{H} \in \mathbb{R}^{d_L \times d_L}$ with $\mathrm{Tr}(\boldsymbol{H}) \leq 1$,

it holds that $\boldsymbol{H} \preceq \boldsymbol{I}_{d_L}$. Then we have

$$\max_{\mathrm{Tr}(\boldsymbol{H}) \leq 1} H(\boldsymbol{H}, \mathcal{D}_+^{d_L}) = \max_{\mathrm{Tr}(\boldsymbol{H}) \leq 1} \min_{\mathrm{diag}(\boldsymbol{D}) \succeq \boldsymbol{H}} \mathrm{Tr}(\boldsymbol{D}) \leq d_L.$$

On the other hand, if we choose $\boldsymbol{H} = \mathbf{1}_{d_L} \mathbf{1}_{d_L}^\top / d_L$, for any diagonal $\boldsymbol{D} \succeq \boldsymbol{H}$, it holds that $\mathrm{Tr}(\boldsymbol{D}) = \mathbf{1}_{d_L}^\top \boldsymbol{D} \mathbf{1}_{d_L} \geq \mathbf{1}_{d_L}^\top \boldsymbol{H} \mathbf{1}_{d_L} = d_L$. So we prove that $\max_{\mathrm{Tr}(\boldsymbol{H}) \leq 1} H(\boldsymbol{H}, \mathcal{D}_+^{d_L}) = d_L$.

We also know that $\max_{\|\boldsymbol{X}^*\|_{\mathrm{op}} \leq 1} \|\overline{\mathrm{vec}}(\boldsymbol{X}^*)\|_\infty \leq 1$ since $\|\boldsymbol{A}\|_\infty \leq \|\boldsymbol{A}\|_{\mathrm{op}}$. If the only nonzero entry of $\boldsymbol{X}^*$ is $\boldsymbol{X}_{1,1}^* = 1$, then $\|\boldsymbol{X}^*\|_\infty = \|\boldsymbol{X}^*\|_{\mathrm{op}} = 1$. Overall, the worst case of convergence rate is $\frac{d_L d_R}{T}$.

**Full-matrix AdaGrad.** We have seen in AdaGrad-Norm that $\max_{\|\boldsymbol{X}^*\|_{\mathrm{op}} \leq 1} \|\overline{\mathrm{vec}}(\boldsymbol{X}^*)\|_2 = \sqrt{\min\{d_L, d_R\}}$, so the worst case of convergence rate is $\frac{\min(d_L, d_R) d_R}{T}$.

To summarize, we have identified a class of optimization problems, namely loss functions like Equation (9), with Hessian of bounded trace and optimizer of bounded spectral norm, for which one-sided Shampoo has much better worst case convergence rate than any other adaptive algorithms that could be derived from our unified analysis.

## 5. Experiments

In this section we empirically demonstrate the superior performance of 1-sided shampoo over other variants of AdaReg (Algorithm 1) on a simple but natural setting. Moreover, such superior performance is predicted by our theoretical analysis in Section 4.3, which in turn validates the practical utility of our theory in guiding optimizer selection.

**Setup.** We consider a linear regression problem $\|\boldsymbol{A}\boldsymbol{X} - \boldsymbol{y}\|_2^2$ where $\boldsymbol{A}$ is the data matrix and $\boldsymbol{y} = \boldsymbol{A}\boldsymbol{X}^*$ is the label vector generated by ground-truth $\boldsymbol{X}^*$. Thus, the loss function can be equivalently written as

$$f(\boldsymbol{X}) = \langle \boldsymbol{H}, (\boldsymbol{X} - \boldsymbol{X}^*)(\boldsymbol{X} - \boldsymbol{X}^*)^\top \rangle,$$

which is the same function that we studied in Section 4.3 and show that 1-sided shampoo outperforms other adaptive algorithms. We consider $\boldsymbol{X} \in \mathbb{R}^{d \times d}$ with $d = 10^3$. We set the eigenvalues of $\boldsymbol{H}$ by $\sigma_1 = \cdots = \sigma_{10} = 1$ and $\sigma_i = \frac{1}{(i-10)^2}$ for $11 \leq i \leq 10^3$. Each element of the solution $\boldsymbol{X}^*$ is independently sampled from $\mathcal{N}(0, \frac{1}{d})$. We run AdaGrad-Norm, AdaGrad, one-sided Shampoo and full-matrix AdaGrad for 100 steps from initialization $\boldsymbol{X}_0 = 0$. We also run the original Shampoo algorithm for comparison. Full-matrix AdaGrad is run in a memory-efficient way and the detail is in Appendix F.1. We will compare the last iterate loss and the average iterate loss separately. The learning rate is tuned over five seeds for last iterate loss and average iterate loss respectfully, selecting the one with the

| Algorithm | Subalgebra $\mathcal{K}$ | $\|\boldsymbol{x}^*\|_{\mathcal{H}}$ | $H(L, \mathcal{H})$ | Convergence Rate |
|---|---|---|---|---|
| AdaGrad-Norm | $\{c\boldsymbol{I}_d \mid c \in \mathbb{R}\}$ | $\frac{1}{\sqrt{d}}\|\boldsymbol{x}^*\|_2$ | $d \cdot \lambda_{\max}(\boldsymbol{H})$ | $\|\mathcal{X}\|_2^2 \, \lambda_{\max}(\boldsymbol{H})/T$ |
| AdaGrad | Diagonal matrices, $\mathcal{D}^d(\mathbb{R})$ | $\|\boldsymbol{x}^*\|_{\infty}$ | $d_R \cdot H(\boldsymbol{H}, \mathcal{D}_+^{d_L})$ | $\|\mathcal{X}\|_{\infty}^2 \, d_R \cdot H(\boldsymbol{H}, \mathcal{D}_+^{d_L})/T$ |
| Full-Matrix AdaGrad | All matrices, $\mathbb{R}^{d \times d}$ | $\|\boldsymbol{x}^*\|_2$ | $d_R \operatorname{Tr}(\boldsymbol{H})$ | $\|\mathcal{X}\|_2^2 \, d_R \operatorname{Tr}(\boldsymbol{H})/T$ |
| One-Sided Shampoo | $\mathbb{R}^{d_L \times d_L} \otimes \boldsymbol{I}_{d_R}$ | $\frac{1}{\sqrt{d_R}}\|\boldsymbol{X}^*\|_{\mathrm{op}}$ | $d_R \operatorname{Tr}(\boldsymbol{H})$ | $\|\mathcal{X}\|_{\mathrm{op}}^2 \operatorname{Tr}(\boldsymbol{H})/T$ |

*Table 2.* Convergence rate for the loss function $L(\boldsymbol{X}) = \langle \boldsymbol{H}, (\boldsymbol{X} - \boldsymbol{X}^*)(\boldsymbol{X} - \boldsymbol{X}^*)^\top \rangle$. The results can be obtained by Theorem 3.8 with $\sigma = 0$ and omitting $\epsilon$. For each algorithm we pick the smallest domain which still ensures the minimizer lies in the domain. Here recall that $\mathcal{D}^d$ is the set of all $d$-dimensional diagonal matrices, and $\mathcal{D}_+^d = \mathcal{D}^d \cap \mathcal{S}_+^d$.

lowest average loss. We use precision float32 and set $\epsilon = 0$ for all the experiments.

We also run the EMA version of the adaptive algorithms, whose results are consistent as shown in Appendix F.2.

**Results.** The overall results are shown in Section 5. We can see that one-sided Shampoo greatly outperforms other algorithms, corroborating the theoretical analysis in Section 4.3. Moreover, full-matrix AdaGrad is apparently the worst, suggesting more or even full adaptivity does not always help optimization for fixed budget of training steps. One-sided Shampoo is also better than the original Shampoo algorithm.

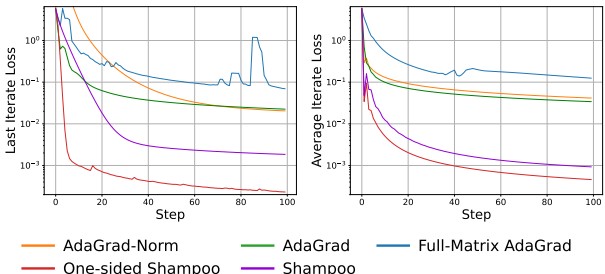

*Figure 1.* For both the last iterate loss $f(\boldsymbol{X}_t)$ and the average iterate training $f(\frac{1}{t}\sum_{s=1}^t \boldsymbol{X}_s)$, one-sided Shampoo performs the best and Full-matrix AdaGrad performs the worst. Here $f(\boldsymbol{X}) = \langle \boldsymbol{H}, (\boldsymbol{X} - \boldsymbol{X}^*)(\boldsymbol{X} - \boldsymbol{X}^*)^\top \rangle$.

## 6. Related Work

**Adaptive optimizers and structured preconditioners.** The extensive costs for training large-scale deep learning models have motivated the development of efficient optimization algorithms, among which adaptive optimizers have been widely studied because of their ability to exploit the rich geometry of the loss landscape (Pascanu & Bengio, 2013; Martens & Grosse, 2015; Dozat, 2016; Loshchilov & Hutter, 2018; Shazeer & Stern, 2018; Reddi et al., 2019; You et al., 2019; Zhuang et al., 2020; Liu et al., 2023; Yuan et al., 2024). For the sake of memory and computational efficiency, many works involve approximations to full-matrix precondi-

tioners (Ba et al., 2017; George et al., 2018; Martens et al., 2018; Yao et al., 2021; Jahani et al., 2021; Zhang et al., 2022; Duvvuri et al., 2024). Indeed, our results suggest that such compromises might not harm the performance of adaptive optimizers in practice, because more adaptivity is not always helpful, as discussed in Section 4.3 and Section 5.

**Understanding Shampoo.** There are also recent efforts to understand the Shampoo optimizer from various perspectives. Bernstein & Newhouse (2024b) interpret Shampoo as the steepest descent with respect to the spectral norm of the layerwise matrix-form parameters of the neural network. Recognizing such structures of matrix-form parameters and role of spectral-norm geometry in deep learning has led to development of new optimizer such as Muon (Jordan et al., 2024). In addition, the second-order perspective on Shampoo (Anil et al., 2020) has also led to fruitful results: Morwani et al. (2024) connect the preconditioner in Shampoo to the optimal Kronecker product approximation of the Gauss-Newton component of the Hessian, and Vyas et al. (2024) propose to view Shampoo as Adafactor in the eigenbasis of the preconditioner of Shampoo.

## 7. Conclusion and Future Works

We present a unified analysis for a broad class of adaptive optimization algorithms with well-structured preconditioners (Definition 3.1) for both online regret minimization and smooth convex optimization. Our analysis not only provides matching rate to several important algorithms including diagonal AdaGrad, full-matrix AdaGrad, and AdaGradNorm, but also gives an improved convergence rate for a one-sided variant of Shampoo over that of the original Shampoo. We reveal a novel trade-off in final convergence rate between domain metric and adaptive gradient norm (Equation (7)) for regret minimization or adaptive smoothness (Definition 3.6) for smooth convex optimization. We hope this insight could be useful towards design of future adaptive optimizers. One important future direction is to identify more subalgebras or other structures that are useful for improving the performance by better adapting to the domain or loss smoothness.

## Impact Statement

The goal of this paper is to advance the field of Machine Learning by providing theoretical and experimental insights for adaptive optimization algorithms. There are many potential societal consequences of our work, none of which we feel must be specifically highlighted here.

## Acknowledgment

The authors sincerely thank Matt Streeter for his valuable discussions and insightful comments throughout this work.

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

## A. Proof for Well-Structured Preconditioner Sets

### A.1. Definition of $\mathcal{H}$ and its properties

Recall that $\mathcal{K}$ is a subalgebra of $d$-by-$d$ real-valued matrices, and the corresponding well-structured preconditioner set is $\mathcal{H} = \mathcal{K} \cap \mathcal{S}_+^d$.

**Lemma A.1.** *For any $\boldsymbol{A} \in \mathcal{K}$ and any polynomial $p$, $p(\boldsymbol{A}) \in \mathcal{K}$. Furthermore, for any invertible $\boldsymbol{A} \in \mathcal{K}$, its inverse $\boldsymbol{A}^{-1} \in \mathcal{K}$. Also, for any symmetric $\boldsymbol{A} \in \mathcal{K}$, its pseudo inverse $\boldsymbol{A}^\dagger \in \mathcal{K}$.*

*Proof of Lemma A.1.* The first statement follows from the fact that $\mathcal{K}$ is a subalgebra, and the second and third statement are consequences of the Cayley-Hamilton theorem. $\square$

**Lemma A.2.** *For $\mathcal{H} = \mathcal{K} \cap \mathcal{S}_+^d$ where $\mathcal{K}$ is a subalgebra of $d$-by-$d$ real-valued matrices, define*

$$\mathcal{H}^* = \{\boldsymbol{A} \in \mathcal{K} : \langle \boldsymbol{H}, \boldsymbol{A} \rangle \geq 0, \forall \boldsymbol{H} \in \mathcal{H}\}. \tag{10}$$

*Then $\mathcal{H}^* = \mathcal{H}$. Consequently, for any $\boldsymbol{H}_1, \boldsymbol{H}_2 \in \mathcal{H}$, if $\langle \boldsymbol{H}_1 - \boldsymbol{H}_2, \boldsymbol{H} \rangle \geq 0$ for all $\boldsymbol{H} \in \mathcal{H}$, then $\boldsymbol{H}_1 \succeq \boldsymbol{H}_2$.*

*Proof of Lemma A.2.* Suppose there exists $\boldsymbol{A} \in \mathcal{H}^*$ such that $\boldsymbol{A}$ has a negative eigenvalue. Let $\lambda_1(\boldsymbol{A}), \ldots, \lambda_d(\boldsymbol{A})$ be the eigenvalues of $\boldsymbol{A}$ in the decreasing order. Consider the matrix $\boldsymbol{B} = (\boldsymbol{A} - 2\max(\lambda_1(\boldsymbol{A}), 1)\boldsymbol{I}_d)^2 \succ 0$. The leading eigenspace of $\boldsymbol{B}$ is the same as the eigenspace of $\boldsymbol{A}$ corresponding to its smallest eigenvalue, which is negative. Consequently, for large enough integer $n$, $\boldsymbol{B}^n / \|\boldsymbol{B}^n\|_{\mathrm{op}}$ is approximately the projection matrix onto the eigenspace of the smallest negative eigenvalue of $\boldsymbol{A}$. Therefore, for large enough integer $n$, $\langle \boldsymbol{B}^n / \|\boldsymbol{B}^n\|_{\mathrm{op}}, \boldsymbol{A} \rangle < 0$. However, since $\boldsymbol{B}^n / \|\boldsymbol{B}^n\|_{\mathrm{op}}$ is a polynomial of $\boldsymbol{A}$, we know that $\boldsymbol{B}^n / \|\boldsymbol{B}^n\|_{\mathrm{op}} \in \mathcal{H}$. This is a contradiction to the definition of $\mathcal{H}^*$. Hence, we conclude that for any $\boldsymbol{A} \in \mathcal{H}^*$, it holds that $\boldsymbol{A} \succeq 0$, and thus $\mathcal{H}^* \subseteq \mathcal{H}$. Moreover, for any $\boldsymbol{A} \in \mathcal{H}$, it holds that $\langle \boldsymbol{H}, \boldsymbol{A} \rangle \geq 0$ for any $\boldsymbol{H} \in \mathcal{H}$ because both $\boldsymbol{A}$ and $\boldsymbol{H}$ are positive semi-definite. This shows that $\mathcal{H} \subseteq \mathcal{H}^*$, and hence $\mathcal{H}^* = \mathcal{H}$. This completes the proof. $\square$

### A.2. Properties of $P_{\mathcal{H}}(\cdot)$

For any $\boldsymbol{M} \succ 0$, recall the regularized optimization problem in (3). Note that we can assume $\eta = 1$ without loss of generality, because the original problem is equivalent to solve for $\boldsymbol{M}/\eta^2$ with regularizer $\mathrm{Tr}(\boldsymbol{H})$ in place of $\eta^2 \mathrm{Tr}(\boldsymbol{H})$. Therefore, in the rest of this section, we focus on the following optimization problem:

$$P_{\mathcal{H}}(\boldsymbol{M}) := \arg\min_{\boldsymbol{H} \in \mathcal{H}} \langle \boldsymbol{M}, \boldsymbol{H}^{-1} \rangle + \mathrm{Tr}(\boldsymbol{H}). \tag{11}$$

The results proved below applies to the original problem (3) after simple rescaling, and Proposition 3.2 follows from Proposition A.3 below.

For notational convenience, given any $\boldsymbol{M} \succ 0$, we define

$$f_{\boldsymbol{M}}(\boldsymbol{H}) := \langle \boldsymbol{M}, \boldsymbol{H}^{-1} \rangle + \mathrm{Tr}(\boldsymbol{H}). \tag{12}$$

**Proposition A.3.** *Let $\mathcal{H}$ be a well-structured preconditioner set under Definition 3.1. For any $\boldsymbol{M} \succ 0$, there exists a unique solution $P_{\mathcal{H}}(\boldsymbol{M}) \succ 0$ to the optimization problem in (11). Furthermore, for any $\boldsymbol{M} \succ 0$, $P_{\mathcal{H}}(\boldsymbol{M})$ satisfies the following properties:*

*(a) $\langle \boldsymbol{M}, P_{\mathcal{H}}(\boldsymbol{M})^{-1} \rangle = \mathrm{Tr}(P_{\mathcal{H}}(\boldsymbol{M}))$.*

*(b) $\overline{P_{\mathcal{H}}(\boldsymbol{M})} = \arg\min_{\boldsymbol{H} \in \mathcal{H}, \mathrm{Tr}(\boldsymbol{H}) \leq 1} \langle \boldsymbol{M}, \boldsymbol{H}^{-1} \rangle$ where we recall that $\overline{P_{\mathcal{H}}(\boldsymbol{M})} = \mathrm{Tr}(P_{\mathcal{H}}(\boldsymbol{M}))^{-1} P_{\mathcal{H}}(\boldsymbol{M})$.*

*(c) For any $\boldsymbol{H} \in \mathcal{H}$, $\langle -P_{\mathcal{H}}(\boldsymbol{M})^{-1} \boldsymbol{M} P_{\mathcal{H}}(\boldsymbol{M})^{-1} + \boldsymbol{I}_d, \boldsymbol{H} - P_{\mathcal{H}}(\boldsymbol{M}) \rangle = 0$.*

*Moreover, for any $\boldsymbol{M}_1 \succeq \boldsymbol{M}_2 \succ 0$, it holds that $P_{\mathcal{H}}(\boldsymbol{M}_1) - P_{\mathcal{H}}(\boldsymbol{M}_2) \in \mathcal{H}$, and in particular, $P_{\mathcal{H}}(\boldsymbol{M}_1) \succeq P_{\mathcal{H}}(\boldsymbol{M}_2)$.*

*Proof of Proposition A.3.* We first show that $P_{\mathcal{H}}(\boldsymbol{M})$ exists and $P_{\mathcal{H}}(\boldsymbol{M}) \succ 0$. Note that for $\mathcal{H} = \mathcal{K} \cap \mathcal{S}_+^d$, since $\mathcal{K}$ is a linear subspace of $\mathcal{M}^d$ and $\mathcal{S}_+^d$ is a closed subset of $\mathcal{M}^d$, we know that $\mathcal{H}$ is also a closed subset of $\mathcal{M}^d$. Moreover, for any sequence $\{\boldsymbol{H}_n\}_{n \geq 1}$ such that $\boldsymbol{H}_n \succ 0$ and either the smallest eigenvalue of $\boldsymbol{H}_n$ converges to 0 or the largest eigenvalue of $\boldsymbol{H}_n$ converges to $\infty$, the objective value $f_{\boldsymbol{M}}(\boldsymbol{H}_n)$ goes to $\infty$ because $\boldsymbol{M} \succ 0$. Therefore, there exists $P_{\mathcal{H}}(\boldsymbol{M}) \succ 0$ that attains the minimum objective value. For the uniqueness of $P_{\mathcal{H}}(\boldsymbol{M})$, it suffices to note that the objective function $f_{\boldsymbol{M}}(\boldsymbol{H})$ is strictly convex in $\boldsymbol{H} \succ 0$ for $\boldsymbol{M} \succ 0$.

**Proof for property (a).** Suppose otherwise that $\langle \boldsymbol{M}, P_{\mathcal{H}}(\boldsymbol{M})^{-1} \rangle \neq \mathrm{Tr}(P_{\mathcal{H}}(\boldsymbol{M}))$, and consider the following matrix:

$$\boldsymbol{H} = P_{\mathcal{H}}(\boldsymbol{M}) \cdot \sqrt{\langle \boldsymbol{M}, P_{\mathcal{H}}(\boldsymbol{M})^{-1} \rangle / \mathrm{Tr}(P_{\mathcal{H}}(\boldsymbol{M}))} \in \mathcal{H}.$$

For this $\boldsymbol{H}$, we have $\langle \boldsymbol{M}, \boldsymbol{H}^{-1} \rangle + \mathrm{Tr}(\boldsymbol{H}) = 2\sqrt{\langle \boldsymbol{M}, P_{\mathcal{H}}(\boldsymbol{M})^{-1} \rangle \cdot \mathrm{Tr}(P_{\mathcal{H}}(\boldsymbol{M}))} < \langle \boldsymbol{M}, P_{\mathcal{H}}(\boldsymbol{M})^{-1} \rangle + \mathrm{Tr}(P_{\mathcal{H}}(\boldsymbol{M}))$, thus contradicting the optimality of $P_{\mathcal{H}}(\boldsymbol{M})$. Therefore, it must be true that $\langle \boldsymbol{M}, P_{\mathcal{H}}(\boldsymbol{M})^{-1} \rangle = \mathrm{Tr}(P_{\mathcal{H}}(\boldsymbol{M}))$.

**Proof for property (b)** Note that we can rewrite the original optimization problem as follows:

$$P_{\mathcal{H}}(\boldsymbol{M}) = \arg\min_{\boldsymbol{H} \in \mathcal{H}} \langle \boldsymbol{M}, \boldsymbol{H}^{-1} \rangle + \mathrm{Tr}(\boldsymbol{H})$$

$$= \arg\min_{\boldsymbol{H} \in \mathcal{H}} \left\langle \boldsymbol{M}, \left(\frac{\boldsymbol{H}}{\mathrm{Tr}(\boldsymbol{H})}\right)^{-1} \right\rangle \cdot \frac{1}{\mathrm{Tr}(\boldsymbol{H})} + \mathrm{Tr}(\boldsymbol{H}).$$

Note that solving the above optimization problem is equivalent to first solving $\overline{P_{\mathcal{H}}(\boldsymbol{M})} = \arg\min_{\boldsymbol{H} \in \mathcal{H}, \mathrm{Tr}(\boldsymbol{H}) \leq 1} \langle \boldsymbol{M}, \boldsymbol{H}^{-1} \rangle$ and then setting $P_{\mathcal{H}}(\boldsymbol{M}) = \mathrm{Tr}(P_{\mathcal{H}}(\boldsymbol{M})) \cdot \overline{P_{\mathcal{H}}(\boldsymbol{M})}$ where the value of $\mathrm{Tr}(P_{\mathcal{H}}(\boldsymbol{M}))$ ensures the previous property (a). Hence, we see that $\overline{P_{\mathcal{H}}(\boldsymbol{M})}$ solves the constrained version of the original optimization problem.

**Proof for property (c).** Since $\nabla_{\boldsymbol{H}} \mathrm{Tr}(\boldsymbol{H}) = \boldsymbol{I}_d$ and $\nabla_{\boldsymbol{H}} \mathrm{Tr}(\boldsymbol{M}^\top \boldsymbol{H}^{-1}) = -\boldsymbol{H}^{-1} \boldsymbol{M} \boldsymbol{H}^{-1}$ (see e.g. Equation (124) in Petersen et al. (2008)), we have

$$\nabla_{\boldsymbol{H}} f_{\boldsymbol{M}}(\boldsymbol{H}) = -\boldsymbol{H}^{-1} \boldsymbol{M} \boldsymbol{H}^{-1} + \boldsymbol{I}_d. \tag{13}$$

Then as $\mathcal{H}$ is a cone, by the optimality of $P_{\mathcal{H}}(\boldsymbol{M})$, it holds for any $\boldsymbol{H} \in \mathcal{H}$ that

$$0 = \langle \nabla_{\boldsymbol{H}} f_{\boldsymbol{M}}(P_{\mathcal{H}}(\boldsymbol{M})), \boldsymbol{H} - P_{\mathcal{H}}(\boldsymbol{M}) \rangle = \langle -P_{\mathcal{H}}(\boldsymbol{M})^{-1} \boldsymbol{M} P_{\mathcal{H}}(\boldsymbol{M})^{-1} + \boldsymbol{I}_d, \boldsymbol{H} - P_{\mathcal{H}}(\boldsymbol{M}) \rangle$$

**Proof for the operator monotonicity of $P_{\mathcal{H}}(\cdot)$.** By property (c) of $P_{\mathcal{H}}(\cdot)$, we know that for any $\boldsymbol{M} \succ 0$, $\langle \boldsymbol{M} - P_{\mathcal{H}}(\boldsymbol{M})^2, P_{\mathcal{H}}(\boldsymbol{M})^{-1} \boldsymbol{H} P_{\mathcal{H}}(\boldsymbol{M})^{-1} - P_{\mathcal{H}}(\boldsymbol{M})^{-1} \rangle = 0$. Note that $\boldsymbol{H} \mapsto P_{\mathcal{H}}(\boldsymbol{M})^{-1} \boldsymbol{H} P_{\mathcal{H}}(\boldsymbol{M})^{-1}$ is a bijection from $\mathcal{H}$ to $\mathcal{H}$ by Lemma A.1, so we have $\langle \boldsymbol{M} - P_{\mathcal{H}}(\boldsymbol{M})^2, \boldsymbol{H} - P_{\mathcal{H}}(\boldsymbol{M})^{-1} \rangle = 0$ for all $\boldsymbol{H} \in \mathcal{H}$. Applying this to both $\boldsymbol{M}_1$ and $\boldsymbol{M}_2$, it follows that for any $\boldsymbol{H} \in \mathcal{H}$

$$\langle \boldsymbol{M}_1 - P_{\mathcal{H}}(\boldsymbol{M}_1)^2, \boldsymbol{H} - P_{\mathcal{H}}(\boldsymbol{M}_1)^{-1} \rangle = \langle \boldsymbol{M}_2 - P_{\mathcal{H}}(\boldsymbol{M}_2)^2, \boldsymbol{H} - P_{\mathcal{H}}(\boldsymbol{M}_2)^{-1} \rangle.$$

Rearranging the above equation, we obtain

$$\langle P_{\mathcal{H}}(\boldsymbol{M}_1)^2 - P_{\mathcal{H}}(\boldsymbol{M}_2)^2, \boldsymbol{H} \rangle = \langle \boldsymbol{M}_1 - \boldsymbol{M}_2, \boldsymbol{H} \rangle - \langle \boldsymbol{M}_1, P_{\mathcal{H}}(\boldsymbol{M}_1)^{-1} \rangle + \mathrm{Tr}(P_{\mathcal{H}}(\boldsymbol{M}_1))$$
$$+ \langle \boldsymbol{M}_2, P_{\mathcal{H}}(\boldsymbol{M}_2)^{-1} \rangle - \mathrm{Tr}(P_{\mathcal{H}}(\boldsymbol{M}_2))$$
$$= \langle \boldsymbol{M}_1 - \boldsymbol{M}_2, \boldsymbol{H} \rangle$$

where the second equality follows from the first property of $P_{\mathcal{H}}(\boldsymbol{M}_1)$ and $P_{\mathcal{H}}(\boldsymbol{M}_2)$ from Proposition A.3. Since $\boldsymbol{M}_1 \succeq \boldsymbol{M}_2$, this implies that $\langle P_{\mathcal{H}}(\boldsymbol{M}_1)^2 - P_{\mathcal{H}}(\boldsymbol{M}_2)^2, \boldsymbol{H} \rangle \geq 0$ for all $\boldsymbol{H} \in \mathcal{H}$. By Lemma A.1, we know that $P_{\mathcal{H}}(\boldsymbol{M}_1)^2 - P_{\mathcal{H}}(\boldsymbol{M}_2)^2 \in \mathcal{H}$, so it further follows from Lemma A.2 that $P_{\mathcal{H}}(\boldsymbol{M}_1)^2 \succeq P_{\mathcal{H}}(\boldsymbol{M}_2)^2$. Since matrix square root is operator monotone, it holds that $P_{\mathcal{H}}(\boldsymbol{M}_1) \succeq P_{\mathcal{H}}(\boldsymbol{M}_2)$. We also know that $P_{\mathcal{H}}(\boldsymbol{M}_1) - P_{\mathcal{H}}(\boldsymbol{M}_2) \in \mathcal{K}$ because $\mathcal{K}$ is a subalgebra. Then we conclude that $P_{\mathcal{H}}(\boldsymbol{M}_1) - P_{\mathcal{H}}(\boldsymbol{M}_2) \in \mathcal{H}$ from the definition of $\mathcal{H}$. This completes the proof. $\qquad\square$

We can further extend the definition of $P_{\mathcal{H}}(\cdot)$ to all positive semi-definite matrices. Specifically, for any $\boldsymbol{M} \succeq 0$, we have $\boldsymbol{M} + \epsilon \boldsymbol{I}_d \succ 0$ for any $\epsilon > 0$, so $P_{\mathcal{H}}(\boldsymbol{M} + \epsilon \boldsymbol{I}_d)$ is well-defined. Also, by the operator monotonicity of $P_{\mathcal{H}}(\cdot)$ from

Proposition A.3, $P_{\mathcal{H}}(\boldsymbol{M} + \epsilon \boldsymbol{I}_d) \preceq P_{\mathcal{H}}(\boldsymbol{M} + \epsilon' \boldsymbol{I}_d)$ for $\epsilon' \geq \epsilon > 0$. This implies that $P_{\mathcal{H}}(\boldsymbol{M} + \epsilon \boldsymbol{I}_d)$ has a limit as $\epsilon \to 0$. Therefore, for any $\boldsymbol{M} \succeq 0$, we define

$$P_{\mathcal{H}}(\boldsymbol{M}) = \lim_{\epsilon \searrow 0} P_{\mathcal{H}}(\boldsymbol{M} + \epsilon \boldsymbol{I}_d). \tag{14}$$

Note that the above equality is also true for $\boldsymbol{M} \succ 0$. To see this, we apply the optimality of every $P_{\mathcal{H}}(\boldsymbol{M} + \epsilon \boldsymbol{I}_d)$ to get

$$\langle \boldsymbol{M} + \epsilon \boldsymbol{I}_d, P_{\mathcal{H}}(\boldsymbol{M})^{-1} \rangle + \mathrm{Tr}(P_{\mathcal{H}}(\boldsymbol{M})) > \langle \boldsymbol{M} + \epsilon \boldsymbol{I}_d, P_{\mathcal{H}}(\boldsymbol{M} + \epsilon \boldsymbol{I}_d)^{-1} \rangle + \mathrm{Tr}(P_{\mathcal{H}}(\boldsymbol{M} + \epsilon \boldsymbol{I}_d)).$$

Also note that $P_{\mathcal{H}}(\boldsymbol{M} + \epsilon \boldsymbol{I}_d) \succ P_{\mathcal{H}}(\boldsymbol{M} - \delta \boldsymbol{I}_d) \succ 0$ for sufficiently small $\delta > 0$ such that $\boldsymbol{M} - \delta \boldsymbol{I}_d \succ 0$. Therefore, letting $\epsilon \to 0$ on both sides of the previous inequality, we can exchange the order of taking the limit and taking the inverse of $P_{\mathcal{H}}(\boldsymbol{M} + \epsilon \boldsymbol{I}_d)$ to obtain

$$\langle \boldsymbol{M}, P_{\mathcal{H}}(\boldsymbol{M})^{-1} \rangle + \mathrm{Tr}(P_{\mathcal{H}}(\boldsymbol{M})) \geq \left\langle \boldsymbol{M}, \left( \lim_{\epsilon \searrow 0} P_{\mathcal{H}}(\boldsymbol{M} + \epsilon \boldsymbol{I}_d) \right)^{-1} \right\rangle + \mathrm{Tr}\left( \lim_{\epsilon \searrow 0} P_{\mathcal{H}}(\boldsymbol{M} + \epsilon \boldsymbol{I}_d) \right).$$

Then by the optimality of $P_{\mathcal{H}}(\boldsymbol{M})$ and its uniqueness, we conclude that (14) is also valid for any $\boldsymbol{M} \succ 0$.

Indeed, the definition of $P_{\mathcal{H}}(\boldsymbol{M})$ in (14) provides a continuous extension of $P_{\mathcal{H}}$ to $\mathcal{S}_+^d$, as summarized in the following proposition.

**Proposition A.4.** *Let $\mathcal{H}$ be a well-structured preconditioner set under Definition 3.1. As a function on $\mathcal{S}_{++}^d$, $P_{\mathcal{H}}$ can be continuously extended to be a function on $\mathcal{S}_+^d$. Moreover, for any $\boldsymbol{M} \succeq 0$ such that $\boldsymbol{M} \neq 0$, $P_{\mathcal{H}}(\boldsymbol{M}) \succeq 0$ satisfies the following properties:*

(a) $\mathrm{span}(\boldsymbol{M}) \subseteq \mathrm{span}(P_{\mathcal{H}}(\boldsymbol{M}))$ *and* $\langle \boldsymbol{M}, P_{\mathcal{H}}(\boldsymbol{M})^{\dagger} \rangle + \mathrm{Tr}(P_{\mathcal{H}}(\boldsymbol{M})) = \inf_{\boldsymbol{H} \in \mathcal{H}} \langle \boldsymbol{M}, \boldsymbol{H}^{-1} \rangle + \mathrm{Tr}(\boldsymbol{H})$.

(b) $\langle \boldsymbol{M}, P_{\mathcal{H}}(\boldsymbol{M})^{\dagger} \rangle = \mathrm{Tr}(P_{\mathcal{H}}(\boldsymbol{M}))$.

(c) $\langle \boldsymbol{M}, \overline{P_{\mathcal{H}}(\boldsymbol{M})}^{\dagger} \rangle = \inf_{\boldsymbol{H} \in \mathcal{H}, \mathrm{Tr}(\boldsymbol{H}) \leq 1} \langle \boldsymbol{M}, \boldsymbol{H}^{-1} \rangle$ *where we recall that* $\overline{P_{\mathcal{H}}(\boldsymbol{M})} = \mathrm{Tr}(P_{\mathcal{H}}(\boldsymbol{M}))^{-1} P_{\mathcal{H}}(\boldsymbol{M})$.

(d) *For any* $\boldsymbol{H} \in \mathcal{H}$, $\langle P_{\mathcal{H}}(\boldsymbol{M})^{\dagger} \boldsymbol{M} P_{\mathcal{H}}(\boldsymbol{M})^{\dagger} - \boldsymbol{\Pi}_{\boldsymbol{M}}, \boldsymbol{H} - P_{\mathcal{H}}(\boldsymbol{M}) \rangle = 0$, *where* $\boldsymbol{\Pi}_{\boldsymbol{M}}$ *is the projection matrix onto* $\mathrm{span}(P_{\mathcal{H}}(\boldsymbol{M}))$.

*Moreover, for any $\boldsymbol{M}_1 \succeq \boldsymbol{M}_2 \succeq 0$, it holds that $P_{\mathcal{H}}(\boldsymbol{M}_1) - P_{\mathcal{H}}(\boldsymbol{M}_2) \in \mathcal{H}$, and in particular, $P_{\mathcal{H}}(\boldsymbol{M}_1) \succeq P_{\mathcal{H}}(\boldsymbol{M}_2)$.*

*Proof of Proposition A.4.* We divide the proof into different parts for different properties of $P_{\mathcal{H}}$.

**Proof for continuous extension of $P_{\mathcal{H}}$.** We first show that $P_{\mathcal{H}}$ can be continuously extended to $\mathcal{S}_+^d$, and we consider the extension of $P_{\mathcal{H}}$ as given in (14). We first show that for any $\boldsymbol{M} \succeq 0$ and any sequence $\{\boldsymbol{M}_n\}_{n=1}^{\infty}$ such that each $\boldsymbol{M}_n \succ 0$ and $\lim_{n \to \infty} \boldsymbol{M}_n = \boldsymbol{M}$, it holds that $\lim_{n \to \infty} P_{\mathcal{H}}(\boldsymbol{M}_n) = P_{\mathcal{H}}(\boldsymbol{M})$. Note that for any $\delta \in (0, 1)$, there exist $\bar{\epsilon}_n \geq \underline{\epsilon}_n > 0$ such that $0 \prec (1 - \delta)(\boldsymbol{M} + \underline{\epsilon}_n \boldsymbol{I}_d) \preceq \boldsymbol{M}_n \preceq \boldsymbol{M} + \bar{\epsilon}_n \boldsymbol{I}_d$ for all large enough $n$ and moreover, $\lim_{n \to \infty} \bar{\epsilon}_n = \lim_{n \to \infty} \underline{\epsilon}_n = 0$. Then by the operator monotonicity of $P_{\mathcal{H}}(\cdot)$, we have $P_{\mathcal{H}}((1 - \delta)(\boldsymbol{M} + \underline{\epsilon}_n \boldsymbol{I}_d)) \preceq P_{\mathcal{H}}(\boldsymbol{M}_n) \preceq P_{\mathcal{H}}(\boldsymbol{M} + \bar{\epsilon}_n \boldsymbol{I}_d)$. Also note that $P_{\mathcal{H}}((1 - \delta)(\boldsymbol{M} + \underline{\epsilon}_n \boldsymbol{I}_d)) = \sqrt{1 - \delta} P_{\mathcal{H}}(\boldsymbol{M} + \underline{\epsilon}_n \boldsymbol{I}_d)$. Letting $n \to \infty$, both $P_{\mathcal{H}}(\boldsymbol{M} + \underline{\epsilon}_n \boldsymbol{I}_d)$ and $P_{\mathcal{H}}(\boldsymbol{M} + \bar{\epsilon}_n \boldsymbol{I}_d)$ converge to $P_{\mathcal{H}}(\boldsymbol{M})$, which then implies that $\sqrt{1 - \delta} P_{\mathcal{H}}(\boldsymbol{M}) \preceq \lim_{n \to \infty} P_{\mathcal{H}}(\boldsymbol{M}_n) \preceq P_{\mathcal{H}}(\boldsymbol{M})$. Since $\delta$ is arbitrary, we conclude that $\lim_{n \to \infty} P_{\mathcal{H}}(\boldsymbol{M}_n) = P_{\mathcal{H}}(\boldsymbol{M})$. Next, consider any general sequence $\{\boldsymbol{M}_n\}_{n=1}^{\infty}$ such that $\boldsymbol{M}_n \succeq 0$ and $\lim_{n \to \infty} \boldsymbol{M}_n = \boldsymbol{M}$. For any $\epsilon > 0$, there exists $\delta > 0$ such that for all $\boldsymbol{M}' \succ 0$ with $\|\boldsymbol{M} - \boldsymbol{M}'\|_{\mathrm{F}} \leq \delta$, it holds that $\|P_{\mathcal{H}}(\boldsymbol{M}) - P_{\mathcal{H}}(\boldsymbol{M}')\|_{\mathrm{F}} \leq \epsilon$. For this $\delta$, there exists $N > 0$ such that for all $n > N$, $\|\boldsymbol{M} - \boldsymbol{M}_n\|_{\mathrm{F}} \leq \delta/2$. Then since $P_{\mathcal{H}}(\boldsymbol{M}_n) = \lim_{m \to \infty} P_{\mathcal{H}}(\boldsymbol{M}_n + \frac{\delta}{2dm} \boldsymbol{I}_d)$, where for every $m \geq 1$ we have $\|\boldsymbol{M} - (\boldsymbol{M}_n + \frac{\delta}{2m\sqrt{d}} \boldsymbol{I}_d)\|_{\mathrm{F}} \leq \|\boldsymbol{M} - \boldsymbol{M}_n\|_{\mathrm{F}} + \|\frac{\delta}{2m\sqrt{d}} \boldsymbol{I}_d\|_{\mathrm{F}} \leq \delta$, so $\|P_{\mathcal{H}}(\boldsymbol{M}) - P_{\mathcal{H}}(\boldsymbol{M}_n + \frac{\delta}{2m\sqrt{d}} \boldsymbol{I}_d)\|_{\mathrm{F}} \leq \epsilon$ for all $m \geq 1$, which implies that $\|P_{\mathcal{H}}(\boldsymbol{M}) - P_{\mathcal{H}}(\boldsymbol{M}_n)\|_{\mathrm{F}} \leq \epsilon$ for all $n > N$. Therefore, it follows that $\lim_{n \to \infty} P_{\mathcal{H}}(\boldsymbol{M}_n) = P_{\mathcal{H}}(\boldsymbol{M})$. In conclusion, $P_{\mathcal{H}}(\cdot)$ can be extended to be a continuous function on $\mathcal{S}_+^d$.

Below, we fix any $\boldsymbol{M} \succeq 0$ such that $\boldsymbol{M} \neq 0$.

**Proof for** $\mathrm{span}(M) \subseteq \mathrm{span}(P_{\mathcal{H}}(M))$**.** Let $\mathbf{\Pi_M} := P_{\mathcal{H}}(M)P_{\mathcal{H}}(M)^{\dagger}$ be the projection matrix onto $\mathrm{span}(P_{\mathcal{H}}(M))$, then $\mathbf{\Pi_M} \in \mathcal{H}$ by Lemma A.1. It suffices to show that $\langle I_d - \mathbf{\Pi_M}, M \rangle = 0$. Using the fact that $\mathrm{Tr}(AB) \leq \mathrm{Tr}(A)\mathrm{Tr}(B)$ for any $A, B \succeq 0$, we can get the following inequality for any $\epsilon > 0$:

$$\langle I_d - \mathbf{\Pi_M}, M \rangle = \mathrm{Tr}((I_d - \mathbf{\Pi_M})M) \leq \mathrm{Tr}((I_d - \mathbf{\Pi_M})P_{\mathcal{H}}(M + \epsilon I_d)) \cdot \mathrm{Tr}(P_{\mathcal{H}}(M + \epsilon I_d)^{-1}M)$$
$$= \langle I_d - \mathbf{\Pi_M}, P_{\mathcal{H}}(M + \epsilon I_d) \rangle \cdot \langle M, P_{\mathcal{H}}(M + \epsilon I_d)^{-1} \rangle \tag{15}$$

By Proposition A.3, we know that $\langle M, P_{\mathcal{H}}(M + \epsilon I_d)^{-1} \rangle = \mathrm{Tr}(P_{\mathcal{H}}(M + \epsilon I_d)) - \epsilon \langle I_d, P_{\mathcal{H}}(M + \epsilon I_d)^{-1} \rangle \leq \mathrm{Tr}(P_{\mathcal{H}}(M + \epsilon I_d))$, which is bounded by an absolute constant for all $\epsilon \in (0, 1)$. Therefore, letting $\epsilon \to 0$ on both sided of (15), since $\langle I_d - \mathbf{\Pi_M}, P_{\mathcal{H}}(M + \epsilon I_d) \rangle \to \langle I_d - \mathbf{\Pi_M}, P_{\mathcal{H}}(M) \rangle = 0$ by the definition of $\mathbf{\Pi_M}$, we conclude that $\langle I_d - \mathbf{\Pi_M}, M \rangle = 0$. This implies that $\mathrm{span}(M) \subseteq \mathrm{span}(P_{\mathcal{H}}(M))$.

**Proof for the optimality of** $P_{\mathcal{H}}(M)$**.** We still use $\mathbf{\Pi_M}$ to denote the projection matrix onto $\mathrm{span}(P_{\mathcal{H}}(M))$. For any $\epsilon > 0$, by the optimality of $P_{\mathcal{H}}(M + \epsilon I_d)$, it holds for any $H \in \mathcal{H}$ that

$$\langle M + \epsilon I_d, P_{\mathcal{H}}(M + \epsilon I_d)^{-1} \rangle + \mathrm{Tr}(P_{\mathcal{H}}(M + \epsilon I_d)) \leq \langle M + \epsilon I_d, H^{-1} \rangle + \mathrm{Tr}(H). \tag{16}$$

Since $\mathrm{span}(M) \subseteq \mathrm{span}(P_{\mathcal{H}}(M))$, we know that $\langle M, P_{\mathcal{H}}(M + \epsilon I_d)^{-1} \rangle = \langle M, \mathbf{\Pi_M} P_{\mathcal{H}}(M + \epsilon I_d)^{-1} \mathbf{\Pi_M} \rangle$. Then since $\lim_{\epsilon \to 0} P_{\mathcal{H}}(M + \epsilon I_d) = P_{\mathcal{H}}(M)$, it holds that $\mathbf{\Pi_M} P_{\mathcal{H}}(M + \epsilon I_d)^{-1} \mathbf{\Pi_M} = (\mathbf{\Pi_M} P_{\mathcal{H}}(M + \epsilon I_d)\mathbf{\Pi_M})^{\dagger}$ for sufficiently small $\epsilon > 0$, and also that $P_{\mathcal{H}}(M)^{\dagger} = \lim_{\epsilon \to 0}(\mathbf{\Pi_M} P_{\mathcal{H}}(M + \epsilon I_d)\mathbf{\Pi_M})^{\dagger}$. This implies

$$\lim_{\epsilon \searrow 0} \langle M, P_{\mathcal{H}}(M + \epsilon I_d)^{-1} \rangle = \langle M, P_{\mathcal{H}}(M)^{\dagger} \rangle. \tag{17}$$

Therefore, assuming the existence of the limit of $\langle \epsilon I_d, P_{\mathcal{H}}(M + \epsilon I_d)^{-1} \rangle$ as $\epsilon \to 0$, taking the limit of $\epsilon \to 0$ on both sides of (16) yields

$$\langle M, P_{\mathcal{H}}(M)^{\dagger} \rangle + \mathrm{Tr}(P_{\mathcal{H}}(M)) + \lim_{\epsilon \to 0} \langle \epsilon I_d, P_{\mathcal{H}}(M + \epsilon I_d)^{-1} \rangle \leq \langle M, H^{-1} \rangle + \mathrm{Tr}(H). \tag{18}$$

Hence, it suffices to show that $\langle \epsilon I_d, P_{\mathcal{H}}(M + \epsilon I_d)^{-1} \rangle \to 0$ as $\epsilon \to 0$. To prove this, note that for any $\epsilon > 0$, $P_{\mathcal{H}}(M) + \sqrt{\epsilon}(I_d - \mathbf{\Pi_M}) \succ 0$, and its inverse is given by $P_{\mathcal{H}}(M)^{\dagger} + \epsilon^{-1/2}(I_d - \mathbf{\Pi_M})$. Also, $P_{\mathcal{H}}(M) + \sqrt{\epsilon}(I_d - \mathbf{\Pi_M}) \in \mathcal{H}$ by Lemma A.1. Therefore, by the optimality of $P_{\mathcal{H}}(M + \epsilon I_d)$, we have

$$\langle M + \epsilon I_d, P_{\mathcal{H}}(M + \epsilon I_d)^{-1} \rangle + \mathrm{Tr}(P_{\mathcal{H}}(M + \epsilon I_d))$$
$$\leq \langle M + \epsilon I_d, (P_{\mathcal{H}}(M) + \sqrt{\epsilon}(I_d - \mathbf{\Pi_M}))^{-1} \rangle + \mathrm{Tr}(P_{\mathcal{H}}(M) + \sqrt{\epsilon}(I_d - \mathbf{\Pi_M}))$$
$$= \langle M + \epsilon I_d, P_{\mathcal{H}}(M)^{\dagger} + \epsilon^{-1/2}(I_d - \mathbf{\Pi_M}) \rangle + \mathrm{Tr}(P_{\mathcal{H}}(M)) + \sqrt{\epsilon}(d - \mathrm{rank}(P_{\mathcal{H}}(M)))$$
$$= \langle M, P_{\mathcal{H}}(M)^{\dagger} \rangle + \epsilon \cdot \mathrm{Tr}(P_{\mathcal{H}}(M)^{\dagger}) + \mathrm{Tr}(P_{\mathcal{H}}(M)) + 2\sqrt{\epsilon}(d - \mathrm{rank}(P_{\mathcal{H}}(M)))$$

where we apply $\mathrm{Tr}(I_d - \mathbf{\Pi_M}) = d - \mathrm{rank}(P_{\mathcal{H}}(M))$ and the last equality follows from the fact that $\langle M, I_d - \mathbf{\Pi_M} \rangle = 0$. Rearranging the above inequality, we obtain

$$\epsilon \langle I_d, P_{\mathcal{H}}(M + \epsilon I_d)^{-1} \rangle \leq \langle M, P_{\mathcal{H}}(M)^{\dagger} \rangle - \langle M, P_{\mathcal{H}}(M + \epsilon I_d)^{-1} \rangle + \mathrm{Tr}(P_{\mathcal{H}}(M)) - \mathrm{Tr}(P_{\mathcal{H}}(M + \epsilon I_d))$$
$$+ \epsilon \cdot \mathrm{Tr}(P_{\mathcal{H}}(M)^{\dagger}) + 2\sqrt{\epsilon}(d - \mathrm{rank}(P_{\mathcal{H}}(M))).$$

Now applying (17) and noting that $\epsilon \langle I_d, P_{\mathcal{H}}(M + \epsilon I_d)^{-1} \rangle \geq 0$, taking the limit $\epsilon \to 0$, we obtain

$$\lim_{\epsilon \to 0} \langle \epsilon I_d, P_{\mathcal{H}}(M + \epsilon I_d)^{-1} \rangle = 0. \tag{19}$$

Then combining (18) and (19), we conclude that for any $H \in \mathcal{H}$,

$$\langle M, P_{\mathcal{H}}(M)^{\dagger} \rangle + \mathrm{Tr}(P_{\mathcal{H}}(M)) \leq \langle M, H^{-1} \rangle + \mathrm{Tr}(H).$$

This confirms the optimality of $P_{\mathcal{H}}(M)$. Further applying the same argument as in the proof of Proposition A.3 yields the optimality of $\overline{P_{\mathcal{H}}(M)} = \mathrm{Tr}(P_{\mathcal{H}}(M))^{-1}P_{\mathcal{H}}(M)$.

**Proof for property (d) of $P_{\mathcal{H}}(M)$.** By the optimality of $P_{\mathcal{H}}(M)$, we have

$$\langle M, P_{\mathcal{H}}(M)^{\dagger}\rangle + \mathrm{Tr}(P_{\mathcal{H}}(M)) = \inf_{H \in \mathcal{H}} \underbrace{\langle M, H^{-1}\rangle + \mathrm{Tr}(H)}_{f_M(H)}$$

$$= \inf_{H \in \mathcal{H}} \underbrace{\langle M, H^{-1}\rangle + \mathrm{Tr}(\Pi_M H \Pi_M)}_{\tilde{f}_M(H)}. \tag{20}$$

To see why the last equality holds, first note that $\inf_{H \in \mathcal{H}} f_M(H) \geq \inf_{H \in \mathcal{H}} \tilde{f}_M(H)$ because $\mathrm{Tr}(H) \geq \mathrm{Tr}(\Pi_M H \Pi_M)$. Then for the other direction, note that we can always approximate $\tilde{f}_M(H)$ using $f_M(H_\delta)$ where $H_\delta = (1-\delta)\Pi_M H \Pi_M^{\top} + \delta I_d$, for which $f_M(H_\delta) \leq \frac{1}{1-\delta}\langle M, H^{-1}\rangle + (1-\delta)\mathrm{Tr}(\Pi_M H \Pi_M) + \delta d \to \tilde{f}_M(H)$ as $\delta \to 0$. This implies that $\inf_{H \in \mathcal{H}} \tilde{f}_M(H) \geq \inf_{H \in \mathcal{H}} f_M(H)$, and thus the optimal objective values of the two optimization problems are the same. Now define $\widetilde{H}_M = P_{\mathcal{H}}(M) + (I_d - \Pi_M) \in \mathcal{H}$ whose inverse is $\widetilde{H}_M^{-1} = P_{\mathcal{H}}(M)^{\dagger} + (I_d - \Pi_M)$, then $\widetilde{H}_M \succ 0$ is a solution to the optimization problem in (20) because $\tilde{f}_M(\widetilde{H}_M) = \langle M, P_{\mathcal{H}}(M)^{\dagger}\rangle + \mathrm{Tr}(P_{\mathcal{H}}(M)) = \inf_{H \in \mathcal{H}} f_M(H)$. The gradient of $\tilde{f}_M$ is given by $\nabla \tilde{f}_M(H) = -H^{-1} M H^{-1} + \Pi_M$. Since $\mathcal{H}$ is a cone, the optimality of $\widetilde{H}_M$ implies that for any $H \in \mathcal{H}$,

$$0 = \langle \nabla f(\widetilde{H}_M), H - \widetilde{H}_M \rangle = \langle -\widetilde{H}_M^{-1} M \widetilde{H}_M^{-1} + \Pi_M, H - \widetilde{H}_M \rangle.$$

By the definition of $\widetilde{H}_M$, we have $\widetilde{H}_M^{-1} M \widetilde{H}_M^{-1} = P_{\mathcal{H}}(M)^{\dagger} M P_{\mathcal{H}}(M)^{\dagger}$ because $\mathrm{span}(M) \subseteq \mathrm{span}(P_{\mathcal{H}}(M))$. Therefore, it follows that

$$0 = \langle P_{\mathcal{H}}(M)^{\dagger} M P_{\mathcal{H}}(M)^{\dagger} - \Pi_M, H - P_{\mathcal{H}}(M) - (I_d - \Pi_M)\rangle$$

$$= \langle P_{\mathcal{H}}(M)^{\dagger} M P_{\mathcal{H}}(M)^{\dagger} - \Pi_M, H - P_{\mathcal{H}}(M)\rangle.$$

**Proof for the operator monotonicity of the extended $P_{\mathcal{H}}$.** For any $M_1 \succeq M_2 \succeq 0$, it follows from Proposition A.3 that for any $\epsilon > 0$, $P_{\mathcal{H}}(M_1 + \epsilon I_d) \succeq P_{\mathcal{H}}(M_2 + \epsilon I_d)$. Taking the limit as $\epsilon \to 0$ on both sides yields $P_{\mathcal{H}}(M_1) \succeq P_{\mathcal{H}}(M_2)$. Similarly, as $P_{\mathcal{H}}(M_1 + \epsilon I_d) - P_{\mathcal{H}}(M_2 + \epsilon I_d) \in \mathcal{H}$ for every $\epsilon > 0$ by Proposition A.3, we have $P_{\mathcal{H}}(M_1) - P_{\mathcal{H}}(M_2) = \lim_{\epsilon \to 0} P_{\mathcal{H}}(M_1 + \epsilon I_d) - P_{\mathcal{H}}(M_2 + \epsilon) \in \mathcal{H}$ because $\mathcal{H}$ is a closed set. This completes the proof. $\square$

### A.3. Adaptive gradient norm corresponds to the dual norm

*Proof of Lemma 3.3.* We begin with the case of $t = 1$. Fix any $g \in \mathbb{R}^d$. First for any $w \in \mathbb{R}^d$ with $\|w\|_{\mathcal{H}} \leq 1$, by Cauchy-Schwarz inequality, it holds for all $H \in \mathcal{H}$ with $\mathrm{Tr}(H) \leq 1$ that

$$g^{\top} w \leq \sqrt{g^{\top} H^{-1} g}\sqrt{w^{\top} H w} \leq \sqrt{\langle gg^{\top}, H^{-1}\rangle} \cdot \|w\|_{\mathcal{H}} \leq \sqrt{\langle gg^{\top}, H^{-1}\rangle}$$

where the second inequality follows from the definition of $\|w\|_{\mathcal{H}}$. Now, further taking infimum over $H \in \mathcal{H}$ with $\mathrm{Tr}(H) \leq 1$ and then taking supremum over $w \in \mathbb{R}^d$ with $\|w\|_{\mathcal{H}} \leq 1$, we obtain that

$$\|g\|_{\mathcal{H}}^* \leq \sqrt{\inf_{H \in \mathcal{H}, \mathrm{Tr}(H) \leq 1} \langle gg^{\top}, H^{-1}\rangle}. \tag{21}$$

Next, we need to show that the above inequality holds also for the other direction.

By Proposition A.4, we define $H_* = \mathrm{Tr}(P_{\mathcal{H}}(gg^{\top}))^{-1} P_{\mathcal{H}}(gg^{\top}) = \inf_{H \in \mathcal{H}, \mathrm{Tr}(H) \leq 1}\langle M, H^{-1}\rangle$. We choose correspondingly $w_* = \frac{H_*^{\dagger} g}{\|H_*^{\dagger} g\|_{\mathcal{H}}}$, which satisfies that $\|w_*\|_{\mathcal{H}} = 1$. Then

$$\|g\|_{\mathcal{H}}^* \geq g^{\top} w_* = \frac{g^{\top} H_*^{\dagger} g}{\|H_*^{\dagger} g\|_{\mathcal{H}}}. \tag{22}$$

Therefore, it suffices to show that $g^{\top} w_* \geq \sqrt{g^{\top} H_*^{\dagger} g}$, which is equivalent to

$$g^{\top} H_*^{\dagger} g \geq \|H_*^{\dagger} g\|_{\mathcal{H}}^2 = \sup_{H \in \mathcal{H}, \mathrm{Tr}(H) \leq 1} g^{\top} H_*^{\dagger} H H_*^{\dagger} g.$$

By the property (d) of $P_{\mathcal{H}}(\boldsymbol{g}\boldsymbol{g}^\top)$ from Proposition A.4, we have $\langle P_{\mathcal{H}}(\boldsymbol{g}\boldsymbol{g}^\top)^\dagger \boldsymbol{g}\boldsymbol{g}^\top P_{\mathcal{H}}(\boldsymbol{g}\boldsymbol{g}^\top)^\dagger - \boldsymbol{\Pi}_{\boldsymbol{g}\boldsymbol{g}^\top}, \boldsymbol{H} - P_{\mathcal{H}}(\boldsymbol{g}\boldsymbol{g}^\top)\rangle = 0$ for any $\boldsymbol{H} \in \mathcal{H}$. Rearranging this equality, we obtain

$$\boldsymbol{g}^\top P_{\mathcal{H}}(\boldsymbol{g}\boldsymbol{g}^\top)^\dagger \boldsymbol{H} P_{\mathcal{H}}(\boldsymbol{g}\boldsymbol{g}^\top)^\dagger \boldsymbol{g} = \boldsymbol{g}^\top P_{\mathcal{H}}(\boldsymbol{g}\boldsymbol{g}^\top)^\dagger \boldsymbol{g} + \langle \boldsymbol{\Pi}_{\boldsymbol{g}\boldsymbol{g}^\top}, \boldsymbol{H} - P_{\mathcal{H}}(\boldsymbol{g}\boldsymbol{g}^\top)\rangle$$
$$= \boldsymbol{g}^\top P_{\mathcal{H}}(\boldsymbol{g}\boldsymbol{g}^\top)^\dagger \boldsymbol{g} + \langle \boldsymbol{\Pi}_{\boldsymbol{g}\boldsymbol{g}^\top}, \boldsymbol{H}\rangle - \mathrm{Tr}(P_{\mathcal{H}}(\boldsymbol{g}\boldsymbol{g}^\top))$$

where the second equality is because $\boldsymbol{\Pi}_{\boldsymbol{g}\boldsymbol{g}^\top}$ is exactly the projection matrix onto $\mathrm{span}(P_{\mathcal{H}}(\boldsymbol{g}\boldsymbol{g}^\top))$. Applying the above equality with $\mathrm{Tr}(P_{\mathcal{H}}(\boldsymbol{g}\boldsymbol{g}^\top)) \cdot \overline{\boldsymbol{H}}$ in place of $\boldsymbol{H}$ and plugging in the definition of $\boldsymbol{H}_*$, we further have

$$\boldsymbol{g}^\top \boldsymbol{H}_*^\dagger \overline{\boldsymbol{H}} \boldsymbol{H}_*^\dagger \boldsymbol{g} = \boldsymbol{g}^\top \boldsymbol{H}_*^\dagger \boldsymbol{g} + \mathrm{Tr}(P_{\mathcal{H}}(\boldsymbol{g}\boldsymbol{g}^\top))\langle \boldsymbol{\Pi}_{\boldsymbol{g}\boldsymbol{g}^\top}, \overline{\boldsymbol{H}}\rangle - \mathrm{Tr}(P_{\mathcal{H}}(\boldsymbol{g}\boldsymbol{g}^\top))$$
$$\leq \boldsymbol{g}^\top \boldsymbol{H}_*^\dagger \boldsymbol{g} + \mathrm{Tr}(P_{\mathcal{H}}(\boldsymbol{g}\boldsymbol{g}^\top))^2 \mathrm{Tr}(\overline{\boldsymbol{H}}) - \mathrm{Tr}(P_{\mathcal{H}}(\boldsymbol{g}\boldsymbol{g}^\top))^2$$
$$= \boldsymbol{g}^\top \boldsymbol{H}_*^\dagger \boldsymbol{g}.$$

This implies that for any $\boldsymbol{H} \in \mathcal{H}$ with $\mathrm{Tr}(\boldsymbol{H}) \leq 1$, we have $\boldsymbol{g}^\top \boldsymbol{H}_*^\dagger \boldsymbol{H} \boldsymbol{H}_*^\dagger \boldsymbol{g} \leq \boldsymbol{g}^\top \boldsymbol{H}_*^\dagger \boldsymbol{g}$. Therefore, it follows from (22) that

$$\|\boldsymbol{g}\|_{\mathcal{H}}^* \geq \sqrt{\boldsymbol{g}^\top \boldsymbol{H}_*^\dagger \boldsymbol{g}} \sqrt{\frac{\boldsymbol{g}^\top \boldsymbol{H}_*^\dagger \boldsymbol{g}}{\inf_{\boldsymbol{H} \in \mathcal{H}, \mathrm{Tr}(\boldsymbol{H}) \leq 1} \boldsymbol{g}^\top \boldsymbol{H}_*^\dagger \boldsymbol{H} \boldsymbol{H}_*^\dagger \boldsymbol{g}}}$$
$$\geq \sqrt{\boldsymbol{g}^\top \boldsymbol{H}_*^\dagger \boldsymbol{g}} = \sqrt{\inf_{\boldsymbol{H} \in \mathcal{H}, \mathrm{Tr}(\boldsymbol{H}) \leq 1} \langle \boldsymbol{g}\boldsymbol{g}^\top, \boldsymbol{H}^{-1}\rangle} \tag{23}$$

where the equality follows from the definition of $\boldsymbol{H}_*$.

Finally, combining (21) and (23), we conclude that

$$\|\boldsymbol{g}\|_{\mathcal{H}}^* = \inf_{\boldsymbol{H} \in \mathcal{H}, \mathrm{Tr}(\boldsymbol{H}) \leq 1} \sqrt{\langle \boldsymbol{g}\boldsymbol{g}^\top, \boldsymbol{H}^{-1}\rangle}. \tag{24}$$

For the general case of $\boldsymbol{g}_{1:t} = (\boldsymbol{g}_1, \ldots, \boldsymbol{g}_t) \in \mathbb{R}^{d \times t}$, note that for any $\boldsymbol{H} \in \mathcal{H}$

$$\left\langle \sum_{s=1}^t \boldsymbol{g}_s \boldsymbol{g}_s^\top, \boldsymbol{H}^{-1} \right\rangle = \langle \overline{\mathrm{vec}}(\boldsymbol{g}_{1:t}) \overline{\mathrm{vec}}(\boldsymbol{g}_{1:t})^\top, \boldsymbol{H}^{-1} \otimes \boldsymbol{I}_t\rangle = \langle \overline{\mathrm{vec}}(\boldsymbol{g}_{1:t}) \overline{\mathrm{vec}}(\boldsymbol{g}_{1:t})^\top, (\boldsymbol{H} \otimes \boldsymbol{I}_t)^{-1}\rangle.$$

Then applying (24) with $\overline{\mathrm{vec}}(\boldsymbol{g}_{1:t})$ in place of $\boldsymbol{g}$ and $\mathcal{H} \otimes \boldsymbol{I}_t$ in place of $\mathcal{H}$, we obtain

$$\|\overline{\mathrm{vec}}(\boldsymbol{g}_{1:t})\|_{\mathcal{H} \otimes \boldsymbol{I}_t}^* = \inf_{\boldsymbol{H} \otimes \boldsymbol{I}_t \in \mathcal{H} \otimes \boldsymbol{I}_t, \mathrm{Tr}(\boldsymbol{H} \otimes \boldsymbol{I}_t) \leq 1} \sqrt{\langle \overline{\mathrm{vec}}(\boldsymbol{g}_{1:t}) \overline{\mathrm{vec}}(\boldsymbol{g}_{1:t})^\top, (\boldsymbol{H} \otimes \boldsymbol{I}_t)^{-1}\rangle}$$
$$= \inf_{\boldsymbol{H} \in \mathcal{H}, \mathrm{Tr}(\boldsymbol{H}) \leq 1} \sqrt{t \langle \overline{\mathrm{vec}}(\boldsymbol{g}_{1:t}) \overline{\mathrm{vec}}(\boldsymbol{g}_{1:t})^\top, (\boldsymbol{H} \otimes \boldsymbol{I}_t)^{-1}\rangle}$$
$$= \sqrt{t} \cdot \inf_{\boldsymbol{H} \in \mathcal{H}, \mathrm{Tr}(\boldsymbol{H}) \leq 1} \sqrt{\left\langle \sum_{s=1}^t \boldsymbol{g}_s \boldsymbol{g}_s^\top, \boldsymbol{H}^{-1} \right\rangle}$$

where the second equality is because $\mathrm{Tr}(\boldsymbol{H} \otimes \boldsymbol{I}_t) = \mathrm{Tr}(\boldsymbol{H}) \mathrm{Tr}(\boldsymbol{I}_t) = t \cdot \mathrm{Tr}(\boldsymbol{H})$. This completes the proof. $\qquad\square$

## A.4. Examples of ill-structured preconditioner sets

### A.4.1. PRECONDITIONER SET OF TWO-SIDED SHAMPOO

Consider $\mathcal{H} = \{\boldsymbol{H} \in \mathcal{S}_+^d : \boldsymbol{H} = \boldsymbol{U} \otimes \boldsymbol{V} \text{ for some } \boldsymbol{U} \in \mathbb{R}^{d_L \times d_L}, \boldsymbol{V} \in \mathbb{R}^{d_R \times d_R}\}$. In particular, we consider the special case of $d_L = d_R = 2$, and the following two matrices

$$\boldsymbol{G}_1(\epsilon) = \mathrm{diag}(1, \epsilon, \epsilon, \epsilon), \qquad \boldsymbol{G}_2(\epsilon) = \mathrm{diag}(1, \epsilon, \epsilon, 1)$$

where $\epsilon > 0$ is a small constant to be determined later. Clearly $G_1(\epsilon) \preceq G_2(\epsilon)$ when $\epsilon \leq 1$, but we will show that $H_{G_1(\epsilon)} \preceq H_{G_2(\epsilon)}$ does not hold for sufficiently small $\epsilon > 0$. For any $H = U \otimes V \in \mathcal{H}$, it holds that both $U$ and $V$ are PSD, and we explicitly parametrize them as

$$U = \begin{pmatrix} u_1 & u_2 \\ u_2 & u_3 \end{pmatrix}, \qquad V = \begin{pmatrix} v_1 & v_2 \\ v_2 & v_3 \end{pmatrix}$$

where $u_1, u_3 \geq 0$, $u_1 u_3 \geq u_2^2$, and similarly, $v_1, v_3 \geq 0$, $v_1 v_3 \geq v_3^2$. Correspondingly,

$$H = U \otimes V = \begin{pmatrix} u_1 v_1 & u_1 v_2 & u_2 v_1 & u_2 v_2 \\ u_1 v_2 & u_1 v_3 & u_2 v_2 & u_2 v_3 \\ u_2 v_1 & u_2 v_2 & u_3 v_1 & u_3 v_2 \\ u_2 v_2 & u_3 v_2 & u_3 v_2 & u_3 v_3 \end{pmatrix}.$$

We first analyze $H_{G_2(\epsilon)}$. For convenience, we consider

$$\begin{aligned} H_{G_2(\epsilon)}^{-1} &= \arg\min_{H \in \mathcal{H}} \langle G_2(\epsilon), H \rangle + \mathrm{Tr}(H^{-1}) \\ &= \arg\min_{U, V \in \mathcal{S}_+^2} \underbrace{\langle G_2(\epsilon), U \otimes V \rangle + \mathrm{Tr}((U \otimes V)^{-1})}_{=: f_{G_2(\epsilon)}(U, V)} \end{aligned}$$

Recall the properties of the Kronecker product that $(U \otimes V)^{-1} = U^{-1} \otimes V^{-1}$ and that $\mathrm{Tr}(U^{-1} \otimes V^{-1}) = \mathrm{Tr}(U^{-1}) \cdot \mathrm{Tr}(V^{-1})$. Then further using the definition of $G_2(\epsilon)$, we have

$$\begin{aligned} f_{G_2(\epsilon)}(U, V) &= u_1 v_1 + u_3 v_3 + \mathrm{Tr}(U^{-1}) \cdot \mathrm{Tr}(V^{-1}) + \epsilon(u_1 v_3 + u_3 v_1) \\ &= u_1 v_1 + u_3 v_3 + \frac{u_1 + u_3}{u_1 u_3 - u_2^2} \cdot \frac{v_1 + v_3}{v_1 v_3 - v_2^2} + \epsilon(u_1 v_3 + u_3 v_1) \end{aligned}$$

where we apply the explicit expression of $U^{-1}$ and $V^{-1}$. First observe that to minimize $f_{G_2(\epsilon)}(U, V)$, we must have $u_2 = v_2 = 0$ because otherwise $\frac{u_1 + u_3}{u_1 u_3 - u_2^2} > \frac{u_1 + u_3}{u_1 u_3}$ and similarly $\frac{v_1 + v_3}{v_1 v_3 - v_2^2} > \frac{v_1 + v_3}{v_1 v_3}$. Therefore, it suffices to further minimize the following:

$$f_{G_2(\epsilon)}(U, V) = \underbrace{u_1 v_1 + u_3 v_3 + \frac{u_1 + u_3}{u_1 u_3} \cdot \frac{v_1 + v_3}{v_1 v_3}}_{\tilde{f}_{G_2}(U, V)} + \epsilon(u_1 v_3 + u_3 v_1).$$

For the first term $\tilde{f}_{G_2}(U, V)$, we can apply the AM-GM inequality to get $\tilde{f}_{G_2}(U, V) \geq (4\sqrt{u_1 u_3 v_1 v_3})^{1/3}$, where the equality is achieved when $u_1 = u_3$, $v_1 = v_3$, and $u_1 v_1 = \sqrt{2}$. Moreover, for sufficiently small $\epsilon > 0$, the contribution of the second term $\epsilon(u_1 v_3 + u_3 v_1)$ is negligible, and thus we conclude that $\lim_{\epsilon \to 0} H_{G_2(\epsilon)} = \frac{\sqrt{2}}{2} I_4$.

Similarly, for $G_1(\epsilon)$, we have

$$\begin{aligned} f_{G_1(\epsilon)}(U, V) &= u_1 v_1 + \frac{u_1 + u_3}{u_1 u_3 - u_2^2} \cdot \frac{v_1 + v_3}{v_1 v_3 - v_2^2} + \epsilon(u_1 v_3 + u_3 v_1 + u_3 v_3) \\ &\geq u_1 v_1 + \frac{u_1 + u_3}{u_1 u_3} \cdot \frac{v_1 + v_3}{v_1 v_3} + \epsilon(u_1 v_3 + u_3 v_1 + u_3 v_3) \\ &= \underbrace{u_1 v_1 + \frac{1}{u_1 v_1}}_{\tilde{f}_{G_1}(U, V)} + \frac{1}{u_1 v_3} + \frac{1}{u_3 v_1} + \frac{1}{u_3 v_3} + \epsilon(u_1 v_3 + u_3 v_1 + u_3 v_3) \end{aligned}$$

where the equality holds when $u_2 = v_2 = 0$. The first term $\tilde{f}_{G_1}(U, V)$ attains the minimum value when $u_1 v_1 = 1$, and the remainder can be made small by choosing large $u_3, v_3$ when $\epsilon$ is sufficiently small. Therefore, we conclude that $\lim_{\epsilon \to 0} H_{G_1(\epsilon)} = \mathrm{diag}(1, 0, 0, 0)$.

Now comparing the limits of $H_{G_1}(\epsilon)$ and $H_{G_2}(\epsilon)$ as $\epsilon \to 0$, we can see that for sufficiently small $\epsilon > 0$, it holds that $H_{G_1(\epsilon)}[1, 1] > H_{G_2(\epsilon)}[1, 1]$. Hence, for sufficiently small $\epsilon > 0$, $H_{G_1(\epsilon)} \preceq H_{G_2(\epsilon)}$ does not hold.

A.4.2. TRIDIAGONAL MATRICES

Consider $\mathcal{H} = \{\boldsymbol{H} \in \mathcal{S}_+^d : \boldsymbol{H} \text{ is tridiagonal}\}$. Here, for $\boldsymbol{H}$ to be tridiagonal, it has nonzero elements on the main diagonal, the first diagonal above the main diagonal, and the first diagonal below the main diagonal. We consider the specific example of 3-by-3 tridiagonal matrices, and examine $P_{\mathcal{H}}(\cdot)$ for the following two matrices

$$\boldsymbol{M} = \begin{pmatrix} 2 & 1 & 1 \\ 1 & 2 & 1 \\ 1 & 1 & 2 \end{pmatrix}, \quad \boldsymbol{M}' = \begin{pmatrix} 10000 & 1 & 1 \\ 1 & 2 & 1 \\ 1 & 1 & 2 \end{pmatrix}.$$

It is clear that $0 \prec \boldsymbol{M} \preceq \boldsymbol{M}'$. We numerically solve for $P_{\mathcal{H}}(\boldsymbol{M})$ and $P_{\mathcal{H}}(\boldsymbol{M}')$ to get

$$P_{\mathcal{H}}(\boldsymbol{M}) \approx \begin{pmatrix} 1.382548 & 0.297594 & 0 \\ 0.297594 & 1.318491 & 0.297594 \\ 0 & 0.297594 & 1.382548 \end{pmatrix}, \quad P_{\mathcal{H}}(\boldsymbol{M}') \approx \begin{pmatrix} 100.000004 & 0.007229 & 0 \\ 0.007229 & 1.365999 & 0.366002 \\ 0 & 0.366002 & 1.366032 \end{pmatrix}.$$

Note that $P_{\mathcal{H}}(\boldsymbol{M}) \preceq P_{\mathcal{H}}(\boldsymbol{M}')$ does not hold because the last diagonal entry of $P_{\mathcal{H}}(\boldsymbol{M})$ is larger than that of $P_{\mathcal{H}}(\boldsymbol{M}')$.

# B. Calculations for Examples of Well-Structured Preconditioner Sets

As mentioned in Section 3.3, we provide calculations for how to derive each specific algorithm from Algorithm 1 with specific choice of $\mathcal{H}$ and explain each entry in Table 1. Recall from Algorithm 1 that

$$\boldsymbol{M}_t = \epsilon \boldsymbol{I}_d + \sum_{s=1}^{t} \boldsymbol{g}_s \boldsymbol{g}_s^{\top}.$$

## B.1. AdaGrad-Norm

For AdaGrad-Norm, we have $\mathcal{H} = \{c \cdot \boldsymbol{I}_d \mid c \geq 0\}$.

**Calculation for $\boldsymbol{H}_t$.** Then for any $\boldsymbol{H} = c \cdot \boldsymbol{I}_d \in \mathcal{H}$,

$$\langle \boldsymbol{M}_t, \boldsymbol{H}^{-1} \rangle + \eta^2 \operatorname{Tr}(\boldsymbol{H}) = \frac{1}{c} \operatorname{Tr}(\boldsymbol{M}_t) + \eta^2 cd \geq 2\eta \sqrt{d \cdot \operatorname{Tr}(\boldsymbol{M}_t)}$$

where the equality is achieved by choosing

$$c = \frac{1}{\eta} \sqrt{\frac{1}{d} \operatorname{Tr}(\boldsymbol{M}_t)} = \frac{1}{\eta} \sqrt{\epsilon + \frac{1}{d} \sum_{s=1}^{t} \|\boldsymbol{g}_s\|_2^2}.$$

This corresponds exactly to the update rule of AdaGrad-Norm, which adjusts the global learning rate based on the accumulated $\ell_2$ norm of past gradients.

**Calculation for $\|\cdot\|_{\mathcal{H}}$.** For the associated $\|\cdot\|_{\mathcal{H}}$, we have

$$\|\boldsymbol{x}\|_{\mathcal{H}} = \sup_{0 \leq c \leq 1/d} \sqrt{c \cdot \boldsymbol{x}^{\top} \boldsymbol{I}_d \boldsymbol{x}} = \frac{\|\boldsymbol{x}\|_2}{\sqrt{d}}.$$

**Calculation for $\|\boldsymbol{g}_{1:t}\|_{\mathcal{H}}$.** We can calculate the adaptive gradient norm similarly:

$$\|\boldsymbol{g}_{1:t}\|_{\mathcal{H}} = \inf_{0 \leq c \leq 1/d} \sqrt{\left\langle \sum_{s=1}^{t} \boldsymbol{g}_s \boldsymbol{g}_s^{\top}, (c\boldsymbol{I}_d)^{-1} \right\rangle} = \sqrt{d} \sqrt{\operatorname{Tr}\left(\sum_{s=1}^{t} \boldsymbol{g}_s \boldsymbol{g}_s^{\top}\right)} = \sqrt{d} \sqrt{\sum_{s=1}^{t} \|\boldsymbol{g}_s\|_2^2}.$$

## B.2. Diagonal AdaGrad

For diagonal AdaGrad, $\mathcal{H} = \mathcal{D}_+^d = \{\operatorname{diag}(c_1, \ldots, c_d) \mid c_1, \ldots, c_d \geq 0\}$.

**Calculation for $H_t$.** For any $H = \text{diag}(c_1, \ldots, c_d) \in \mathcal{H}$, we have

$$\left\langle M_t, H^{-1} \right\rangle + \eta^2 \text{Tr}(H) = \sum_{i=1}^{d} \left( \frac{M_{t,ii}}{c_i} + \eta^2 c_i \right) \geq \sum_{i=1}^{d} 2\eta \sqrt{M_{t,ii}}$$

where the equality is achieved by choosing

$$c_i = \frac{1}{\eta}\sqrt{M_{t,ii}} = \frac{1}{\eta}\sqrt{\epsilon + \sum_{s=1}^{t} g_{s,i}^2}, \quad \text{for } i = 1, \ldots, d.$$

This corresponds to the update rule diagonal AdaGrad, which computes the historical sum of squared gradients for each individual coordinate.

**Calculation for $\|\cdot\|_{\mathcal{H}}$.** For the associated $\|\cdot\|_{\mathcal{H}}$, we have

$$\|x\|_{\mathcal{H}} = \sup_{\sum_{i=1}^{d} c_i \leq 1} \sqrt{x^\top \text{diag}(c_1, \ldots, c_d) x} = \sup_{\sum_{i=1}^{d} c_i \leq 1} \sqrt{\sum_{i=1}^{d} c_i x_i^2} = \max_{i \in [d]} |x_i| = \|x\|_\infty$$

**Calculation for $\||g_{1:t}\||_{\mathcal{H}}$.** We can calculate the adaptive gradient norm similarly:

$$\||g_{1:t}\||_{\mathcal{H}} = \inf_{\sum_{i=1}^{d} c_i \leq 1} \sqrt{\left\langle \sum_{s=1}^{t} g_s g_s^\top, \text{diag}(c_1 \ldots, c_d)^{-1} \right\rangle}$$

$$= \inf_{\sum_{i=1}^{d} c_i \leq 1} \sqrt{\sum_{i=1}^{d} \frac{1}{c_i} \sum_{s=1}^{t} g_{s,i}^2}$$

$$= \sum_{i=1}^{d} \sqrt{\sum_{s=1}^{t} g_{s,i}^2}$$

where the last equality is because of the Cauchy inequality: for $c_1, \ldots, c_d \geq 0$ such that $\sum_{i=1}^{d} c_i \leq 1$,

$$\sqrt{\sum_{i=1}^{d} \frac{1}{c_i} \sum_{s=1}^{t} g_{s,i}^2} \geq \sqrt{\sum_{i=1}^{d} \frac{1}{c_i} \sum_{s=1}^{t} g_{s,i}^2} \sqrt{\sum_{i=1}^{d} c_i} \geq \sum_{i=1}^{d} \sqrt{\sum_{s=1}^{t} g_{s,i}^2}.$$

### B.3. Full-matrix AdaGrad

For full-matrix AdaGrad, $\mathcal{H} = \mathcal{S}_+^d$.

**Calculation for $H_t$.** Note that $\langle M_t, H^{-1} \rangle + \eta^2 \text{Tr}(H)$ is a convex function of $H \succ 0$, so we can get $H_t$ by first calculating the gradient of the objective function and then setting it to zero. Specifically, for any $H \succ 0$, we have

$$\nabla_H \left( \langle M_t, H^{-1} \rangle + \eta^2 \text{Tr}(H) \right) = -H^{-1} M_t H^{-1} + \eta^2 I_d.$$

Setting it to zero yields

$$H_t = \frac{1}{\eta} M_t^{\frac{1}{2}} = \frac{1}{\eta} \left( \epsilon I_d + \sum_{s=1}^{t} g_s g_s^\top \right)^{\frac{1}{2}}.$$

This corresponds to the update rule of full-matrix AdaGrad.

**Calculation for $\|\cdot\|_{\mathcal{H}}$.** For the associated $\|\cdot\|_{\mathcal{H}}$, we have

$$\|\boldsymbol{x}\|_{\mathcal{H}} = \sup_{\boldsymbol{H} \succeq 0, \mathrm{Tr}(\boldsymbol{H}) \leq 1} \sqrt{\boldsymbol{x}^\top \boldsymbol{H} \boldsymbol{x}}$$

$$= \sup_{\boldsymbol{H} \succeq 0, \mathrm{Tr}(\boldsymbol{H}) \leq 1} \sqrt{\langle \boldsymbol{x}\boldsymbol{x}^\top, \boldsymbol{H} \rangle} = \sqrt{\lambda_1(\boldsymbol{x}\boldsymbol{x}^\top)} = \|\boldsymbol{x}\|_2 \,.$$

**Calculation for $\|\!|\boldsymbol{g}_{1:t}\|\!|_{\mathcal{H}}$.** We can adapt the calculation for $\boldsymbol{H}_t$ to the constrained optimization problem in the definition of $\|\!|\boldsymbol{g}_{1:t}\|\!|_{\mathcal{H}}$ to get

$$\|\!|\boldsymbol{g}_{1:t}\|\!|_{\mathcal{H}} = \inf_{\boldsymbol{H} \in \mathcal{H}, \mathrm{Tr}(\boldsymbol{H}) \leq 1} \sqrt{\left\langle \sum_{s=1}^{t} \boldsymbol{g}_s \boldsymbol{g}_s^\top, \boldsymbol{H}^{-1} \right\rangle}$$

$$= \sqrt{\left\langle \sum_{s=1}^{t} \boldsymbol{g}_s \boldsymbol{g}_s^\top, \left( \frac{1}{\mathrm{Tr}[(\sum_{s=1}^{t} \boldsymbol{g}_s \boldsymbol{g}_s^\top)^{\frac{1}{2}}]} \left( \sum_{s=1}^{t} \boldsymbol{g}_s \boldsymbol{g}_s^\top \right)^{\frac{1}{2}} \right)^{-1} \right\rangle}$$

$$= \mathrm{Tr}\left[ \left( \sum_{s=1}^{t} \boldsymbol{g}_s \boldsymbol{g}_s^\top \right)^{\frac{1}{2}} \right].$$

## B.4. One-sided Shampoo

For one-sided Shampoo, $\mathcal{H} = \mathcal{S}_+^{d_L} \otimes \boldsymbol{I}_{d_R}$.

**Calculation for $\boldsymbol{H}_t$.** For any $\boldsymbol{H} = \boldsymbol{H}_L \otimes \boldsymbol{I}_{d_R} \in \mathcal{H}$, $\boldsymbol{H}^{-1} = \boldsymbol{H}_L^{-1} \otimes \boldsymbol{I}_{d_R}$, so we have $\mathrm{Tr}(\boldsymbol{H}) = d_R \mathrm{Tr}(\boldsymbol{H}_L)$ and

$$\langle \boldsymbol{M}_t, \boldsymbol{H}^{-1} \rangle = \langle \boldsymbol{M}_t, (\boldsymbol{H}_L \otimes \boldsymbol{I}_{d_R})^{-1} \rangle$$

$$= \langle \epsilon \boldsymbol{I}_d, \boldsymbol{H}_L^{-1} \otimes \boldsymbol{I}_{d_R} \rangle + \sum_{s=1}^{t} \langle \overline{\mathrm{vec}}(\boldsymbol{G}_s) \overline{\mathrm{vec}}(\boldsymbol{G}_s)^\top, \boldsymbol{H}_L^{-1} \otimes \boldsymbol{I}_{d_R} \rangle$$

$$= \epsilon d_R \mathrm{Tr}(\boldsymbol{H}_L^{-1}) + \sum_{s=1}^{t} \mathrm{Tr}(\boldsymbol{G}_s^\top \boldsymbol{H}_L^{-1} \boldsymbol{G}_s)$$

$$= \mathrm{Tr}\left[ \left( \epsilon d_R \boldsymbol{I}_{d_L} + \sum_{s=1}^{t} \boldsymbol{G}_s \boldsymbol{G}_s^\top \right) \boldsymbol{H}_L^{-1} \right]$$

where we use the fact that $\langle \overline{\mathrm{vec}}(\boldsymbol{X}) \overline{\mathrm{vec}}(\boldsymbol{X})^\top, \boldsymbol{H}_L \otimes \boldsymbol{I}_{d_R} \rangle = \langle \boldsymbol{X}\boldsymbol{X}^\top, \boldsymbol{H}_L \rangle$ for any $\boldsymbol{X} \in \mathbb{R}^{d_L \times d_R}$. Again, since the objective function is convex in $\boldsymbol{H}_L \succ 0$, we can derive $\boldsymbol{H}_L$ by first calculating the gradient and then setting it to zero. Taking derivative with respect to $\boldsymbol{H}_L$, we obtain

$$\nabla_{\boldsymbol{H}_L} \left( \langle \boldsymbol{M}_t, \boldsymbol{H}^{-1} \rangle + \eta^2 \mathrm{Tr}(\boldsymbol{H}) \right) = -\boldsymbol{H}_L^{-1} \left( \epsilon d_R \boldsymbol{I}_{d_L} + \sum_{s=1}^{t} \boldsymbol{G}_s \boldsymbol{G}_s^\top \right) \boldsymbol{H}_L^{-1} + \eta^2 d_R \boldsymbol{I}_{d_L} \,.$$

Setting it to 0, we obtain

$$\underset{\boldsymbol{H}_L \in \mathcal{S}_+^{d_L}}{\arg\min} \langle \boldsymbol{M}_t, (\boldsymbol{H}_L \otimes \boldsymbol{I}_{d_R})^{-1} \rangle + \eta^2 d_R \mathrm{Tr}(\boldsymbol{H}_L \otimes \boldsymbol{I}_{d_R}) = \frac{1}{\eta \sqrt{d_R}} \left( \epsilon d_R \boldsymbol{I}_{d_L} + \sum_{s=1}^{t} \boldsymbol{G}_s \boldsymbol{G}_s^\top \right)^{\frac{1}{2}}$$

$$= \frac{1}{\eta} \left( \epsilon \boldsymbol{I}_{d_L} + \frac{1}{d_R} \sum_{s=1}^{t} \boldsymbol{G}_s \boldsymbol{G}_s^\top \right)^{\frac{1}{2}}.$$

This is exactly the $\boldsymbol{L}_t$ in Algorithm 2. Therefore, the preconditioner $\boldsymbol{H}_t$ in one-sided Shampoo is given by

$$\boldsymbol{H}_t = \frac{1}{\eta} \left( \epsilon \boldsymbol{I}_{d_L} + \frac{1}{d_R} \sum_{s=1}^{t} \boldsymbol{G}_s \boldsymbol{G}_s^\top \right)^{\frac{1}{2}} \otimes \boldsymbol{I}_{d_R} \,.$$

As a result, Algorithm 1 with $\mathcal{H} = \mathcal{S}_+^{d_L} \otimes \boldsymbol{I}_{d_R}$ recovers Algorithm 2.

**Calculation for** $\|\cdot\|_{\mathcal{H}}$**.** For the associated $\|\cdot\|_{\mathcal{H}}$, we have

$$
\|\boldsymbol{x}\|_{\mathcal{H}} = \sup_{\boldsymbol{H}=\boldsymbol{H}_L\otimes\boldsymbol{I}_{d_R},\boldsymbol{H}_L\succeq0,\mathrm{Tr}(\boldsymbol{H})\leq1} \sqrt{\boldsymbol{x}^\top\boldsymbol{H}\boldsymbol{x}}
$$

$$
= \sup_{\boldsymbol{H}_L\succeq0,\mathrm{Tr}(\boldsymbol{H}_L)\leq1/d_R} \sqrt{\langle\overline{\mathrm{vec}}\,(\boldsymbol{X})\,\overline{\mathrm{vec}}\,(\boldsymbol{X})^\top,\boldsymbol{H}_L\otimes\boldsymbol{I}_{d_R}\rangle}
$$

$$
= \sup_{\boldsymbol{H}_L\succeq0,\mathrm{Tr}(\boldsymbol{H}_L)\leq1/d_R} \sqrt{\langle\boldsymbol{X}\boldsymbol{X}^\top,\boldsymbol{H}_L\rangle}
$$

$$
= \frac{1}{\sqrt{d_R}}\,\|\boldsymbol{X}\|_{\mathrm{op}}
$$

where the last equality is achieved at $\boldsymbol{H}_L = \boldsymbol{u}\boldsymbol{u}^\top/d_R$ for $\boldsymbol{u}$ being the leading eigenvector of $\boldsymbol{X}\boldsymbol{X}^\top$. The derivation above provides a proof for Lemma 4.1.

**Calculation for** $\|\|\boldsymbol{g}_{1:t}\|\|_{\mathcal{H}}$**.** Again, for the adaptive gradient norm, we can adapt the calculation for $\boldsymbol{H}_t$ to the constrained optimization problem in the definition of $\|\|\boldsymbol{g}_{1:t}\|\|_{\mathcal{H}}$ to get

$$
\|\|\boldsymbol{g}_{1:t}\|\|_{\mathcal{H}} = \inf_{\boldsymbol{H}\in\mathcal{H},\mathrm{Tr}(\boldsymbol{H})\leq1} \sqrt{\left\langle\sum_{s=1}^{t}\boldsymbol{g}_s\boldsymbol{g}_s^\top,\boldsymbol{H}^{-1}\right\rangle}
$$

$$
= \inf_{\boldsymbol{H}_L\in\mathcal{S}_+^{d_L},\mathrm{Tr}(\boldsymbol{H}_L)\leq\frac{1}{d_R}} \sqrt{\left\langle\sum_{s=1}^{t}\boldsymbol{G}_s\boldsymbol{G}_s^\top,\boldsymbol{H}_L^{-1}\right\rangle}
$$

$$
= \sqrt{\left\langle\sum_{s=1}^{t}\boldsymbol{G}_s\boldsymbol{G}_s^\top,\left[\frac{1}{d_R\,\mathrm{Tr}[(\sum_{s=1}^{t}\boldsymbol{G}_s\boldsymbol{G}_s^\top)^{\frac{1}{2}}]}\left(\sum_{s=1}^{t}\boldsymbol{G}_s\boldsymbol{G}_s^\top\right)^{\frac{1}{2}}\right]^{-1}\right\rangle}
$$

$$
= \sqrt{d_R}\,\mathrm{Tr}\left[\left(\sum_{s=1}^{t}\boldsymbol{G}_s\boldsymbol{G}_s^\top\right)^{\frac{1}{2}}\right] = \mathrm{Tr}\left[\left(d_R\sum_{s=1}^{t}\boldsymbol{G}_s\boldsymbol{G}_s^\top\right)^{\frac{1}{2}}\right].
$$

## C. Analysis for EMA adaptive optimization

As mentioned in Section 3, our analysis can be generalized to Algorithm 4, which is an EMA variant of Algorithm 1. To simplify the theoretical analysis, we instead focus on Algorithm 5 in which the current gradient is not scaled by $1-\beta_2$ when computing $\boldsymbol{M}_t$. It serves as a bridge between Algorithms 1 and 4. When $\beta_2=1$, Algorithm 5 exactly recovers Algorithm 1. When the learning rate $\tilde{\eta}$ is rescaled by $\sqrt{1-\beta_2}$ and $\epsilon$ is rescaled by $1/(1-\beta_2)$, Algorithm 5 is equivalent to Algorithm 4. Theorem C.1 extends Theorem 3.4 to get online regret bound for Algorithm 5. The proof is in Appendix D.3.

**Theorem C.1.** *Let $\mathcal{H}$ be a well-structured preconditioner set under Definition 3.1. Then for any convex loss functions $\tilde{L}_1,\ldots,\tilde{L}_T$, the regret of Algorithm 5 compared to any $\boldsymbol{x}^*\in\mathcal{X}$ can be bounded as*

$$
\sum_{t=1}^{T}\beta_2^{\frac{T-t}{2}}\left[\tilde{L}_t(\boldsymbol{x}_t)-\tilde{L}_t(\boldsymbol{x}^*)\right] \leq \left(\frac{D^2}{2\tilde{\eta}}+\tilde{\eta}\right)\inf_{\boldsymbol{H}\in\mathcal{H},\mathrm{Tr}(\boldsymbol{H})\leq1}\sqrt{\left\langle\tilde{\boldsymbol{M}}_T,\boldsymbol{H}^{-1}\right\rangle}
$$

*where $D = \max_{t\in[T]}\|\boldsymbol{x}_t-\boldsymbol{x}^*\|_{\mathcal{H}}$.*

Theorem C.2 provides the convergence rate for Algorithm 5, which generalizes Theorem 3.8. The proof is in Appendix D.3.

**Theorem C.2.** *Let $\mathcal{H}$ be a well-structured preconditioner set under Definition 3.1. Consider any independent stochastic convex loss functions $\tilde{L}_1,\ldots,\tilde{L}_T$ satisfying Assumption 3.7, and let $H(\tilde{L},\mathcal{H})$ be the $\mathcal{H}$-smoothness of their expectation $\tilde{L}$. Suppose the global minimizer of $L$, denoted by $\boldsymbol{x}^*$, is in $\mathcal{X}$. Then for the iterates $\boldsymbol{x}_1,\ldots,\boldsymbol{x}_T$ of Algorithm 5, denoting $\bar{\boldsymbol{x}}_{1:T} = (\sum_{t=1}^{T}\beta_2^{\frac{T-t}{2}})^{-1}\sum_{t=1}^{T}\beta_2^{\frac{T-t}{2}}\boldsymbol{x}_t$, it holds that*

$$
\mathbb{E}\tilde{L}(\bar{\boldsymbol{x}}_{1:T})-\tilde{L}_t(\boldsymbol{x}^*) \leq \frac{16}{\sum_{t=1}^{T}\beta_2^{\frac{T-t}{2}}}\|\mathcal{X}\|_{\mathcal{H}}^2 H(\tilde{L},\mathcal{H}) + \frac{4\sqrt{2}}{\sqrt{\sum_{t=1}^{T}\beta_2^{T-t}}}\|\mathcal{X}\|_{\mathcal{H}}\sigma + \frac{4\sqrt{2}}{\sum_{t=1}^{T}\beta_2^{\frac{-t}{2}}}\|\mathcal{X}\|_{\mathcal{H}}d\sqrt{\epsilon}
$$

**Algorithm 4** EMA Adaptive Regularization Meta-Algorithm

---

> **Hyperparam:** $\epsilon > 0$, convex set $\mathcal{X} \subseteq \mathbb{R}^d$, learning rate $\eta$, preconditioners $\mathcal{H} \subset \mathcal{S}_+^d$, $\beta_2 \in (0, 1)$
> **Input:** initialization $\boldsymbol{x}_1$, loss functions $\{L_t\}_{t=1}^T : \mathbb{R}^d \to \mathbb{R}$
> $\boldsymbol{M}_0 \leftarrow \epsilon \boldsymbol{I}_d$
> **for** $t = 1, 2, \ldots, T$ **do**
> $\quad \boldsymbol{g}_t \leftarrow \nabla L_t(\boldsymbol{x}_t)$
> $\quad \boldsymbol{M}_t \leftarrow \beta_2 \boldsymbol{M}_{t-1} + (1 - \beta_2) \boldsymbol{g}_t \boldsymbol{g}_t^\top$
> $\quad \boldsymbol{H}_t \leftarrow \arg\min_{\boldsymbol{H} \in \mathcal{H}} \langle \boldsymbol{M}_t, \boldsymbol{H}^{-1} \rangle + \eta^2 \operatorname{Tr}(\boldsymbol{H})$
> $\quad \boldsymbol{x}_{t+1} \leftarrow \Pi_{\mathcal{X}}^{\boldsymbol{H}_t} \left( \boldsymbol{x}_t - \boldsymbol{H}_t^{-1} \boldsymbol{g}_t \right)$
> **Return** $\boldsymbol{x}_1, \ldots, \boldsymbol{x}_T$

**Algorithm 5** Weighted Adaptive Regularization Meta-Algorithm

---

> **Hyperparam:** $\epsilon > 0$, convex set $\mathcal{X} \subseteq \mathbb{R}^d$, learning rate $\tilde{\eta}$, preconditioners $\mathcal{H} \subset \mathcal{S}_+^d$, $\beta_2 \in (0, 1)$
> **Input:** initialization $\boldsymbol{x}_1$, loss functions $\{\tilde{L}_t\}_{t=1}^T : \mathbb{R}^d \to \mathbb{R}$
> $\tilde{\boldsymbol{M}}_0 \leftarrow \epsilon \boldsymbol{I}_d$
> **for** $t = 1, 2, \ldots, T$ **do**
> $\quad \tilde{\boldsymbol{g}}_t \leftarrow \nabla \tilde{L}_t(\tilde{\boldsymbol{x}}_t)$
> $\quad \tilde{\boldsymbol{M}}_t \leftarrow \beta_2 \tilde{\boldsymbol{M}}_{t-1} + \tilde{\boldsymbol{g}}_t \tilde{\boldsymbol{g}}_t^\top$
> $\quad \tilde{\boldsymbol{H}}_t \leftarrow \arg\min_{\tilde{\boldsymbol{H}} \in \mathcal{H}} \langle \tilde{\boldsymbol{M}}_t, \tilde{\boldsymbol{H}}^{-1} \rangle + \tilde{\eta}^2 \operatorname{Tr}(\tilde{\boldsymbol{H}})$
> $\quad \tilde{\boldsymbol{x}}_{t+1} \leftarrow \Pi_{\mathcal{X}}^{\tilde{\boldsymbol{H}}_t} \left( \tilde{\boldsymbol{x}}_t - \tilde{\boldsymbol{H}}_t^{-1} \tilde{\boldsymbol{g}}_t \right)$
> **Return** $\tilde{\boldsymbol{x}}_1, \ldots, \tilde{\boldsymbol{x}}_T$

*where* $\sigma = \inf_{\boldsymbol{H} \in \mathcal{H}, \operatorname{Tr}(\boldsymbol{H}) \leq 1} \sqrt{\langle \boldsymbol{\Sigma}, \boldsymbol{H}^{-1} \rangle}$.

When $\beta_2 = 1$, Theorem C.2 recovers the result in Theorem 3.8 and provides a $O(T^{-\frac{1}{2}})$ convergence rate. When $\beta_2 < 1$, the optimality gap is upper bounded by $O\big((1 - \beta_2) \|\mathcal{X}\|_{\mathcal{H}}^2 H(\tilde{L}, \mathcal{H}) + \sqrt{1 - \beta_2} \|\mathcal{X}\|_{\mathcal{H}} \sigma\big)$ when $T = \Theta\left(\frac{1}{1 - \beta_2}\right)$.

## D. Proof for the Unified Analysis

### D.1. Regret bound

We first present the proof for the main result on the regret bound for Algorithm 1 with a well-structured preconditioner set $\mathcal{H}$.

**Theorem 3.4.** *Let $\mathcal{H}$ be a well-structured preconditioner set under Definition 3.1. Then for any convex loss functions $L_1, \ldots, L_T$, the regret of Algorithm 1 compared to any $\boldsymbol{x}^* \in \mathcal{X}$ can be bounded as*

$$\sum_{t=1}^T L_t(\boldsymbol{x}_t) - \sum_{t=1}^T L_t(\boldsymbol{x}^*) \leq \left( \frac{D^2}{2\eta} + \eta \right) \left( G + d\sqrt{\epsilon} \right)$$

*where* $G = \|\boldsymbol{g}_{1:T}\|_{\mathcal{H}}$, $D = \max_{t \in [T]} \|\boldsymbol{x}_t - \boldsymbol{x}^*\|_{\mathcal{H}}$.

*Proof of Theorem 3.4.* First we will analyze the property of each $\boldsymbol{H}_t$. Recall from Proposition 3.2 that $\boldsymbol{H}_t$ satisfies $\langle \boldsymbol{M}_t, \boldsymbol{H}_t^{-1} \rangle = \eta^2 \operatorname{Tr}(\boldsymbol{H}_t)$ and that $\overline{\boldsymbol{H}}_t := \operatorname{Tr}(\boldsymbol{H}_t)^{-1} \boldsymbol{H}_t = \arg\min_{\boldsymbol{H} \in \mathcal{H}, \operatorname{Tr}(\boldsymbol{H}) \leq 1} \langle \boldsymbol{M}_t, \boldsymbol{H}^{-1} \rangle$. Therefore,

$$\langle \boldsymbol{M}_t, \boldsymbol{H}_t^{-1} \rangle = \eta^2 \operatorname{Tr}(\boldsymbol{H}_t) = \eta \sqrt{\langle \boldsymbol{M}_t, \boldsymbol{H}_t^{-1} \rangle \operatorname{Tr}(\boldsymbol{H}_t)} = \eta \sqrt{\langle \boldsymbol{M}_t, \overline{\boldsymbol{H}}_t^{-1} \rangle}. \tag{25}$$

Now recall the regret bound from Theorem 2.1:

$$\sum_{t=1}^T L_t(\boldsymbol{x}_t) - \sum_{t=1}^T L_t(\boldsymbol{x}^*) \leq \frac{1}{2} \left( \langle \boldsymbol{M}_T, \boldsymbol{H}_T^{-1} \rangle + \eta^2 \operatorname{Tr}(\boldsymbol{H}_T) - \eta^2 \operatorname{Tr}(\boldsymbol{H}_0) \right)$$

$$+ \frac{1}{2} \sum_{t=1}^T \left( \|\boldsymbol{x}_t - \boldsymbol{x}^*\|_{\boldsymbol{H}_t}^2 - \|\boldsymbol{x}_{t+1} - \boldsymbol{x}^*\|_{\boldsymbol{H}_t}^2 \right)$$

$$= \frac{1}{2} \left( 2\eta \sqrt{\langle \boldsymbol{M}_T, \overline{\boldsymbol{H}}_T^{-1} \rangle} - \eta^2 \operatorname{Tr}(\boldsymbol{H}_0) \right)$$

$$+ \frac{1}{2} \sum_{t=1}^T \left( \|\boldsymbol{x}_t - \boldsymbol{x}^*\|_{\boldsymbol{H}_t}^2 - \|\boldsymbol{x}_{t+1} - \boldsymbol{x}^*\|_{\boldsymbol{H}_t}^2 \right) \tag{26}$$

where the equality follows from the facts in (25). Next, for the second term on the right-hand side of (26), we rearrange the summation to obtain

$$\sum_{t=1}^{T} \left( \|\boldsymbol{x}_t - \boldsymbol{x}^*\|_{\boldsymbol{H}_t}^2 - \|\boldsymbol{x}_{t+1} - \boldsymbol{x}^*\|_{\boldsymbol{H}_t}^2 \right) \le \|\boldsymbol{x}_1 - \boldsymbol{x}^*\|_{\boldsymbol{H}_1}^2 + \sum_{t=2}^{T} \left( \|\boldsymbol{x}_t - \boldsymbol{x}^*\|_{\boldsymbol{H}_t}^2 - \|\boldsymbol{x}_t - \boldsymbol{x}^*\|_{\boldsymbol{H}_{t-1}}^2 \right)$$

$$= \|\boldsymbol{x}_1 - \boldsymbol{x}^*\|_{\boldsymbol{H}_1}^2 + \sum_{t=2}^{T} \|\boldsymbol{x}_t - \boldsymbol{x}^*\|_{\boldsymbol{H}_t - \boldsymbol{H}_{t-1}}^2 .$$

Notice that $\boldsymbol{M}_t - \boldsymbol{M}_{t-1} = \boldsymbol{g}_t \boldsymbol{g}_t^\top \succeq 0$, and thus $\boldsymbol{H}_t - \boldsymbol{H}_{t-1} \in \mathcal{H}$ by Proposition 3.2. This implies

$$\|\boldsymbol{x}_t - \boldsymbol{x}^*\|_{\boldsymbol{H}_t - \boldsymbol{H}_{t-1}}^2 = (\boldsymbol{x}_t - \boldsymbol{x}^*)^\top (\boldsymbol{H}_t - \boldsymbol{H}_{t-1})(\boldsymbol{x}_t - \boldsymbol{x}^*) \le \operatorname{Tr}(\boldsymbol{H}_t - \boldsymbol{H}_{t-1}) \|\boldsymbol{x}_t - \boldsymbol{x}^*\|_{\mathcal{H}}^2$$

where the inequality follows from the definition of $\|\cdot\|_{\mathcal{H}}$ in (1). It then follows that

$$\sum_{t=1}^{T} \left( \|\boldsymbol{x}_t - \boldsymbol{x}^*\|_{\boldsymbol{H}_t}^2 - \|\boldsymbol{x}_{t+1} - \boldsymbol{x}^*\|_{\boldsymbol{H}_t}^2 \right) \le \operatorname{Tr}(\boldsymbol{H}_1) \|\boldsymbol{x}_1 - \boldsymbol{x}^*\|_{\mathcal{H}}^2 + \sum_{t=2}^{T} \operatorname{Tr}(\boldsymbol{H}_t - \boldsymbol{H}_{t-1}) \|\boldsymbol{x}_t - \boldsymbol{x}^*\|_{\mathcal{H}}^2$$

$$\le \operatorname{Tr}(\boldsymbol{H}_T) \max_{1 \le t \le T} \|\boldsymbol{x}_t - \boldsymbol{x}^*\|_{\mathcal{H}}^2$$

$$= \frac{1}{\eta} \sqrt{\langle \boldsymbol{M}_T, \overline{\boldsymbol{H}}_T^{-1} \rangle} \max_{1 \le t \le T} \|\boldsymbol{x}_t - \boldsymbol{x}^*\|_{\mathcal{H}}^2$$

where the equality again follows from (25). Plugging this back into (26), we obtain

$$\sum_{t=1}^{T} L_t(\boldsymbol{x}_t) - \sum_{t=1}^{T} L_t(\boldsymbol{x}^*) \le \eta \sqrt{\langle \boldsymbol{M}_T, \overline{\boldsymbol{H}}_T^{-1} \rangle} + \frac{1}{2\eta} \sqrt{\langle \boldsymbol{M}_T, \overline{\boldsymbol{H}}_T^{-1} \rangle} \max_{1 \le t \le T} \|\boldsymbol{x}_t - \boldsymbol{x}^*\|_{\mathcal{H}}^2$$

$$= \left( \frac{\max_{1 \le t \le T} \|\boldsymbol{x}_t - \boldsymbol{x}^*\|_{\mathcal{H}}^2}{2\eta} + \eta \right) \inf_{\boldsymbol{H} \in \mathcal{H}, \operatorname{Tr}(\boldsymbol{H}) \le 1} \sqrt{\langle \boldsymbol{M}_T, \boldsymbol{H}^{-1} \rangle}$$

$$\le \left( \frac{\max_{1 \le t \le T} \|\boldsymbol{x}_t - \boldsymbol{x}^*\|_{\mathcal{H}}^2}{2\eta} + \eta \right) \sqrt{\langle \boldsymbol{M}_T - \boldsymbol{M}_0, \boldsymbol{H}^{-1} \rangle + \langle \boldsymbol{M}_0, \boldsymbol{H}^{-1} \rangle}$$

for any $\boldsymbol{H} \in \mathcal{H}$ with $\operatorname{Tr}(\boldsymbol{H}) = 1$. In particular, we choose $\boldsymbol{H} = \alpha \boldsymbol{H}_T^* + \frac{1-\alpha}{d} \boldsymbol{I}_d$ for some $\alpha \in (0,1)$ where $\boldsymbol{H}_T^* = \arg\min_{\boldsymbol{H} \in \mathcal{H}, \operatorname{Tr}(\boldsymbol{H}) \le 1} \sqrt{\langle \boldsymbol{M}_T - \boldsymbol{M}_0, \boldsymbol{H}^{-1} \rangle}$, so $\langle \boldsymbol{M}_T - \boldsymbol{M}_0, \boldsymbol{H}^{-1} \rangle = \|\|\boldsymbol{g}_{1:T}\|\|_{\mathcal{H}}^2$ according to the definition of the adaptive gradient norm in (7). Since $\boldsymbol{H}^{-1} \preceq \frac{1}{\alpha} (\boldsymbol{H}_T^*)^{-1}$ and $\boldsymbol{H}^{-1} \preceq \frac{d}{1-\alpha} \boldsymbol{I}_d$, we further have

$$\sum_{t=1}^{T} L_t(\boldsymbol{x}_t) - \sum_{t=1}^{T} L_t(\boldsymbol{x}^*) \le \left( \frac{\max_{1 \le t \le T} \|\boldsymbol{x}_t - \boldsymbol{x}^*\|_{\mathcal{H}}^2}{2\eta} + \eta \right) \sqrt{\frac{1}{\alpha} \langle \boldsymbol{M}_T - \boldsymbol{M}_0, (\boldsymbol{H}_T^*)^{-1} \rangle + \frac{d}{1-\alpha} \langle \boldsymbol{M}_0, \boldsymbol{I}_d \rangle}$$

$$= \left( \frac{\max_{1 \le t \le T} \|\boldsymbol{x}_t - \boldsymbol{x}^*\|_{\mathcal{H}}^2}{2\eta} + \eta \right) \sqrt{\frac{1}{\alpha} \|\|\boldsymbol{g}_{1:T}\|\|_{\mathcal{H}}^2 + \frac{d^2 \epsilon}{1-\alpha}} .$$

Finally choosing $\alpha = \frac{\|\|\boldsymbol{g}_{1:T}\|\|_{\mathcal{H}}}{\|\|\boldsymbol{g}_{1:T}\|\|_{\mathcal{H}} + d\sqrt{\epsilon}}$, we obtain

$$\sum_{t=1}^{T} L_t(\boldsymbol{x}_t) - \sum_{t=1}^{T} L_t(\boldsymbol{x}^*) = \left( \frac{\max_{1 \le t \le T} \|\boldsymbol{x}_t - \boldsymbol{x}^*\|_{\mathcal{H}}^2}{2\eta} + \eta \right) \left( \|\|\boldsymbol{g}_{1:T}\|\|_{\mathcal{H}} + d\sqrt{\epsilon} \right)$$

This completes the proof. □

The proof for Corollary 3.5 is straightforward.

**Corollary D.1.** *Under the setting of Theorem 2.1, further suppose that $\mathcal{X}$ is a bounded set in $\mathbb{R}^d$. Then choosing $\eta = \sqrt{2} \|\mathcal{X}\|_{\mathcal{H}}$, the regret bound for Algorithm 1 becomes*

$$\sum_{t=1}^{T} L_t(\boldsymbol{x}_t) - \sum_{t=1}^{T} L_t(\boldsymbol{x}^*) \le 2\sqrt{2} \|\mathcal{X}\|_{\mathcal{H}} \left( G + d\sqrt{\epsilon} \right) .$$

*Proof of Corollary 3.5.* Note that $D = \max_{t \in [T]} \|\boldsymbol{x}_t - \boldsymbol{x}^*\|_{\mathcal{H}} \leq \max_{t \in [T]} \|\boldsymbol{x}_t\|_{\mathcal{H}} + \|\boldsymbol{x}^*\|_{\mathcal{H}} \leq 2\|\mathcal{X}\|_{\mathcal{H}}$ because $\boldsymbol{x}_1, \ldots, \boldsymbol{x}_T$ and $\boldsymbol{x}^*$ are all in $\mathcal{X}$. The proof is completed by setting $\eta = \sqrt{2}\|\mathcal{X}\|_{\mathcal{H}}$ to minimize $2\frac{\|\mathcal{X}\|_{\mathcal{H}}^2}{\eta} + \eta$. $\qquad\square$

### D.2. Convergence rate

Next, we present the proof for the convergence rate of Algorithm 1 with a well-structured preconditioner set $\mathcal{H}$.

**Theorem 3.8.** *Let $\mathcal{H}$ be a well-structured preconditioner set under Definition 3.1. Consider any independent stochastic convex loss functions $L_1, \ldots, L_T$ satisfying Assumption 3.7, and let $H(L, \mathcal{H})$ be the $\mathcal{H}$-smoothness of their expectation $L$. Suppose the global minimizer of $L$, denoted by $\boldsymbol{x}^*$, is in $\mathcal{X}$. Then for the iterates $\boldsymbol{x}_1, \ldots, \boldsymbol{x}_T$ of Algorithm 1, denoting $\bar{\boldsymbol{x}}_{1:T} = \frac{1}{T}\sum_{t=1}^{T}\boldsymbol{x}_t$, it holds that*

$$\mathbb{E}\left[L(\bar{\boldsymbol{x}}_{1:T}) - L(\boldsymbol{x}^*)\right]$$
$$\leq \frac{16}{T}\|\mathcal{X}\|_{\mathcal{H}}^2 H(L, \mathcal{H}) + \frac{4\sqrt{2}}{\sqrt{T}}\|\mathcal{X}\|_{\mathcal{H}}\sigma + \frac{4\sqrt{2}d\sqrt{\epsilon}}{T}\|\mathcal{X}\|_{\mathcal{H}}$$

*where $\sigma = \inf_{\boldsymbol{H} \in \mathcal{H}, \mathrm{Tr}(\boldsymbol{H}) \leq 1}\sqrt{\langle\boldsymbol{\Sigma}, \boldsymbol{H}^{-1}\rangle}$.*

*Proof of Theorem 3.8.* Let $\boldsymbol{H}^* \in \mathcal{H}$ be the matrix given in Definition 3.6 for the loss function $L$, such that $\nabla^2 L(\boldsymbol{x}) \preceq \boldsymbol{H}^*$. Therefore, for any $\boldsymbol{x}, \boldsymbol{x}' \in \mathcal{X}$,

$$L(\boldsymbol{x}') \leq L(\boldsymbol{x}) + \nabla L(\boldsymbol{x})^{\top}(\boldsymbol{x}' - \boldsymbol{x}) + \frac{1}{2}(\boldsymbol{x}' - \boldsymbol{x})^{\top}\boldsymbol{H}^*(\boldsymbol{x}' - \boldsymbol{x})$$
$$= L(\boldsymbol{x}) + \frac{1}{2}\left(\boldsymbol{x}' - \boldsymbol{x} + (\boldsymbol{H}^*)^{-1}\nabla L(\boldsymbol{x})\right)^{\top}\boldsymbol{H}^*\left(\boldsymbol{x}' - \boldsymbol{x} + (\boldsymbol{H}^*)^{-1}\nabla L(\boldsymbol{x})\right) - \frac{1}{2}\nabla L(\boldsymbol{x})^{\top}(\boldsymbol{H}^*)^{-1}\nabla L(\boldsymbol{x}).$$

Then we have that

$$L(\boldsymbol{x}^*) = \min_{\boldsymbol{x}'} L(\boldsymbol{x}')$$
$$\leq \min_{\boldsymbol{x}'} L(\boldsymbol{x}) + \frac{1}{2}\left(\boldsymbol{x}' - \boldsymbol{x} + (\boldsymbol{H}^*)^{-1}\nabla L(\boldsymbol{x})\right)^{\top}\boldsymbol{H}^*\left(\boldsymbol{x}' - \boldsymbol{x} + (\boldsymbol{H}^*)^{-1}\nabla L(\boldsymbol{x})\right) - \frac{1}{2}\nabla L(\boldsymbol{x})^{\top}(\boldsymbol{H}^*)^{-1}\nabla L(\boldsymbol{x})$$
$$= L(\boldsymbol{x}) - \frac{1}{2}\nabla L(\boldsymbol{x})^{\top}(\boldsymbol{H}^*)^{-1}\nabla L(\boldsymbol{x}).$$

Applying the above inequality to $\boldsymbol{x}_1, \ldots, \boldsymbol{x}_T$ and denoting $\bar{\boldsymbol{g}}_t = \nabla L(\boldsymbol{x}_t)$, we then have

$$\mathbb{E}\left[\sum_{t=1}^{T}(L(\boldsymbol{x}_t) - L(\boldsymbol{x}^*))\right] \geq \frac{1}{2}\mathbb{E}\left[\sum_{t=1}^{T}\bar{\boldsymbol{g}}_t^{\top}(\boldsymbol{H}^*)^{-1}\bar{\boldsymbol{g}}_t\right]$$
$$= \frac{1}{2}\left\langle\mathbb{E}\left[\sum_{t=1}^{T}\bar{\boldsymbol{g}}_t\bar{\boldsymbol{g}}_t^{\top}\right], (\boldsymbol{H}^*)^{-1}\right\rangle$$
$$= \frac{1}{2\mathrm{Tr}(\boldsymbol{H}^*)}\left\langle\mathbb{E}\left[\sum_{t=1}^{T}\bar{\boldsymbol{g}}_t\bar{\boldsymbol{g}}_t^{\top}\right], (\boldsymbol{H}^*/\mathrm{Tr}(\boldsymbol{H}^*))^{-1}\right\rangle$$
$$\geq \frac{1}{2H(L, \mathcal{H})}\inf_{\mathrm{Tr}(\boldsymbol{H}) \leq 1, \boldsymbol{H} \in \mathcal{H}}\left\langle\mathbb{E}\left[\sum_{t=1}^{T}\bar{\boldsymbol{g}}_t\bar{\boldsymbol{g}}_t^{\top}\right], \boldsymbol{H}^{-1}\right\rangle. \tag{27}$$

For any $\delta > 0$, we choose $\boldsymbol{H}_{\boldsymbol{g}} \in \mathcal{H}$ such that $\langle\mathbb{E}[\sum_{t=1}^{T}\bar{\boldsymbol{g}}_t\bar{\boldsymbol{g}}_t^{\top}], \boldsymbol{H}_{\boldsymbol{g}}^{-1}\rangle \leq \delta + \inf_{\boldsymbol{H} \in \mathcal{H}, \mathrm{Tr}(\boldsymbol{H}) \leq 1}\langle\mathbb{E}[\sum_{t=1}^{T}\bar{\boldsymbol{g}}_t\bar{\boldsymbol{g}}_t^{\top}], \boldsymbol{H}^{-1}\rangle$ and $\mathrm{Tr}(\boldsymbol{H}_{\boldsymbol{g}}) \leq 1$. Similarly, we choose $\boldsymbol{H}_{\boldsymbol{\Sigma}} \in \mathcal{H}$ such that $\langle\boldsymbol{\Sigma}, \boldsymbol{H}_{\boldsymbol{\Sigma}}^{-1}\rangle \leq \delta + \inf_{\boldsymbol{H} \in \mathcal{H}, \mathrm{Tr}(\boldsymbol{H}) \leq 1}\langle\boldsymbol{\Sigma}, \boldsymbol{H}^{-1}\rangle$ and $\mathrm{Tr}(\boldsymbol{H}_{\boldsymbol{\Sigma}}) \leq 1$. Correspondingly, we define $\boldsymbol{H}' = \alpha\boldsymbol{H}_{\boldsymbol{g}} + (1 - \alpha)\boldsymbol{H}_{\boldsymbol{\Sigma}}$, which satisfies that $\boldsymbol{H}' \in \mathcal{H}$ and $\mathrm{Tr}(\boldsymbol{H}') \leq 1$. Then by Jensen's inequality, we have

$$\mathbb{E}\inf_{\boldsymbol{H} \in \mathcal{H}, \mathrm{Tr}(\boldsymbol{H}) \leq 1}\sqrt{\langle\boldsymbol{M}_T - \boldsymbol{M}_0, \boldsymbol{H}^{-1}\rangle} \leq \mathbb{E}\sqrt{\langle\boldsymbol{M}_T - \boldsymbol{M}_0, (\boldsymbol{H}')^{-1}\rangle} \leq \sqrt{\mathbb{E}\langle\boldsymbol{M}_T - \boldsymbol{M}_0, (\boldsymbol{H}')^{-1}\rangle}.$$

Plugging in $M_T - M_0 = \sum_{t=1}^{T} g_t g_t^\top$, since $\mathbb{E}[\sum_{t=1}^{t} g_t g_t^\top] \preceq \mathbb{E}[\sum_{t=1}^{T} \bar{g}_t \bar{g}_t^\top] + T\Sigma$, we further have

$$\mathbb{E} \inf_{H \in \mathcal{H}, \mathrm{Tr}(H) \leq 1} \sqrt{\langle M_T - M_0, H^{-1} \rangle} \leq \sqrt{\left\langle \mathbb{E}\left[\sum_{t=1}^{T} \bar{g}_t \bar{g}_t^\top\right], (H')^{-1} \right\rangle + T \langle \Sigma, (H')^{-1} \rangle}$$

$$\leq \sqrt{\frac{1}{\alpha} \left\langle \mathbb{E}\left[\sum_{t=1}^{T} \bar{g}_t \bar{g}_t^\top\right], H_g^{-1} \right\rangle + \frac{1}{1-\alpha} T \langle \Sigma, H_\Sigma^{-1} \rangle}$$

$$= \sqrt{\left\langle \mathbb{E}\left[\sum_{t=1}^{T} \bar{g}_t \bar{g}_t^\top\right], H_g^{-1} \right\rangle} + \sqrt{T \langle \Sigma, H_\Sigma^{-1} \rangle}$$

where in the last step we choose $\alpha$ to be

$$\alpha = \frac{\sqrt{\langle \mathbb{E}[\sum_{t=1}^{T} \bar{g}_t \bar{g}_t^\top], H_g^{-1} \rangle}}{\sqrt{\langle \mathbb{E}[\sum_{t=1}^{T} \bar{g}_t \bar{g}_t^\top], H_g^{-1} \rangle} + \sqrt{T \langle \Sigma, H_\Sigma^{-1} \rangle}}.$$

Recall that $\sigma = \inf_{H \in \mathcal{H}, \mathrm{Tr}(H) \leq 1} \sqrt{\langle \Sigma, H^{-1} \rangle}$. Then it follows from (27) and the definitions of $H_g, H_\Sigma$ that

$$\mathbb{E}\left[ \inf_{H \in \mathcal{H}, \mathrm{Tr}(H) \leq 1} \sqrt{\langle M_T - M_0, H^{-1} \rangle} \right] \leq \sqrt{2H(L, \mathcal{H}) \cdot \mathbb{E}\left[\sum_{t=1}^{T}(L_t(x_t) - L_t(x^*))\right] + \delta} + \sqrt{T\sigma^2 + T\delta}$$

Since $\delta > 0$ is arbitrary, this implies that

$$\mathbb{E}\left[ \inf_{H \in \mathcal{H}, \mathrm{Tr}(H) \leq 1} \sqrt{\langle M_T - M_0, H^{-1} \rangle} \right] \leq \sqrt{2H(L, \mathcal{H}) \cdot \mathbb{E}\left[\sum_{t=1}^{T}(L_t(x_t) - L_t(x^*))\right]} + \sqrt{T\sigma^2}$$

Now recall the regret bound from Corollary 3.5, and it follows from the above inequality that

$$\mathbb{E}\left[\sum_{t=1}^{T}(L_t(x_t) - L_t(x^*))\right] \leq 2\sqrt{2}\|\mathcal{X}\|_{\mathcal{H}} \cdot \mathbb{E}\left[ \inf_{H \in \mathcal{H}, \mathrm{Tr}(H) \leq 1} \sqrt{\langle M_t - M_0, H^{-1} \rangle} + d\sqrt{\epsilon} \right]$$

$$\leq 2\sqrt{2}\|\mathcal{X}\|_{\mathcal{H}} \left( \sqrt{2H(L, \mathcal{H}) \cdot \mathbb{E}\left[\sum_{t=1}^{T}(L_t(x_t) - L_t(x^*))\right]} + \sqrt{T\sigma^2} + d\sqrt{\epsilon} \right).$$

Solving the above inequality yields

$$\mathbb{E}\left[\frac{1}{T}\sum_{t=1}^{T}(L_t(x_t) - L_t(x^*))\right] \leq \frac{16}{T}\|\mathcal{X}\|_{\mathcal{H}}^2 H(L, \mathcal{H}) + \frac{4\sqrt{2}}{\sqrt{T}}\|\mathcal{X}\|_{\mathcal{H}} \sigma + \frac{4\sqrt{2}}{T}\|\mathcal{X}\|_{\mathcal{H}} d\sqrt{\epsilon}.$$

This completes the proof. $\qquad\qquad\qquad\qquad\qquad\qquad\qquad\qquad\qquad\qquad\qquad\qquad\qquad\qquad\qquad\qquad\quad$ $\square$

### D.3. Analysis for EMA style optimizers

Here we present the proof for the theorems in Appendix C.

**Theorem C.1.** *Let $\mathcal{H}$ be a well-structured preconditioner set under Definition 3.1. Then for any convex loss functions $\tilde{L}_1, \ldots, \tilde{L}_T$, the regret of Algorithm 5 compared to any $x^* \in \mathcal{X}$ can be bounded as*

$$\sum_{t=1}^{T} \beta_2^{\frac{T-t}{2}} \left[ \tilde{L}_t(x_t) - \tilde{L}_t(x^*) \right] \leq \left( \frac{D^2}{2\tilde{\eta}} + \tilde{\eta} \right) \inf_{H \in \mathcal{H}, \mathrm{Tr}(H) \leq 1} \sqrt{\langle \tilde{M}_T, H^{-1} \rangle}$$

*where $D = \max_{t \in [T]} \|x_t - x^*\|_{\mathcal{H}}.$*

*Proof of Theorem C.1.* We can choose $L_t = \beta_2^{-\frac{t}{2}} \tilde{L}_t$, $\epsilon = \tilde{\epsilon}$ and $\eta = \tilde{\eta}$. Then we claim Algorithm 5 is equivalent to Algorithm 1 with hyperparameter $\epsilon, \eta$ and loss functions $\{L_t\}_{t=1}^T$, which will be shown by induction.

Assume $\tilde{x}_s = x_s$ for $s \le t$. Then we know $g_s = \nabla L_s(x_s) = \beta_2^{-\frac{s}{2}} \nabla \tilde{L}_s(\tilde{x}_s)$ for $s \le t$.

We consider the update in the step $t$ of Algorithm 5.

$$\tilde{M}_t = \sum_{i=1}^t \beta_2^{t-i} \tilde{g}_i \tilde{g}_i^\top + \beta_2^t \tilde{M}_0 = \beta_2^t \left[ \sum_{i=1}^t g_t g_t^\top + \epsilon I_d \right] = \beta_2^t M_t,$$

$$\tilde{H}_t = \arg\min_{\tilde{H} \in \mathcal{H}} \left\langle \tilde{M}_t, \tilde{H}^{-1} \right\rangle + \tilde{\eta}^2 \operatorname{Tr}(\tilde{H}) = \arg\min_{H \in \mathcal{H}} \left\langle \beta_2^t M_t, H^{-1} \right\rangle + \eta^2 \operatorname{Tr}(H) = \beta_2^{\frac{t}{2}} H_t,$$

$$\tilde{H}_t^{-1} \tilde{g}_t = \beta_2^{-\frac{t}{2}} H_t^{-1} \beta_2^{\frac{t}{2}} g_t = H_t^{-1} g_t,$$

$$\tilde{x}_{t+1} = \Pi_{\mathcal{X}}^{\tilde{H}_t} \left( \tilde{x}_t - \tilde{H}_t^{-1} \tilde{g}_t \right) = \Pi_{\mathcal{X}}^{\tilde{H}_t} \left( x_t - H_t^{-1} g_t \right) = \Pi_{\mathcal{X}}^{H_t} \left( x_t - H_t^{-1} g_t \right) = x_{t+1}.$$

Therefore, we can obtain the regret bound for Algorithm 4 with Theorem 3.4.

$$\sum_{t=1}^T \beta_2^{-\frac{t}{2}} \left[ \tilde{L}_t(x_t) - \tilde{L}_t(x^*) \right] = \sum_{t=1}^T [L_t(x_t) - L_t(x^*)]$$

$$\le \left( \frac{\max_{1 \le t \le T} \|x_t - x^*\|_{\mathcal{H}}^2}{2\eta} + \eta \right) \inf_{H \in \mathcal{H}, \operatorname{Tr}(H) \le 1} \sqrt{\langle M_T, H^{-1} \rangle}$$

$$= \beta_2^{-\frac{T}{2}} \left( \frac{\max_{1 \le t \le T} \|x_t - x^*\|_{\mathcal{H}}^2}{2\tilde{\eta}} + \tilde{\eta} \right) \inf_{H \in \mathcal{H}, \operatorname{Tr}(H) \le 1} \sqrt{\langle \tilde{M}_T, H^{-1} \rangle}$$

and

$$\sum_{t=1}^T \beta_2^{-\frac{t-1}{2}} \frac{\beta_2^{-\frac{1}{2}} - 1}{\beta_2^{-\frac{T}{2}} - 1} \left[ \tilde{L}_t(x_t) - \tilde{L}_t(x^*) \right] \le \frac{1 - \sqrt{\beta_2}}{1 - \beta_2^{\frac{T}{2}}} \left( \frac{\max_{1 \le t \le T} \|x_t - x^*\|_{\mathcal{H}}^2}{2\tilde{\eta}} + \tilde{\eta} \right) \inf_{H \in \mathcal{H}, \operatorname{Tr}(H) \le 1} \sqrt{\langle \tilde{M}_T, H^{-1} \rangle}$$

$\square$

**Theorem C.2.** *Let $\mathcal{H}$ be a well-structured preconditioner set under Definition 3.1. Consider any independent stochastic convex loss functions $\tilde{L}_1, \ldots, \tilde{L}_T$ satisfying Assumption 3.7, and let $H(\tilde{L}, \mathcal{H})$ be the $\mathcal{H}$-smoothness of their expectation $\tilde{L}$. Suppose the global minimizer of $L$, denoted by $x^*$, is in $\mathcal{X}$. Then for the iterates $x_1, \ldots, x_T$ of Algorithm 5, denoting $\bar{x}_{1:T} = (\sum_{t=1}^T \beta_2^{\frac{T-t}{2}})^{-1} \sum_{t=1}^T \beta_2^{\frac{T-t}{2}} x_t$, it holds that*

$$\mathbb{E}\tilde{L}(\bar{x}_{1:T}) - \tilde{L}_t(x^*) \le \frac{16}{\sum_{t=1}^T \beta_2^{\frac{T-t}{2}}} \|\mathcal{X}\|_{\mathcal{H}}^2 H(\tilde{L}, \mathcal{H}) + \frac{4\sqrt{2}}{\sqrt{\sum_{t=1}^T \beta_2^{T-t}}} \|\mathcal{X}\|_{\mathcal{H}} \sigma + \frac{4\sqrt{2}}{\sum_{t=1}^T \beta_2^{\frac{-t}{2}}} \|\mathcal{X}\|_{\mathcal{H}} d\sqrt{\epsilon}$$

*where $\sigma = \inf_{H \in \mathcal{H}, \operatorname{Tr}(H) \le 1} \sqrt{\langle \Sigma, H^{-1} \rangle}$.*

*Proof of Theorem C.2.* Similar to the proof of Theorem 3.8, we know

$$\tilde{L}(\tilde{x}_t) - \tilde{L}(\tilde{x}^*) \ge \left\langle \mathbb{E}\tilde{g}_t \mathbb{E}\tilde{g}_t^\top, (\tilde{H}^*)^{-1} \right\rangle$$

and

$$\sum_{t=1}^T \beta_2^{\frac{T-t}{2}} \tilde{L}(\tilde{x}_t) - \tilde{L}(\tilde{x}^*) \ge \left\langle \sum_{t=1}^T \beta_2^{\frac{T-t}{2}} \mathbb{E}\tilde{g}_t \mathbb{E}\tilde{g}_t^\top, (\tilde{H}^*)^{-1} \right\rangle$$

$$\ge \frac{1}{2H(\tilde{L}, \mathcal{H})} \inf_{\operatorname{Tr}(H) \le 1, H \in \mathcal{H}} \left\langle \sum_{t=1}^T \beta_2^{\frac{T-t}{2}} \mathbb{E}\tilde{g}_t \mathbb{E}\tilde{g}_t^\top, H^{-1} \right\rangle$$

For any $\delta > 0$, we choose $\boldsymbol{H_g} \in \mathcal{H}$ such that $\mathrm{Tr}(\boldsymbol{H_g}) \leq 1$ and

$$\left\langle \sum_{t=1}^{T} \beta_2^{\frac{T-t}{2}} \mathbb{E}\tilde{\boldsymbol{g}}_t \mathbb{E}\tilde{\boldsymbol{g}}_t^{\top}, \boldsymbol{H_g}^{-1} \right\rangle \leq \delta + \inf_{\boldsymbol{H} \in \mathcal{H}, \mathrm{Tr}(\boldsymbol{H}) \leq 1} \left\langle \sum_{t=1}^{T} \beta_2^{\frac{T-t}{2}} \mathbb{E}\tilde{\boldsymbol{g}}_t \mathbb{E}\tilde{\boldsymbol{g}}_t^{\top}, \boldsymbol{H}^{-1} \right\rangle.$$

We also choose $\boldsymbol{H_\Sigma} \in \mathcal{H}$ such that $\left\langle \boldsymbol{\Sigma}, \boldsymbol{H_\Sigma}^{-1} \right\rangle \leq \delta + \inf_{\boldsymbol{H} \in \mathcal{H}, \mathrm{Tr}(\boldsymbol{H}) \leq 1} \left\langle \boldsymbol{\Sigma}, \boldsymbol{H}^{-1} \right\rangle$ and $\mathrm{Tr}(\boldsymbol{H_\Sigma}) \leq 1$. We define $\boldsymbol{H}' = \alpha \boldsymbol{H_g} + (1-\alpha)\boldsymbol{H_\Sigma}$. Then $\boldsymbol{H}' \in \mathcal{H}$ and $\mathrm{Tr}(\boldsymbol{H}') \leq 1$. We have that

$$\mathbb{E}\inf_{\boldsymbol{H} \in \mathcal{H}, \mathrm{Tr}(\boldsymbol{H}) \leq 1} \sqrt{\left\langle \tilde{\boldsymbol{M}}_T - \beta_2^T \tilde{\boldsymbol{M}}_0, \boldsymbol{H}^{-1} \right\rangle} \leq \mathbb{E}\sqrt{\left\langle \tilde{\boldsymbol{M}}_T - \beta_2^T \tilde{\boldsymbol{M}}_0, (\boldsymbol{H}')^{-1} \right\rangle}$$

$$\leq \sqrt{\mathbb{E}\left\langle \tilde{\boldsymbol{M}}_T - \beta_2^T \tilde{\boldsymbol{M}}_0, (\boldsymbol{H}')^{-1} \right\rangle}$$

$$\leq \sqrt{\left\langle \sum_{t=1}^{T} \beta_2^{T-t} \mathbb{E}\tilde{\boldsymbol{g}}_t \mathbb{E}\tilde{\boldsymbol{g}}_t^{\top}, (\boldsymbol{H}')^{-1} \right\rangle + \sum_{t=1}^{T} \beta_2^{T-t} \left\langle \boldsymbol{\Sigma}, (\boldsymbol{H}')^{-1} \right\rangle}.$$

Now plugging in the definition of $\boldsymbol{H}'$, we obtain

$$\mathbb{E}\inf_{\boldsymbol{H} \in \mathcal{H}, \mathrm{Tr}(\boldsymbol{H}) \leq 1} \sqrt{\left\langle \tilde{\boldsymbol{M}}_T - \beta_2^T \tilde{\boldsymbol{M}}_0, \boldsymbol{H}^{-1} \right\rangle} \leq \sqrt{\left\langle \sum_{t=1}^{T} \beta_2^{T-t} \mathbb{E}\tilde{\boldsymbol{g}}_t \mathbb{E}\tilde{\boldsymbol{g}}_t^{\top}, (\alpha \boldsymbol{H_g})^{-1} \right\rangle + \sum_{t=1}^{T} \beta_2^{T-t} \left\langle \boldsymbol{\Sigma}, ((1-\alpha)\boldsymbol{H_\Sigma})^{-1} \right\rangle}$$

$$= \sqrt{\frac{1}{\alpha}\left\langle \sum_{t=1}^{T} \beta_2^{\frac{T-t}{2}} \mathbb{E}\tilde{\boldsymbol{g}}_t \mathbb{E}\tilde{\boldsymbol{g}}_t^{\top}, (\boldsymbol{H_g})^{-1} \right\rangle + \frac{1}{1-\alpha} \sum_{t=1}^{T} \beta_2^{T-t} \left\langle \boldsymbol{\Sigma}, (\boldsymbol{H_\Sigma})^{-1} \right\rangle}$$

$$\leq \sqrt{\left\langle \sum_{t=1}^{T} \beta_2^{\frac{T-t}{2}} \mathbb{E}\tilde{\boldsymbol{g}}_t \mathbb{E}\tilde{\boldsymbol{g}}_t^{\top}, (\boldsymbol{H_g})^{-1} \right\rangle} + \sqrt{\sum_{t=1}^{T} \beta_2^{T-t} \left\langle \boldsymbol{\Sigma}, (\boldsymbol{H_\Sigma})^{-1} \right\rangle}.$$

where in the last step we choose $\alpha$ to be

$$\alpha = \frac{\sqrt{\left\langle \sum_{t=1}^{T} \beta_2^{\frac{T-t}{2}} \mathbb{E}\tilde{\boldsymbol{g}}_t \mathbb{E}\tilde{\boldsymbol{g}}_t^{\top}, (\boldsymbol{H_g})^{-1} \right\rangle}}{\sqrt{\left\langle \sum_{t=1}^{T} \beta_2^{\frac{T-t}{2}} \mathbb{E}\tilde{\boldsymbol{g}}_t \mathbb{E}\tilde{\boldsymbol{g}}_t^{\top}, (\boldsymbol{H_g})^{-1} \right\rangle} + \sqrt{\sum_{t=1}^{T} \beta_2^{T-t} \left\langle \boldsymbol{\Sigma}, (\boldsymbol{H_\Sigma})^{-1} \right\rangle}}.$$

Then we have that

$$\mathbb{E}\inf_{\boldsymbol{H} \in \mathcal{H}, \mathrm{Tr}(\boldsymbol{H}) \leq 1} \sqrt{\left\langle \tilde{\boldsymbol{M}}_T - \beta_2^T \tilde{\boldsymbol{M}}_0, \boldsymbol{H}^{-1} \right\rangle} \leq \sqrt{2H(\tilde{L}, \mathcal{H})\left(\mathbb{E}\sum_{t=1}^{T} \beta_2^{\frac{T-t}{2}} [L_t(\boldsymbol{x}_t) - L_t(\boldsymbol{x}^*)]\right) + \delta} + \sqrt{\sum_{t=1}^{T} \beta_2^{T-t}} \sqrt{\sigma^2 + \delta}.$$

When taking $\delta$ to 0, we have that

$$\mathbb{E}\sum_{t=1}^{T} \beta_2^{\frac{T-t}{2}} \left[\tilde{L}_t(\boldsymbol{x}_t) - \tilde{L}_t(\boldsymbol{x}^*)\right] \leq 2\sqrt{2}D\left[\sqrt{2H(\tilde{L}, \mathcal{H})\left(\mathbb{E}\sum_{t=1}^{T} \beta_2^{\frac{T-t}{2}} [L_t(\boldsymbol{x}_t) - L_t(\boldsymbol{x}^*)]\right)} + \sqrt{\sum_{t=1}^{T} \beta_2^{T-t}}\sigma + \beta_2^{\frac{T}{2}} d\sqrt{\epsilon}\right]$$

and

$$\mathbb{E}\sum_{t=1}^{T} \beta_2^{\frac{T-t}{2}} \left[\tilde{L}_t(\boldsymbol{x}_t) - \tilde{L}_t(\boldsymbol{x}^*)\right] \leq 16D^2 H(\tilde{L}, \mathcal{H}) + 4\sqrt{2\sum_{t=1}^{T} \beta_2^{T-t} D\sigma + 4\sqrt{2}D\beta_2^{\frac{T}{2}} d\sqrt{\epsilon}}.$$

If we choose $\bar{\boldsymbol{x}}_{1:T} = \frac{1}{\sum_{t=1}^{T} \beta_2^{\frac{T-t}{2}}} \sum_{t=1}^{T} \beta_2^{\frac{T-t}{2}} \boldsymbol{x}_t$, then from the convexity of $\tilde{L}$ we know that

$$\mathbb{E}[\tilde{L}(\bar{\boldsymbol{x}}_{1:T}) - \tilde{L}(\boldsymbol{x}^*)] \leq \mathbb{E} \frac{1}{\sum_{t=1}^{T} \beta_2^{\frac{T-t}{2}}} \sum_{t=1}^{T} \beta_2^{\frac{T-t}{2}} \left[ \tilde{L}_t(\boldsymbol{x}_t) - \tilde{L}_t(\boldsymbol{x}^*) \right]$$

$$\leq \frac{1}{\sum_{t=1}^{T} \beta_2^{\frac{T-t}{2}}} \left( 16 D^2 H(\tilde{L}, \mathcal{H}) + 4\sqrt{2 \sum_{t=1}^{T} \beta_2^{T-t} D\sigma + 4\sqrt{2} D \beta_2^{\frac{T}{2}} d\sqrt{\epsilon}} \right)$$

$$\leq \frac{16}{\sum_{t=1}^{T} \beta_2^{\frac{T-t}{2}}} D^2 H(\tilde{L}, \mathcal{H}) + \frac{4\sqrt{2}}{\sqrt{\sum_{t=1}^{T} \beta_2^{T-t}}} D\sigma + \frac{4\sqrt{2}}{\sum_{t=1}^{T} \beta_2^{\frac{-t}{2}}} D d\sqrt{\epsilon}$$

This completes the proof. $\square$

## E. Proof for One-Sided Shampoo

### E.1. Proof for left smoothness

**Lemma E.1** (Left smoothness for one-sided Shampoo). *Let $\mathcal{H} = \mathcal{S}_+^{d_L} \otimes \boldsymbol{I}_{d_R}$ be the well-structured preconditioner set for one-sided Shampoo. Then the $\mathcal{H}$-smoothness $H(L, \mathcal{H})$ defined in Definition 3.6 is equal to the smallest number $H \geq 0$ such that there exists $\boldsymbol{H}_{d_L}^* \in \mathbb{R}^{d_L \times d_L}$ satisfying that $H = d_R \operatorname{Tr}(\boldsymbol{H}_{d_L}^*)$ and that for any $\boldsymbol{X}, \boldsymbol{\Delta} \in \mathbb{R}^{d_L \times d_R}$,*

$$\left| \nabla^2 L(\boldsymbol{X})[\boldsymbol{\Delta}, \boldsymbol{\Delta}] \right| \leq \left\langle \boldsymbol{H}_{d_L}^*, \boldsymbol{\Delta}\boldsymbol{\Delta}^\top \right\rangle.$$

*In this case, the $\mathcal{H}$-smoothness is denoted by $H_{\text{left}}(L)$.*

*Proof of Lemma 4.3.* First, for any $\boldsymbol{X}, \boldsymbol{\Delta} \in \mathbb{R}^{d_L \times d_R}$ and $\boldsymbol{x} = \overline{\operatorname{vec}}(\boldsymbol{X})$, we have

$$\overline{\operatorname{vec}}(\boldsymbol{\Delta})^\top \nabla^2 L(\boldsymbol{x}) \overline{\operatorname{vec}}(\boldsymbol{\Delta}) = \nabla^2 L(\boldsymbol{X})[\boldsymbol{\Delta}, \boldsymbol{\Delta}].$$

Therefore, given the Kronecker product form of all $\boldsymbol{H} \in \mathcal{H}$, to find $\boldsymbol{H} \in \mathcal{H}$ with the smallest trace such that $-\boldsymbol{H} \preceq \nabla^2 L(\boldsymbol{x}) \preceq \boldsymbol{H}$, it is equivalent to find $\boldsymbol{H} = \boldsymbol{H}_{d_L} \otimes \boldsymbol{I}_{d_R} \in \mathcal{H}$ with the smallest trace such that for any $\boldsymbol{\Delta} \in \mathbb{R}^{d_L \times d_R}$,

$$|\nabla^2 L(\boldsymbol{X})[\boldsymbol{\Delta}, \boldsymbol{\Delta}]| \leq \overline{\operatorname{vec}}(\boldsymbol{\Delta})^\top (\boldsymbol{H}_{d_L} \otimes \boldsymbol{I}_{d_R}) \overline{\operatorname{vec}}(\boldsymbol{\Delta})$$

$$= \operatorname{Tr}(\boldsymbol{\Delta}^\top \boldsymbol{H}_{d_L} \boldsymbol{\Delta})$$

$$= \langle \boldsymbol{H}_{d_L}, \boldsymbol{\Delta}\boldsymbol{\Delta}^\top \rangle.$$

Further note that $\operatorname{Tr}(\boldsymbol{H}) = d_R \operatorname{Tr}(\boldsymbol{H}_{d_L})$, and thus we conclude that it is equivalent to find $\boldsymbol{H}_{d_L} \in \mathcal{S}_+^{d_L}$ with the smallest trace such that the above inequality holds for all $\boldsymbol{\Delta} \in \mathbb{R}^{d_L \times d_R}$. This completes the proof. $\square$

### E.2. Proof for regret bound

**Theorem 4.2** (Regret bound for one-sided Shampoo). *For convex functions $L_1, \ldots, L_T$, the regret of one-sided Shampoo (Algorithm 2) compared to any $\boldsymbol{X}^* \in \mathbb{R}^{d_L \times d_R}$ satisfies*

$$\sum_{t=1}^{T} L_t(\boldsymbol{X}_t) - \sum_{t=1}^{T} L_t(\boldsymbol{X}^*) \leq \left( \frac{D_{\text{op}}^2}{2 d_R \eta} + \eta \right) \left( G + d\sqrt{\epsilon} \right),$$

*where $D_{\text{op}} = \max_{t \in [T]} \|\boldsymbol{X}_t - \boldsymbol{X}^*\|_{\text{op}}$ and $G = \sqrt{d_R} \operatorname{Tr}\left[ \left( \sum_{t=1}^{T} \boldsymbol{G}_t \boldsymbol{G}_t^\top \right)^{\frac{1}{2}} \right]$. When the domain $\mathcal{X}$ is bounded in operator norm, i.e., $\|\mathcal{X}\|_{\text{op}} < \infty$, further choosing $\eta = \sqrt{2/d_R} \|\mathcal{X}\|_{\text{op}}$, it holds*

$$\sum_{t=1}^{T} L_t(\boldsymbol{X}_t) - \sum_{t=1}^{T} L_t(\boldsymbol{X}^*)$$

$$\leq 2\sqrt{2} \|\mathcal{X}\|_{\text{op}} \left( \operatorname{Tr}\left[ \left( \sum_{t=1}^{T} \boldsymbol{G}_t \boldsymbol{G}_t^\top \right)^{\frac{1}{2}} \right] + \frac{d}{\sqrt{d_R}} \sqrt{\epsilon} \right).$$

*Proof of Theorem 4.2.* We will apply Theorem 3.4 to one-sided Shampoo. According to the analysis in Appendix B.4, Algorithm 1 with $\mathcal{H} = \left( \mathbb{R}^{d_L \times d_L} \otimes \boldsymbol{I}_{d_R} \right) \cap \mathcal{S}_+^d$ recovers one-sided Shampoo. We can plug in $\|\boldsymbol{x}\|_{\mathcal{H}} = \frac{\|\boldsymbol{X}\|_{\mathrm{op}}}{\sqrt{d_R}}$ and $\|\boldsymbol{g}_{1:T}\|_{\mathcal{H}} = \sqrt{d_R} \operatorname{Tr}\left[ \left( \sum_{t=1}^T \boldsymbol{G}_t \boldsymbol{G}_t^\top \right)^{\frac{1}{2}} \right]$ into Theorem 3.4 and get that

$$\sum_{t=1}^T L_t(\boldsymbol{X}_t) - \sum_{t=1}^T L_t(\boldsymbol{X}^*) \leq \sqrt{2} \left( \frac{D_{\mathrm{op}}^2}{2 d_R \eta} + \eta \right) \left( G + \min \left( d\sqrt{\epsilon}, \frac{d^2 \epsilon}{2G} \right) \right)$$

with $D_{\mathrm{op}} = \max_{t \in [T]} \|\boldsymbol{X}_t - \boldsymbol{X}^*\|_{\mathrm{op}}$ and $G = \sqrt{d_R} \operatorname{Tr}\left[ \left( \sum_{t=1}^T \boldsymbol{G}_t \boldsymbol{G}_t^\top \right)^{\frac{1}{2}} \right]$. $\qquad\square$

# F. Additional Results for Experiments

## F.1. Efficient implementation of full-matrix AdaGrad.

Directly applying full-matrix AdaGrad to this $10^6$-dimensional problem is impractical. Instead, we consider the eigendecomposition of $\boldsymbol{H}$ to be $\boldsymbol{U}^\top \boldsymbol{\Sigma} \boldsymbol{U}$ and define the transformation $\mathcal{T}(\boldsymbol{X}) = \boldsymbol{U}\boldsymbol{X}$. We further define $d$ orthogonal matrices $\boldsymbol{V}_1, \ldots, \boldsymbol{V}_d \in \mathbb{R}^{d \times d}$ such that the first row of $\boldsymbol{V}_i$ is in the same direction of $\mathcal{T}(\boldsymbol{X}^* - \boldsymbol{X}_0)_{i,:}$ and define $\boldsymbol{V} = \operatorname{diag}(\boldsymbol{V}_1, \ldots, \boldsymbol{V}_d)$. We can know that $\boldsymbol{V} \overline{\operatorname{vec}}(\mathcal{T}(\boldsymbol{X}^* - \boldsymbol{X}_0) = \boldsymbol{v} \otimes \boldsymbol{e}_1$ where $\boldsymbol{v}_i = \|\mathcal{T}(\boldsymbol{X}^* - \boldsymbol{X}_0)_{i,:}\|_2$ for $i \in [d]$.

Then it holds that

$$
\begin{aligned}
f(\boldsymbol{X}) = \left\langle \boldsymbol{H}, (\boldsymbol{X} - \boldsymbol{X}^*)(\boldsymbol{X} - \boldsymbol{X}^*)^\top \right\rangle &= \left\langle \boldsymbol{\Sigma}, (\boldsymbol{U}\boldsymbol{X} - \boldsymbol{U}\boldsymbol{X}^*)(\boldsymbol{U}\boldsymbol{X} - \boldsymbol{U}\boldsymbol{X}^*)^\top \right\rangle \\
&= \left\langle \boldsymbol{\Sigma}, (\mathcal{T}(\boldsymbol{X}) - \mathcal{T}(\boldsymbol{X}^*))(\mathcal{T}(\boldsymbol{X}) - \mathcal{T}(\boldsymbol{X}^*))^\top \right\rangle \\
&= \left\langle \boldsymbol{\Sigma} \otimes \boldsymbol{I}_d, \overline{\operatorname{vec}}(\mathcal{T}(\boldsymbol{X} - \boldsymbol{X}^*)^\top) \overline{\operatorname{vec}}(\mathcal{T}(\boldsymbol{X} - \boldsymbol{X}^*)^\top)^\top \right\rangle \\
&= \left\langle \boldsymbol{\Sigma} \otimes \boldsymbol{I}_d, \boldsymbol{V} \overline{\operatorname{vec}}(\mathcal{T}(\boldsymbol{X} - \boldsymbol{X}^*)^\top) \overline{\operatorname{vec}}(\mathcal{T}(\boldsymbol{X} - \boldsymbol{X}^*)^\top)^\top \boldsymbol{V}^\top \right\rangle
\end{aligned}
$$

Further denoting $\tilde{\boldsymbol{x}} = \boldsymbol{V} \overline{\operatorname{vec}}(\mathcal{T}(\boldsymbol{X})^\top)$ and $\tilde{\boldsymbol{y}} = \boldsymbol{V} \overline{\operatorname{vec}}(\mathcal{T}(\boldsymbol{X}_0)^\top)$, then we obtain

$$
\begin{aligned}
f(\boldsymbol{X}) &= \sum_{i=1}^d \sigma_i \left\| \tilde{\boldsymbol{x}}_{(i-1)d+1:id} - (\boldsymbol{v} \otimes \boldsymbol{e}_1)_{(i-1)d+1:id} \right\|_2^2 \\
&= \sum_{i=1}^d \sigma_i \left[ (\tilde{\boldsymbol{x}}_{(i-1)d+1} - \tilde{\boldsymbol{y}}_{(i-1)d+1} - \boldsymbol{v}_i)^2 + \sum_{j=2}^d (\tilde{\boldsymbol{x}}_{(i-1)d+j} - \tilde{\boldsymbol{y}}_{(i-1)d+j})^2 \right]
\end{aligned}
$$

Running full-matrix AdaGrad on $f(\boldsymbol{X})$ starting from $\boldsymbol{X}_0$ can be implemented equivalently by using $\tilde{\boldsymbol{x}}$ as variable starting from $\tilde{\boldsymbol{y}}$. Only $\tilde{\boldsymbol{x}}_{(i-1)d+1}$ will receive non-zero gradient so full-matrix AdaGrad actually only cares these $d$ coordinates, which reduces the original problem to a problem only with $d$ variables.

## F.2. Results for EMA algorithms

As mentioned in Section 5, we compare AdaSGD, Adam, one-sided EMA Shampoo and full-matrix AdaSGD, which are EMA version of AdaGrad-Norm, diagonal AdaGrad, one-sided Shampoo and full-matrix Adagrad. The results are plotted in Figure 2. We set $\beta_2 = 0.95$ and disable first-order momentum, i.e., $\beta_1 = 0$ in Adam.

We tried 60 learning rates between $1 \times 10^{-4}$ and $1 \times 10^2$. The relationship between loss and learning rate is shown in Figure 3.

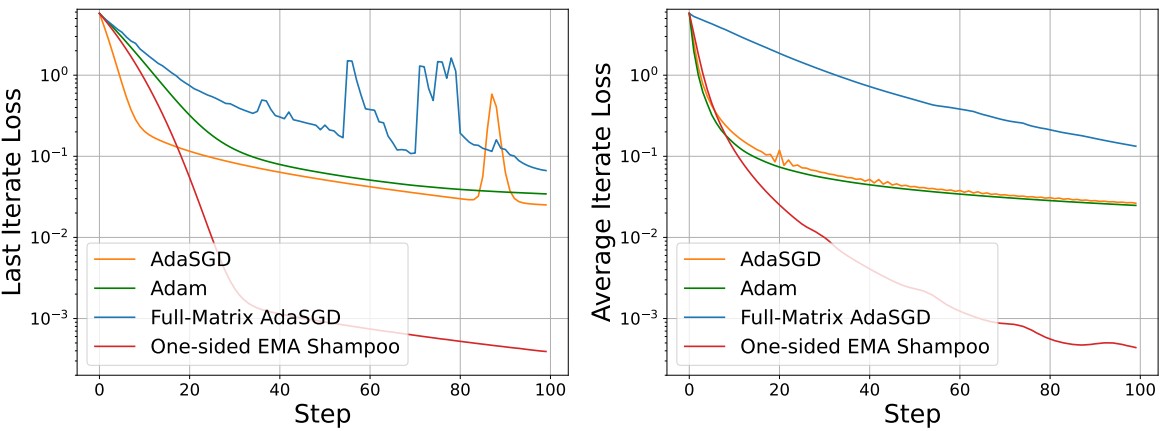

*Figure 2.* We plot the last iterate training loss $f(\boldsymbol{X}_t) = \left\langle \boldsymbol{H}, (\boldsymbol{X}_t - \boldsymbol{X}^*)(\boldsymbol{X}_t - \boldsymbol{X}^*)^\top \right\rangle$ and the average iterate training loss $f\left(\frac{1-\beta_2}{1-\beta_2^t} \sum_{s=1}^{t} \beta_2^{t-s} \boldsymbol{X}_s\right)$ over steps for optimizers obtained from Algorithm 5.

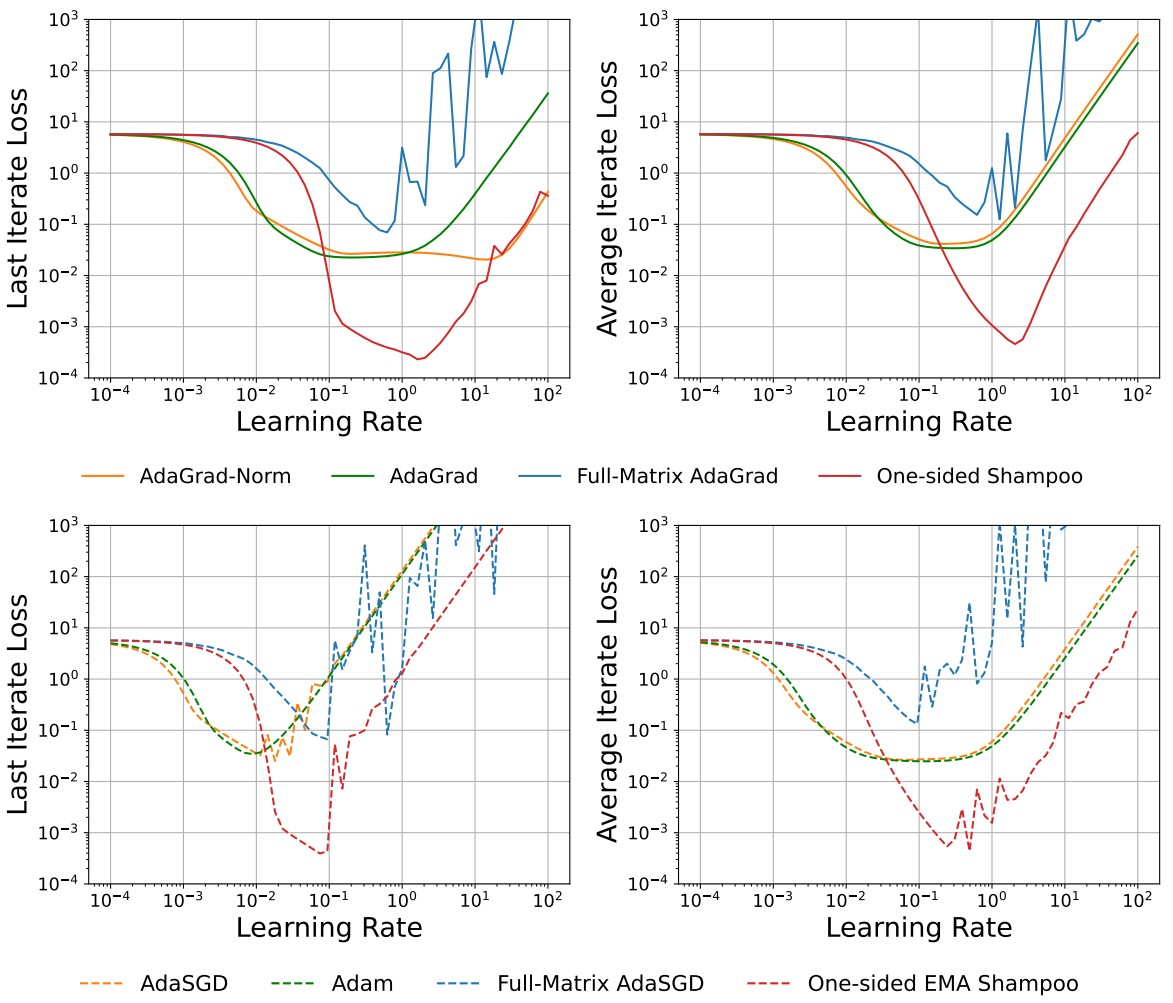

*Figure 3.* We plot the last iterate loss and average iterate loss versus learning rate. For each learning rate the plotted value is the average of last iterate loss and average of average iterate loss across five random seeds.

