# OpenReview forum: "Structured Preconditioners in Adaptive Optimization: A Unified Analysis"
_ICML.cc/2025/Conference — ICML 2025 poster_

### Official Review · Reviewer_tCyu · 2025-03-10

**Overall Recommendation:** 3

**Summary:**

This paper presents a unified analysis of adaptive optimization algorithms with structured preconditioners, challenging the assumption that better approximations of full-matrix Adagrad or less structured preconditioners always yield superior performance. The authors introduce "well-structured preconditioners" enabling a general regret bound, revealing a trade-off between domain metric and adaptive gradient norm. They demonstrate that a one-sided Shampoo variant achieves improved regret bounds and outperforms full-matrix Adagrad theoretically and empirically, suggesting simpler, more structured preconditioners can be more effective. The work provides a new perspective on adaptive optimizer design and insights for efficient large-scale training.

**Claims And Evidence:**

Most of the claims are well-evidenced, except for the claim that "Conceptually, our findings challenge the conventional wisdom that using a larger set of preconditioners which require more memory and compute leads to better optimization performance in terms of number of steps.". This is because of two major reasons: (a) The analysis of one-sided Shampoo focuses heavily on a particular type of loss function (Equation 7) and a simplified experimental setting. It's unclear if these findings generalize to other types of loss functions or more complex problems. (b) The experimental setup is highly specific and doesn't reflect the complexities of typical deep learning scenarios.

**Essential References Not Discussed:**

As far as I know, no other reference needs to be discussed.

**Experimental Designs Or Analyses:**

While the experimental design appears logically sound for the specific problem under consideration (quadratic optimization), the lack of realism raises concerns about the validity of extrapolating these results to more complex, real-world scenarios. The absence of experiments on standard deep learning datasets and models is a significant limitation.

**Methods And Evaluation Criteria:**

The proposed methods have theoretical merit, but the evaluation criteria are not sufficiently comprehensive to support strong claims about the practical benefits of one-sided Shampoo in real-world deep learning scenarios. More extensive and realistic experiments are needed.

**Other Comments Or Suggestions:**

While Section 3.3 demonstrates that the proposed well-structured preconditioner framework encompasses a range of existing adaptive optimizers, the paper misses an opportunity to provide comparative insights within this unified framework. A more detailed analysis of the relative strengths and weaknesses of different optimizers, as revealed by the framework itself, would have further strengthened the contribution.

**Other Strengths And Weaknesses:**

# Strengths
- The theoretical analysis is generally well-presented and easy to follow (assuming sufficient background knowledge).
# Weaknesses
- The lack of experiments on standard deep learning benchmarks limits the practical significance of the findings.
- The motivation for focusing on the specific quadratic problem in the experiments could be strengthened.

**Questions For Authors:**

Could you elaborate on the intuition behind the development of the well-structured preconditioner sets? What key insights led you to believe that these specific algebraic properties (closure under scalar multiplication, addition, and multiplication) would be sufficient to address the limitations in previous analyses?

**Relation To Broader Scientific Literature:**

The paper's key contributions relate to the broader literature in the following ways:

- Adaptive Optimization: Builds upon and refines existing adaptive optimization methods like AdaGrad, Adam, and Shampoo (Duchi et al., 2011; Kingma & Ba, 2014; Gupta et al., 2, particularly Gupta et al. (2017), but overcomes limitations by introducing the concept of "well-structured preconditioners."

- Shampoo Optimizer: Contributes to the understanding and improvement of the Shampoo optimizer (Gupta et al., 2018; Anil et al., 2020) by proposing and analyzing a one-sided variant with improved theoretical properties.

**Theoretical Claims:**

I haven't conducted a detailed verification of the proofs. However, the theoretical claims appear to be generally sensible and don't immediately contradict known principles in adaptive optimization. A rigorous validation of the proofs would be necessary for complete confirmation.

---

> ### Author Rebuttal · Authors · 2025-04-01
>
> We thank the reviewer for acknowledging the theoretical insight of our work. We will address your concerns below.
>
> **A larger set of preconditioner is not necessarily better:** We appreciate the reviewer’s concern regarding our claim. Due to computational limitations, it is indeed difficult to compare full-matrix AdaGrad with those structured preconditioners on real-world tasks. However, our intention is not to assert that simpler preconditioners always perform better, but rather to challenge the assumption that **a** **larger set is always superior**. We successfully show it is wrong via the theoretical analysis and synthetic task.
>
> **Practical advantage of one-sided Shampoo:** Thanks for the clarification. We realize that our paper might have unintentionally suggested that one-sided Shampoo has practical benefits on real-world tasks, which we do not claim. Our work is mainly theoretical, and our experiments are limited to a simplified setting that aligns with our analysis. We agree that demonstrating practical benefits would require larger-scale experiments, which is beyond the scope of this submission. We leave this for future work and welcome pointers to such work if it already exists. However, we want to clarify that the theoretical analysis (Theorem 3.3 and 3.7) holds for any convex functions so the analysis of one-sided Shampoo is not restricted to a specific type of function (equation 7).
>
> **Motivation for specific quadratic problem:** We propose a simple quadratic function so that we can easily compute those convergence rates in table 2 and verify the effectiveness of our theory. We agree it seems unclear why to choose such specific quadratic problem so we decide to add a new quadratic loss function on which the empirical results are consistent. We only introduce the loss function below and more details can be found in our response to reviewer N1Ca.
>
> We consider a synthetic linear regression problem $||\mathbf{A} \mathbf{X} - \mathbf{y}||\_2^2$ where $\mathbf{A}$ is the data matrix and $\mathbf{y} =\mathbf{A} \mathbf{X}^*$ is the label vector generated by ground-truth $\mathbf{X}^*$. Thus, the loss function can be equivalently written as $f(\mathbf{X}) = \langle \mathbf{H}, (\mathbf{X}-\mathbf{X}^*)(\mathbf{X}-\mathbf{X}^*)^\top \rangle$ for $\mathbf{H} = \mathbf{A}^\top  \mathbf{A}$, which is the same function that we studied in section 4.3 and show that 1-sided shampoo outperforms other adaptive algorithms.
>
> **Detailed comparison between optimizers:** We would like to elaborate on the comparative insights our theory can offer. As mentioned in line 74-80, line 222-231 and line 430-434, we should really consider the tradeoff between different terms in the regret bound or convergence rate. A less structured preconditioner such as full-matrix AdaGrad may have strength in having smaller $|||\mathbf{g}\_{1:t}|||\_\mathcal{H}$ and its weakness lies in inducing larger $||\mathcal{X}||_\mathcal{H}$.
>
> **Property of subalgebra:** The conditions for the well-structured $\mathcal{H}$ are required to ensure that $\mathbf{H}\_t\succeq \mathbf{H}\_{t-1}$. In particular, the condition for being closed under scalar multiplication and matrix addition (i.e., $\mathcal{K}$ is a linear subspace) and the condition for being closed under matrix multiplication are important in the proof of the desired property. Indeed, without the linear subspace condition, we have the failure example of two-sided Shampoo as discussed in Example 2.2 and Appendix A.3.1; moreover, without the matrix multiplication condition, we have the failure example of tridiagonal matrices as discussed in Example 2.3 and Appendix A.3.2. Also, as can be seen from the proofs in Appendix A.2, for any $\mathbf{H}\in\mathcal{H}$, we require $p(\mathbf{H}) \in \mathcal{H}$ for any polynomial $p$, which guarantees especially that $\mathbf{H}^{-1}\in\mathcal{H}$. All these requirements naturally inspire us to formulate the well-structured preconditioner sets using the notion of matrix subalgebra.

---

### Official Review · Reviewer_25wV · 2025-03-13

**Overall Recommendation:** 2

**Summary:**

The paper studies preconditioned adaptive methods for online convex optimization in a unified manner. They define a particular class of “well-structured” preconditioners that recovers all 3 variants of the AdaGrad and the Shampoo algorithm as special instances. The main take away message of the paper is that by selecting the set of preconditioners in some “well-structured” sense, the metric for the norm of the sum of gradient outer products and the distance to solution could be determined in a principled sense for achieving better regret bounds (under certain scenarios).

**Claims And Evidence:**

The main claim is that more structured preconditioners, i.e., the set of preconditioners which are relatively small (e.g., AdaGrad with scalar steo size or diagonalmatric preconditioner) could outperform less structured (e.g., AdaGrad with full-matrix preconditioner) versions. Their claim is supported for the Shampoo algorithm; they propose a new variant called one-sided Shampoo which essentially keeps track of only the “left-matrix”. They prove a regret bound which is better than the original shampoo by improving the dependence on the norm of the optimal solution (Frobenius to Spectral).

This results shows that a more principled idea of preconditioner design might help design new adaptive algorithm.

However, it is not clear how would such a design process work. For instance, the authors claim that regret bound for the full-matrix AdaGrad is worse than AdaGrad-norm or AdaGrad with diagonal preconditioner. However, there is no rigorous axplanation as to how this is the case. In fact, comparing the regret bounds from the original paper for the composite mirror descent update version, full-matrix AdaGrad had better regret; (i) it is robust against rotations unlike diagonal preconditioner and (ii) measures the variance of gradients across all coordinates as well as jointly between coordinates to adapt the step size per coordinate accordingly unlike the fixed step size rule for AdaGrad-Norm. The authors must be precise with their claim and provide an analytical verification.

**Essential References Not Discussed:**

-

**Experimental Designs Or Analyses:**

Yes. The experiments show some interesting results but unfortunately they are based on synthetic data (and for a matrix problem). Therefore, it is not enough to make any meaningful conclusion for the algorithms. I would be very surprised to see AdaGrad-norm beating diagonal version for a larger vector regression problem with real data (full-matrix would be too expensive).

**Methods And Evaluation Criteria:**

Yes. They compare methods using their regret bounds and dependence of those bounds to dimension, problem structures etc.

**Other Comments Or Suggestions:**

1.	I don’t agree that full-matrix AdaGrad is considered as the ideal preconditioning method and there is no mathematical evidence that supports the argument. Please update the first paragraph accordingly or verify the claim with references, examples or derivations.

2.	Please clarify the idea of structure for preconditioners clearly, before using the term for the first time in the introduction. In the current version, it is confusing as "structure" has different connotations in the context of preconditioned methods.

**Other Strengths And Weaknesses:**

Strengths:

1.	I think the authors’ attempt at unifying adaptive preconditioning using matrix sub-algebra on top of positive semi-definiteness is an interesting perspective into understanding the set structure of different preconditioned algorithm classes.

Weaknesses:

1.	The claims about AdaGrad-class of algorithms is not verified mathematically, as opposed to the proposed variant of Shampoo that has better regret than the original algorithm. Similarly, the experiments are designed for matrix problems and might not capture the advantages of preconditioning for AdaGrad family.

2.	Although I find the definition of well-structuredness genuine, it is not clear how we could use this framework to improve existing methods and develop new algorithm. The recipe is not clear, which the authors could explain more in the main text.

**Questions For Authors:**

1.	Have you tried running Shampoo and one-sided Shampoo? It would make more sense to see them side by side on a matrix problem, as it is not directly comparable to AdaGrad family in an apples to apples sense.

2.	Have you compared AdaGrad family of algorithm for a vector problem? It is more informative to observe their performance for a regression problem, for instance, with a real-world dataset in terms of the comparison of regret bounds.

3.	Can you prove that AdaGrad-norm has better regret bounds than other variants using your framework?

4.	Do you propose that there is a variation of AdaGrad family (similar to what you do for Shampoo) that will yield better regret guarantee?

**Relation To Broader Scientific Literature:**

Adaptive methods are the main workhorse in many large scale problems. For instance, adaptive methods outperform SGD for training transformer-based models.

1.	The paper attempts to unite the preconditioning of adaptive methods and understand their regret/convergence bounds based on how the preconditioner class is constructed.

2.	They explain the possible effects of structure (i.e., size of the set of preconditioners) under certain scenarios.

3.	They give a concrete example for Shampoo algorithm and propose a provably better version in terms of regret bounds.

**Theoretical Claims:**

I have checked the proof in details until Section 4 and also went over the proofs for Section 4. I haven’t spotted any mistakes.

---

> ### Author Rebuttal · Authors · 2025-04-01
>
> We sincerely thank the reviewer for recognizing the value of our unified framework. We will address your concern below.
>
> **Clarification of our results:** We never claim that the regret bound for the full-matrix AdaGrad is worse than AdaGrad-norm or AdaGrad with diagonal preconditioner and we apologize if any words cause such misunderstanding.  We only argue that it is a misleading belief that **adding more structures to the preconditioner will always lead to worse performance**. We use the specific comparison between one-sided Shampoo and full-matrix AdaGrad to illustrate this clearly. And we only focus on optimization speed, specifically characterized by regret bound or convergence rate. Other properties such rotation-equivariance are beyond the scope of this paper.
>
> **Common belief that full-matrix preconditioner is “ideal”:** First we would like to clarify we agree with the reviewer on the statement that full-matrix preconditioner is not necessarily ideal as there is no theoretical justification on that. However, this is a common belief held by many researchers in the community. For example, Agarwal et al., 2019 explicitly argues that full-matrix preconditioning allows for better exploitation of anisotropic curvature in loss landscapes and their experiments show that full-matrix version can work better than Adam in some cases. The belief that full-matrix preconditioners are “ideal” is also implicit in the design of many optimizers, such as Shampoo, and Shampoo^2, which all attempt to approximate the full-matrix version more efficiently.
>
> **Vector vs Matrix Problems:** The reviewer argues our comparison is unfair when using a matrix-structured problem to highlight the benefits of one-sided Shampoo. We disagree with this interpretation. First, vector and matrix problems are mathematically **equivalent** up to reshaping and vectorization. Second, the models in practice are often built by complicated matrix-based modules and thus our formulation is relevant to practice.
>
> Most importantly, we do **not** claim that one-sided Shampoo is better than the AdaGrad family in general. We only demonstrate that it can be better on certain losses since it already sufficiently supports our claim. We agree that AdaGrad family may perform better on simple vector problems, which doesn’t contradict our argument.
>
> **Comparison within AdaGrad family:** Our goal is **not** to provide a comprehensive comparison among all AdaGrad variants. As mentioned above, we never make any claims within the AdaGrad family. That said, we found that AdaGrad-Norm and diagonal AdaGrad can outperform full-matrix AdaGrad empirically from figure 1. We are also happy to clarify that AdaGrad-Norm can outperform full-matrix AdaGrad in some setting in response to the next question.
>
> **Can AdaGrad-Norm be the best:** Yes. We can show that the regret bound of AdaGrad-Norm is the best for a specific setting when $\mathcal{X}$ is the $\ell_2$-norm constrained ball. Due to space constraints, we are happy to share the complete proof in follow-up discussion.
>
> Even in practice, Xie et al., 2025 shows that AdaSGD (EMA version of AdaGrad-Norm) can outperform rotated Adam on a GPT-2 model. While rotated Adam differs from the original, it shows an important insight that supports our claim: **coordinate-wise adaptivity is not always better the global adaptivity**. We think this is even a better example than a regression problem with real data for which AdaGrad-Norm can beat diagonal AdaGrad.
>
> **Definition of Structure:** We agree that the notion of “structure” can be more clearly explained even though we have discussed this in line 29-40. In our usage, a structured preconditioner imposes constraints on the preconditioning, formalized via a subalgebra $\mathcal{K}$. As mentioned in the abstract and introduction, some examples of structured preconditioner includes layerwise, diagonal and kronecker-factored. In contrary, full-matrix preconditoner is the least structured preconditioner because there is no constraint.
>
> **Experiments on Shampoo:** Please see the response to reviewer N1Ca.
>
> **Insights for Developing new algorithms:** Please see the response to reviewer N1Ca.
>
> References
>
> 1. Efficient full-matrix adaptive regularization. (Agarwal et al., ICML 2019)
> 2. Adam Exploits $\ell_\infty$-geometry of Loss Landscape via Coordinate-wise Adaptivity. (Xie et al., ICLR 2025)

---

> > ### Comment · Reviewer_25wV · 2025-04-08
> >
> > **Clarification of our results:** To clarify my point, let’s consider the structure of the preconditioner and regret bounds particularly for AdaGrad-family. The regret bound for full-matrix is better than the others (ignoring the cost of full-matrix inversions). They might clearly coincide in certain scenarios, which are of little interest as such simple functions are not encountered in complex, non-convex network architectures. Therefore, I don’t think we are on the same page on this matter. Maybe you could explain your point with an example where the regret bound for a more structured method (AdaGrad-Norm) could be better than full-matrix AdaGrad.
> >
> > **Vector vs Matrix Problems:** Technically speaking, there is a difference between treating matrices as they are vs vectorizing them in the context of preconditioners. For instance, let’s take Shampoo. As Shampoo is clearly developed for matrix problems, writing it in the vectorized form gives us a matrix preconditioner which is written as the Kronecker product of 2 matrices, which is structurally different than AdaGrad-type updates for vectors. I understand that one can apply AdaGrad on matrices by flattening, but comparing it to a method which is designed for matrix-valued variables might not be fair.
> >
> > **Comparison within AdaGrad family:** One can see observe empirical gains for the more structured preconditioners on less structued ones, but this needs more detailed empirical analysis to explain the reason behind it. It is not completely clear to me how this behavior could be consistently observed or whether this is a tuning/initialization related matter?
> >
> > **Can AdaGrad-Norm be the best:** Could you please share the regret bounds for all three AdaGrad-family methods where AdaGrad-Norm is claimed to be the best?
> >
> > **Insights for Developing new algorithms:** Thank you for the clarification. Please include this discussion in the main text for the final version of the paper.

---

> > > ### Author Response · Authors · 2025-04-08
> > >
> > > **Clarification of our result:** To clarify, we only compare regret bounds shown in table 1, which matches the standard regret bound for AdaGrad family in previous papers. We restate the results below for reference.
> > >
> > > | Algorithm | Regret Bound |
> > > | --- | --- |
> > > | AdaGrad-Norm | $\|\|\mathcal{X}\|\|\_2 \sqrt{\sum_{t=1}^T \|\|\mathbf{g}_t\|\|_2^2}$ |
> > > | Diagonal AdaGrad | $\|\|\mathcal{X}\|\|\_\infty \sum_{i=1}^d \sqrt{\sum_{t=1}^T g_{t,i}^2 }$ |
> > > | Full-matrix AdaGrad | $\|\|\mathcal{X}\|\|\_2 \text{Tr} [(\sum_{t=1}^T \mathbf{g}_t \mathbf{g}_t^\top )^\frac{1}{2}]$ |
> > > | One-sided Shampoo | $\|\|\mathcal{X}\|\|\_{op} \text{Tr} [(\sum_{t=1}^T \mathbf{G}_t \mathbf{G}_t^\top )^\frac{1}{2}]$ |
> > >
> > > As we can see from this table, the regret bound of AdaGrad-Norm is always no larger than that of full-matrix AdaGrad, because it holds that $$\sqrt{\sum_{t=1}^T ||\mathbf{g}\_t||\_2^2} = \sqrt{\text{Tr} (\sum_{t=1}^T \mathbf{g}\_t \mathbf{g}\_t^\top)} \leq \text{Tr} [(\sum_{t=1}^T \mathbf{g}_t \mathbf{g}_t^\top)^{\frac{1}{2}}].$$ Here the inequality holds because $\sqrt{\text{Tr}(A)} \leq \text{Tr}(A^{\frac{1}{2}})$ for any positive semi-definite matrix $A$.
> > >
> > > **Can AdaGrad-Norm be the best:** Again we will focus on the regret bounds in table 1. As mentioned in our first response, when $\mathcal{X}$ is chosen to be the $\ell_2$-norm ball $\\{ ||\mathbf{x}||_2 \leq r \\}$, AdaGrad-Norm has the smallest regret bound. We have already proved that AdaGrad-Norm always has smaller regret bound than full-matrix AdaGrad for any choice of $\mathcal{X}$. We will compare with diagonal AdaGrad and one-sided Shampoo below.
> > >
> > > - Comparison with diagonal AdaGrad
> > >
> > > For this set, we have that $||\mathcal{X}||\_\infty=\max_{\mathbf{x} \in \mathcal{X}} ||\mathbf{x}||\_\infty = r =||\mathcal{X}||\_2$. On the other hand, it holds that $$\sqrt{\sum_{t=1}^T ||\mathbf{g}\_t||\_2^2} = \sqrt{\sum_{i=1}^d \sum_{t=1}^T g_{t,i}^2 } \leq \sum_{i=1}^d \sqrt{ \sum_{t=1}^T g_{t,i}^2 }. $$ The inequality is because $\sqrt{\sum_{i=1}^d a_i} \leq \sum_{i=1}^d \sqrt{a_i}$. Therefore, the regret bound of AdaGrad-Norm is no larger than diagonal AdaGrad for this choice of $\mathcal{X}$.
> > >
> > > - Comparison with one-sided Shampoo
> > >
> > > We start with computing $||\mathcal{X}||\_{op}$. For any matrix-valued $X \in \mathcal{X}$, $||X||\_{op} \leq ||X||\_F = ||\text{vec}(X)||\_2=r$. When $X$ only has one non-zero element $X\_{1,1}=r$, its operator norm is exactly $r$. So $||\mathcal{X}||\_{op} = \max_{\mathbf{x} \in \mathcal{X}} ||\mathbf{x}||\_{op}=r$. On the other hand, it holds that $$\sqrt{\sum_{t=1}^T ||\mathbf{g}\_t||\_2^2} = \sqrt{\text{Tr} \sum\_{t=1}^T \mathbf{G}\_t \mathbf{G}\_t^\top } \leq \text{Tr} [(\sum_{t=1}^T \mathbf{G}_t \mathbf{G}_t^\top)^{\frac{1}{2}}]. $$ The inequality is again because $\sqrt{\text{Tr}(A)} \leq \text{Tr}(A^{\frac{1}{2}})$ for any PSD matrix $A$.
> > >
> > > **Vector vs Matrix Problem:** Our example above can be equivalently cast as a vector problem, thus providing a concrete example of vector problems that shows the advantage of AdaGrad-Norm using a more structured preconditioner.
> > >
> > > **Empirical Comparison:** We are glad that you agree with the empirical gains of more structured preconditioner. We don’t expect or aim to see more structured preconditioner **consistently** works better. In contrast, **what we argue is that more structured preconditioner is not necessarily always worse**, and we already see several empirical examples that can support the claim we draw based on theoretical analysis.
> > >
> > > **Insights for developing new algorithms:** We will include the discussion in the revision.

---

### Official Review · Reviewer_N1Ca · 2025-03-14

**Overall Recommendation:** 3

**Summary:**

The paper provides regret bounds for a family of adaptive algorithms with structured preconditioner matrices. The analysis generalizes the technique introduced in the original Shampoo work and applies to Adagrad, Adagrad-norm, Adagrad-diag and one-sided Shampoo. Intriguingly, the paper shows that, for certain loss functions, the bound provided by one-sided Shampoo could be tighter than the bound for full-matrix Adagrad and also demonstrates this empirically on a simplified loss surface.

**Claims And Evidence:**

The paper mostly has theoretical contributions and provides a simple empirical experiment supporting the theory.

**Essential References Not Discussed:**

N/A

**Experimental Designs Or Analyses:**

N/A

**Methods And Evaluation Criteria:**

N/A

**Other Comments Or Suggestions:**

I would suggest adding two sided Shampoo and Shampoo^2 (based on the recent connections shown by Morwani et al. 2024) to the experimental plot.

**Other Strengths And Weaknesses:**

The main strength of the work is proposing a general unified analysis for various preconditioner matrices, and showing that the bounds for a more general class could be worse than a structured one. Although this should not be considered a weakness, the analysis does not lead to an improved algorithm. This could be considered as a part of future work.

**Questions For Authors:**

1. What do the authors think are plausible directions for using these bounds in designing better optimizers?

**Relation To Broader Scientific Literature:**

This work is well placed as a generalization of previous analysis of structured preconditioner matrices.

**Theoretical Claims:**

I took a brief look and verified (not thoroughly) all the proofs in the Appendix

---

> ### Author Rebuttal · Authors · 2025-04-01
>
> We thank the reviewer for acknowledging the value of our unified analysis. Below we will address your concerns.
>
> **Comparison with Shampoo and Shampoo^2:** We appreciate the suggestion to include two-sided Shampoo and Shampoo^2 in the experiments. As reviewer tCyu pointed out there is lack of motivation for our current experiment, we conduct experiments on a better setting which is more practical and better motivated by our theory analysis. We will add this experiment and focus more on it in the future revision.
>
> We consider a synthetic linear regression problem $||\mathbf{A} \mathbf{X} - \mathbf{y}||\_2^2$ where $\mathbf{A}$ is the data matrix and $\mathbf{y} =\mathbf{A} \mathbf{X}^*$ is the label vector generated by ground-truth $\mathbf{X}^*$. Thus, the loss function can be equivalently written as $f(\mathbf{X}) = \langle \mathbf{H}, (\mathbf{X}-\mathbf{X}^*)(\mathbf{X}-\mathbf{X}^*)^\top \rangle$ for $\mathbf{H} = \mathbf{A}^\top  \mathbf{A}$, which is the same function that we studied in section 4.3 and show that 1-sided shampoo outperforms other adaptive algorithms. We consider $\mathbf{X} \in \mathbb{R}^{d \times d}$ with $d=10^3$. We set the eigenvalues of $\mathbf{H}$ by $\sigma_1 = \cdots = \sigma_{10}=1$ and $\sigma_i = \frac{1}{(i-10)^2}$ for $11 \leq i \leq 10^3$. Each element of the solution $\mathbf{X}^*$ is independently sampled from $\mathcal{N}(0, \frac{1}{d})$.
>
> We compared all the optimization algorithms by sweeping learning rate from 1e-4 to 1e2. The plots are [here](https://docs.google.com/document/d/e/2PACX-1vR37RVh5tPyZFbelExgmMfoXk_y7Egv0TvxXw4WqzVJ9zYHECzzRYjv-2zE_QGwcbkBYvXFCSSrweVl/pub). The results are consistent with the original experiment: (1). **one-sided Shampoo outperforms other variants of AdaReg;** **(2). Shampoo is slightly worse than one-sided Shampoo**; **(3). Shampoo^2 fails to make progress in optimizing this loss function.**  We are unsure why Shampoo^2 underperforms on this loss function but note that Morwani et al. (2024) also does not provide empirical evidence of the effectiveness of Shampoo^2. Investigating this behavior is beyond the scope of our current work, but may be a worthwhile direction for future study. The implementation of Shampoo and Shampoo^2 is [here](https://limewire.com/d/4fwuX#gtS4Jg1Wpi).
>
> **One-sided Shampoo as a new, improved algorithm:** We disagree with the reviewer’s opinion that this work doesn’t lead to an improved algorithm. To the best of our knowledge, we are the first to explicitly define and analyze the one-sided Shampoo algorithm even though it is informally mentioned on social media. And our section 4.2 shows that it improves original Shampoo theoretically and section 4.3 shows it can achieve the best rate on some specific loss functions, which is empirically verified by our added experiment above.
>
> **Plausible directions for designing better optimizer:** Thanks for this insightful question. We believe the definition of well-structured preconditioners can help designing novel optimization algorithms as discussed in line 261-308 though they may not necessarily be better.
>
> Notably, our framework already encompasses recent algorithms such as Adam-mini (Zhang et al., 2024) and Adalayer (Zhao et al., 2024) via specific choices of the subalgebra $\mathcal{K}$. Inspired by these two examples, we identify three basic operations that can construct new matrix subalgebras that can be used for defining well-structured preconditioner.
>
> 1. **Direct sum** of matrix subalgebras. This is defined in line 261-308.
> 2. **Kronecker product** between a matrix subalgebra and identity matrix. For a matrix subalgebra $\mathcal{K}$, we can define $\mathcal{K}’$ by $\mathcal{K} \otimes \mathbf{I}_d = \\{ \mathbf{A} \otimes \mathbf{I}_d | \mathbf{A} \in \mathcal{K}\\}$.
> 3. **Rotations** via orthogonal matrices. For a matrix subalgebra $\mathcal{K}$ and an orthogonal matrix $\mathbf{U}$, we can define $\mathcal{K’}$ by $\mathbf{U}^\top \mathcal{K} \mathbf{U} = \\{ \mathbf{U}^\top \mathbf{A} \mathbf{U} | \mathbf{A} \in \mathcal{K} \\}$. Such $\mathcal{K’}$ is still a matrix subalgebra.
>
> These operations offer a principled way to design new preconditioners and optimizers. As noted in our conclusion (lines 430–435), a theory-driven strategy would be to evaluate terms in the regret bound or convergence rate under different $\mathcal{K}$ and choose $\mathcal{K}$ to optimize the trade-off. While this remains challenging to evaluate empirically on large models, we view this as an interesting direction for future work.

---

### Decision · Program_Chairs · 2025-05-01

**Decision:**

Accept (poster)

**Comment:**

The paper provides a unified analysis for establishing regret bounds for adaptive gradient methods with structured preconditioners. The analysis is a generalization of the analysis in the original Shampoo work and it applies to widely studied variants of Adagrad and Shampoo.  The analysis shows improved guarantees for a one-sided version of Shampoo. Additionally, the analysis shows that the regret bounds for a more general class of preconditioners can be worse than a more structured class.

The reviewers appreciated the contribution and the new insights derived from the proposed analysis. Following the discussion, all reviewers expressed support for the paper but there remained some concerns. In particular, one of the reviewers noted during the discussion that the arguments on the regret rates and particular scenarios of comparison are straightforward, and that the empirical results are not extensive enough to draw a clear conclusion on the effectiveness of structure in pre-conditioner design.